# RNA binding protein ZCCHC24 promotes tumorigenicity in triple-negative breast cancer

Yutaro Uchida [ID][1], Ryota Kurimoto[1], Tomoki Chiba[1], Takahide Matsushima[1], Goshi Oda[2], Iichiroh Onishi[3], Yasuto Takeuchi[4,5], Noriko Gotoh [ID][4,5] & Hiroshi Asahara [ID][1,6 ✉]

## Abstract

**Triple-negative breast cancer (TNBC) lacks the expression of hormone and HER2 receptors and is highly malignant with no effective therapeutic targets. In TNBC, the cancer stem-like cell (CSC) population is considered to be the main cause of resistance to treatment. Thus, the therapeutic targeting of this population could substantially improve patient survival. Here, we identify the RNA-binding protein ZCCHC24 as enriched in the mesenchymal-like TNBC population. ZCCHC24 promotes the expression of a set of genes related to tumorigenicity and treatment resistance by directly binding to the *cis*-element "UGUWHWWA" in their mRNAs, thereby stabilizing them. One of the ZCCHC24 targets, ZEB1, is a transcription factor that promotes the expression of cancer stemness genes and reciprocally induces ZCCHC24 expression. ZCCHC24 knockdown by siRNAs shows a therapeutic effect and reduces the mesenchymal-like cell population in TNBC patient-derived xenografts. ZCCHC24 knockdown also has additive effects with the BET inhibitor JQ1 in suppressing tumor growth in TNBC patient-derived xenografts.**

**Keywords** Breast Cancer; Cancer Stem Cells; mRNA Stabilization; RNA Binding Protein; ZEB1
**Subject Categories** Cancer; Chromatin, Transcription & Genomics; RNA Biology

## Introduction

Breast cancer is highly prevalent in women worldwide. Triple-negative breast cancer (TNBC), which lacks the expression of hormones and HER2 receptors, accounts for 20% of all breast cancers. TNBC is known for its biological malignancy, higher recurrence ratio, and shorter survival time after recurrence than the other breast cancer subtypes (Bianchini et al, 2022; Loibl et al, 2021). Therefore, there is an urgent need to identify new molecular targets for the treatment of TNBC.

Cancer cell populations with high tumorigenicity and treatment resistance have been described in solid tumors. In breast cancer, one such cell population is characterized by several molecular markers (CD44 positive, NRP1 high, ALDH1 positive, and CD24 low) and is often referred to as the cancer stem-like cell (CSC) population or tumor-initiating cells (Al-Hajj et al, 2003; Bianchini et al, 2016; Ginestier et al, 2007; Tominaga et al, 2019; Zhang et al, 2019). Normal stem cells within organs and tissues require a specific cellular environment called a niche for their maintenance and receive signals from this niche to enable them to maintain their cellular state (Hicks and Pyle, 2023). However, in the context of the CSC population, cancer cells have been suggested to gain plasticity and generate a population of treatment-resistant cells through specific transcriptional and post-transcriptional gene expression control mechanisms (Kim et al, 2018).

We focused on cellular gene expression networks across multiple levels to identify these mechanisms, including the RNA hierarchy tightly regulated by RNA-binding proteins (RBPs). Aberrant RBP-RNA interactions have garnered attention for their implications in the progression of multiple types of cancer (Dixit et al, 2021; Einstein et al, 2021; Park et al, 2015). However, the potential roles of RBPs in TNBC treatment resistance and CSC populations have not been studied.

This study identified ZCCHC24 as an RBP predominantly upregulated in the mesenchymal-like TNBC population. ZCCHC24 specifically stabilizes and upregulates the mRNA expression of key genes involved in tumorigenicity and treatment resistance. Importantly, we also showed that ZCCHC24 transcription was promoted by the common epithelial-mesenchymal transition (EMT) transcription factor ZEB1, which is one of the targets of ZCCHC24-mediated post-transcriptional regulation and provides an auto-amplified gene expression circuit. Owing to the efficacy of targeting ZCCHC24 through siRNA treatment in the TNBC patient-derived xenografts (PDX) models, ZCCHC24 shows potential as an effective therapeutic target in TNBC.

[1]Department of Systems Biomedicine, Institute of Science Tokyo, Tokyo 113-8510, Japan. [2]Department of Surgery, Breast Surgery, Institute of Science Tokyo, Tokyo 113-8510, Japan. [3]Department of Comprehensive Pathology, Institute of Science Tokyo, Tokyo 113-8510, Japan. [4]Division of Cancer Cell Biology, Kanazawa University, Kanazawa 920-1192, Japan. [5]Institute for Frontier Science Initiative, Kanazawa University, Kanazawa 920-1192, Japan. [6]Department of Molecular and Cellular Biology, Scripps Research, La Jolla, CA 92037, USA. ✉E-mail: asahara.syst@tmd.ac.jp

# Results

## ZCCHC24 expression is enriched in a mesenchymal-like cell population

To gain insight into the cell populations that shape the pathogenesis, scRNA-seq analysis of PDX was performed. We aimed to identify RBPs specifically expressed within a mesenchymal-like population that included cells with stemness potential (Guo et al, 2012; Mani et al, 2008; Puisieux et al, 2014; Ye et al, 2015). PDX was roughly divided into three populations: an epithelial-like population characterized by the expression of *CD24*, *KRT8*, and *KRT19*; a mesenchymal-like population characterized by the expression of *NRP1*, *ZEB1*, *JUN*, *VIM*, and *ALDH1A3*, and low expression of *CD24* and *TGFB1*; and a TGFB1 positive population, characterized by the expression of *TGFB1*, *CDC42* (Azios et al, 2007; Keely et al, 1997; Zhang et al, 2014), and *ITGA6* (Vassilopoulos et al, 2014) and low expression of *ZEB1* (Fig. 1A–C; Appendix Fig. S1).

We also reanalyzed the previously reported scRNA-seq data of TNBC tumor samples from five patients (Wu et al, 2020). We identified a mesenchymal-like population expressing *CD44*, *ZEB1*, and *NRP1* with low levels of *CD24* expression (Appendix Fig. S2A,B). In total, 240 genes were identified as commonly expressed in mesenchymal-like populations in PDX and from the reanalysis of TNBC data. Among these genes, we identified 17 genes annotated with the gene ontology of "RNA binding" (Fig. 1D; Dataset EV1). Among these 17 "RNA-binding" genes, *RBFOX2* (Braeutigam et al, 2014; Maurin et al, 2023) and *NOVA1* (Qu et al, 2022) have been reported to be essential for maintaining cancer stemness and are known for their roles in the EMT. In addition, *VIM*, *DCN*, and *COL14A1* are known for their roles as extracellular matrices rather than RBPs. Notably, the function of *ZCCHC24* has not been well characterized, and its function in cancer remains unknown.

As *ZCCHC24* showed a specific expression pattern (*P* value as a marker gene: 1.79E-169) in the mesenchymal-like population in both scRNA-seq datasets, we evaluated its potential role in tumorigenicity and treatment resistance in TNBC (Fig. 1E; Appendix Fig. S2C,D).

## ZCCHC24 stabilizes mRNAs encoding proteins that characterize the CSC population

First, to examine the function of ZCCCH24 in regulating gene expression in TNBC-related cells, we knocked down ZCCHC24 using siRNA in the mesenchymal-like basal TNBC cell line MDAMB231 or PDX and performed RNA-seq. This revealed that the expression of a set of genes critical for the characterization of CSC populations, such as *CD44*, *NRP1*, *ZEB1*, and *NOTCH2*, and a set of genes important for cancer progression and invasion, such as *JAG1*, *ADAMTS1*, and *ACVR1*, were specifically and significantly downregulated (Fig. 2A; Dataset EV2). Quantitative PCR analyses of MDAMB231 cells, the claudin-low TNBC cell line HCC38, and PDX knockdown with siRNAs against *ZCCHC24* also demonstrated downregulation of *CD44*, *NRP1*, *ZEB1*, *NOTCH2*, and *ADAMTS1* expression. In contrast, overexpression of *ZCCHC24* in MDAMB231 cells increased the expression of *NOTCH2*, *ZEB1*, *CDH11*, and *ZEB1* (Figs. 2B and EV1A; Appendix Fig. S3).

Moreover, analysis of cell-surface markers by flow cytometry showed a decrease in the CD44-positive and NRP1-positive populations in MDAMB231, HCC38, and PDX upon the knockdown of *ZCCHC24* (Fig. EV1B; Appendix Fig. S4). Downregulation of the target genes of ZCCHC24 was also confirmed at the protein level by western blotting (Fig. EV1C).

ZCCHC24 is a CCHC zinc finger-type RNA-binding protein that may be associated with specific mRNAs and may contribute to their stability. To test the potential role of ZCCHC24 in mRNA stability, we performed BRIC-Seq on MDAMB231 treated with siRNAs against ZCCHC24. BRIC-Seq is a method for analyzing transcriptome-wide mRNA stability by incubating cells with bromouridine (BrU) and then analyzing BrU-labeled mRNA at certain times after BrU removal (in this case, 0, 1, and 2 h) using RNA-seq (Fig. 2C) (Imamachi et al, 2014). We observed reduced mRNA stability of the genes involved in tumorigenicity and treatment resistance (Fig. 2D; Dataset EV3), including *NOTCH2*, *CD44*, *ZEB1*, *CDH11*, and *ADAMTS1*. Based on the RNA-Seq results, 93 genes with destabilized transcripts were identified as differently expressed genes (DEGs) with destabilized transcripts (Dataset EV3). Gene Ontology (GO) analysis of these 93 genes with DAVID showed that genes function with "negative regulation of the apoptotic process," "angiogenesis," "cell-matrix adhesion," and "cell migration," which are important for cancer stemness, are regulated by ZCCHC24 (Fig. EV1D). To confirm these results, we analyzed changes in RNA stability after stopping transcription by adding actinomycin D. The results showed that the knockdown of *ZCCHC24* decreased the stability of mRNAs derived from genes important for tumorigenicity, tumor progression, and treatment resistance, such as *ZEB1*, *NRP1*, *CD44*, *NOTCH2*, *CDH11*, *ADAMTS1*, and *JAG1* (Fig. 2E; Appendix Fig. S5). These results suggest that ZCCHC24 stabilizes the mRNAs encoding proteins that are important for the characterization of the CSC population.

## ZCCHC24 stabilizes target mRNAs by directly binding to specific *cis*-elements

To determine whether ZCCHC24 directly regulates mRNA stability, we performed PAR-CLIP in ZCCHC24-expressing MDAMB231 cells to identify the mRNA-binding sites of ZCCHC24 (Fig. 3A; Appendix Fig. S6A). ZCCHC24 binds predominantly to 3'UTRs on mRNAs (Fig. 3B,C). Moreover, motif analysis for peak-called regions revealed that ZCCHC24 bound to "UGUWHWWA" motifs on mRNAs (Fig. 3D). The violin plot also showed that the number of binding sites of ZCCHC24 correlated with expression changes upon ZCCHC24 knockdown (Fig. 3E; Dataset EV4), and the motif sites within 3'UTR of mRNAs were enriched in downregulated DEGs in transcriptome analysis (Appendix Fig. S6B; Dataset EV5), supporting the validity of the analysis. Importantly, peak binding sites were observed for ZCCHC24 on the motif sequences on 3'UTRs of genes critical for the characterization of the CSC population (Fig. 3F). We identified 31 pure target genes (bound in PAR-CLIP, stabilized or destabilized in BRIC-Seq, and upregulated or downregulated in RNA-Seq), including ZEB1 and NOTCH2, that were positively regulated by ZCCHC24 expression, and 16 genes that were negatively regulated by ZCCHC24 expression (Dataset EV6). We also confirmed the binding of ZCCHC24 to the mRNAs of *ZEB1* and *NOTCH2* using an RNA-immunoprecipitation (RIP) assay; however, the mRNAs of

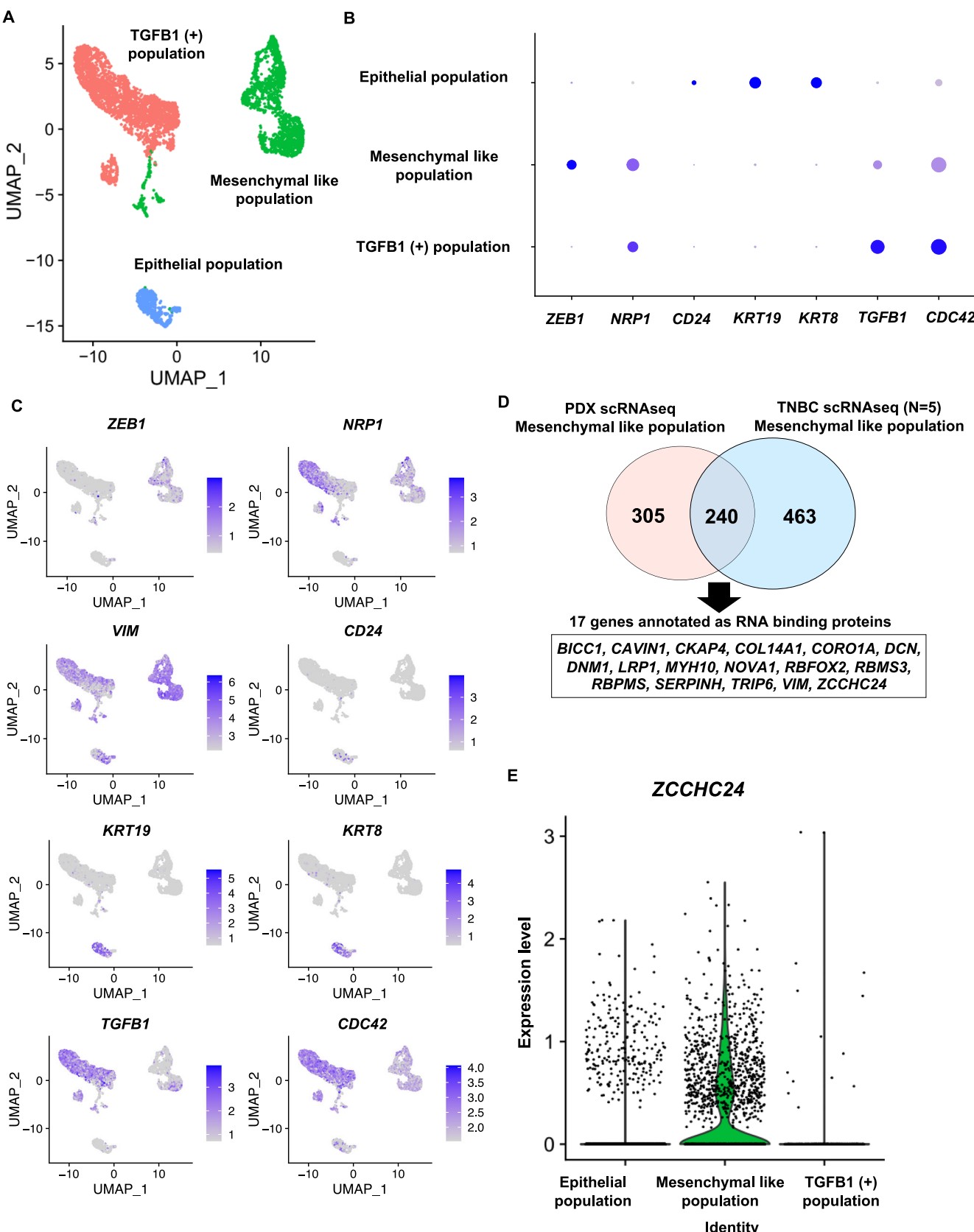

**Figure 1. RNA-binding protein ZCCHC24 shows specific expression in a mesenchymal-like population.**

(A) Dimensional reduction plot of single-cell RNA sequencing (scRNA-seq) of a patient-derived xenograft (PDX) (Patient #1). (B) Dot plot for PDX scRNA-seq (Patient #1). (C) Feature plots of PDX scRNA-seq for genes characterizing each population (*ZEB1, NRP1, VIM, CD24, KRT19, KRT8, TGFB1, CDC42*). (D) Venn diagram of mesenchymal-like population marker genes from scRNA-seq analyses of PDX and tumor tissues from five patients with TNBC. Among the 240 common marker genes, 17 genes with GO Term "RNA binding" were extracted and shown in the text box. (E) Violin plot of ZCCHC24 for scRNA-seq of PDX. (*P* value as a marker gene: $1.79 \times 10^{-169}$) (Cell numbers: Epithelial population: 494, Mesenchymal-like population: 1801, TGFB1(+) population: 2717).

ACTB, used as negative control transcripts, were not enriched in ZCCHC24-expressing cells (Fig. EV2). Next, we performed reporter assays using the 3'UTRs of *CD44*, *NRP1*, and *ZEB1* to evaluate whether these binding sites function as *cis*-elements to regulate target genes upon ZCCHC24 binding. The results showed that ZCCHC24 expression led to the upregulation of reporter activity (Fig. 3G; Appendix Fig. S7A). We also performed a reporter assay using 3'UTR of *ZEB1* with the knockdown of endogenous ZCCHC24 of MDAMB231 and found that the suppression of ZCCHC24 expression decreased reporter activity levels (Fig. 3H). Moreover, introducing mutations in the ZCCHC24-binding motifs suppressed the reporter activity (Fig. 3I; Appendix Fig. S7B). This confirms the functionality of this motif in mRNA stability and suggests that it is directly mediated by ZCCHC24.

## ZEB1 reciprocally promotes ZCCHC24 expression

Next, we identified *cis* and *trans* upstream regulators of ZCCHC24 expression. To this end, we combined and analyzed the deposited data from Hi-C, ATAC-Seq, and ChIP-Seq for H3K4Me1 and H3K27Ac in MDAMB231 cells, which suggested a candidate enhancer site for ZCCHC24 (Beesley et al, 2020; Feldker et al, 2020; He et al, 2021; Zanconato et al, 2015). Transcription factors that bind to this candidate enhancer can activate the enhancer activity to regulate ZCCHC24 expression. Unexpectedly, ChIP-Seq reanalysis showed that the common EMT transcription factor ZEB1, which was found to be post-transcriptionally regulated by ZCCHC24 and is also known to induce CSC (Brabletz et al, 2011; Chaffer et al, 2013; Jiang et al, 2020), as well as YAP1 and JUN, was bound to the ZCCHC24 candidate enhancer site. This is consistent with previous reports that ZEB1 cooperates with YAP1 and AP-1 to activate enhancer activity (Feldker et al, 2020). In support of this, ATAC-seq of ZEB1-knockdown MDAMB231 cells revealed loss of the peak representing the candidate enhancer site (Fig. 4A). Furthermore, knockdown of ZEB1, YAP1, or JUN in MDAMB231 cells and PDX treatment downregulated *ZCCHC24* expression. In contrast, the overexpression of ZEB1 upregulated *ZCCHC24* expression, as evaluated by qPCR (Figs. 4B and EV3A; Appendix Fig. S8).

To detect the endogenous ZCCHC24 protein in MDAMB231 cells, we used the HiBiT tag, which is a small part of the nano-luciferase, on the C-terminus of ZCCHC24 in MDAMB231 cells (231-KI), which allows for easy detection of endogenous ZCCHC24 protein expression (Appendix Fig. S9A) (Uchida et al, 2021). We confirmed this validity by western blotting and the luminescence of the nano-luciferase (Appendix Fig. S9B,C). Western blot analyses showed that ZCCHC24 was downregulated at the protein level by the knockdown of ZEB1 in 231-KI and PDX cells (Fig. EV3B).

Next, reporter assays were performed by integrating the candidate enhancer sites into the upstream sequence of the phosphoglycerate kinase (PGK) promoter. The results showed that overexpression of ZEB1 with YAP and AP-1 upregulated the reporter activities, whereas knockdown of ZEB1 suppressed these activities (Fig. 4C,D). Moreover, with the knockdown of ZEB1, a decrease in the enrichment of H3K27Ac at the enhancer candidate site was confirmed by ChIP-qPCR (Fig. EV3C). Supporting these results, the RNA expression levels of ZEB1 and ZCCHC24 in the Cancer Genome Atlas Breast Invasive Carcinoma (TCGA-BRCA) dataset (1178 samples) were strongly correlated (Fig. 4E).

Consistent with these results, immunostaining analysis of ZEB1 and ZCCHC24 in multiple TNBC pathological samples showed a tendency toward ZEB1 and ZCCHC24 co-expression (Fig. 4G). Our results highlight the potential role of ZCCHC24 in tumorigenicity and treatment resistance through a positive feedback loop in which its expression is induced by stemness-associated transcription factors, specifically in the mesenchymal population. Furthermore, it appeared to bind directly to and promote the stability of mRNAs transcribed from genes important for tumorigenicity and treatment resistance (Fig. 4F).

## ZCCHC24 downregulation attenuates tumorigenicity and chemoresistance

These results support the idea that ZCCHC24 is a potential therapeutic target for TNBC. To clarify the role of ZCCHC24 in tumorigenicity, we performed sphere-forming assays with ZCCHC24 knockdown in MDAMB231 cells and observed a decrease in their sphere-forming ability (Fig. EV4A). We also performed extremely limited dilution analysis (ELDA) in vitro with MDAMB231, HCC38, and PDX cells and found that ZCCHC24 knockdown decreased their tumor-forming abilities (Fig. EV4B–D). We evaluated the role of ZCCHC24 in tumorigenicity in vivo by subcutaneously transplanting cancer cells diluted with siRNAs targeting ZCCHC24 (Fig. 5A; Appendix Fig. S10). The results showed that ZCCHC24 knockdown decreased the tumorigenicity of MDAMB231 and PDX in vivo (Figs. 5B and EV4E). Consistent with our hypothesis that ZCCHC24 is critical for maintaining CSC properties, scRNA-seq of the tumors showed a large decrease in the mesenchymal-like population with *ZCCHC24* expression, characterized by *NRP1*-positive, *ZEB1*-positive, and *CD24*-low expression, upon ZCCHC24 knockdown (Fig. 5C–E). Moreover, the genes downstream of ZCCHC24, such as *ZEB1*, *NRP1*, and *ADAMTS1*, were downregulated in this population, indicating the importance of ZCCHC24 in maintaining mesenchymal-like populations (Fig. 5F; Appendix Fig. S11).

CSCs have been reported to show resistance to chemotherapy and are the main cause of disease recurrence (Dean et al, 2005). To determine whether ZCCHC24 plays a role in this resistance mechanism, we reanalyzed clinical samples from two clinical studies in which patients were treated with neoadjuvant

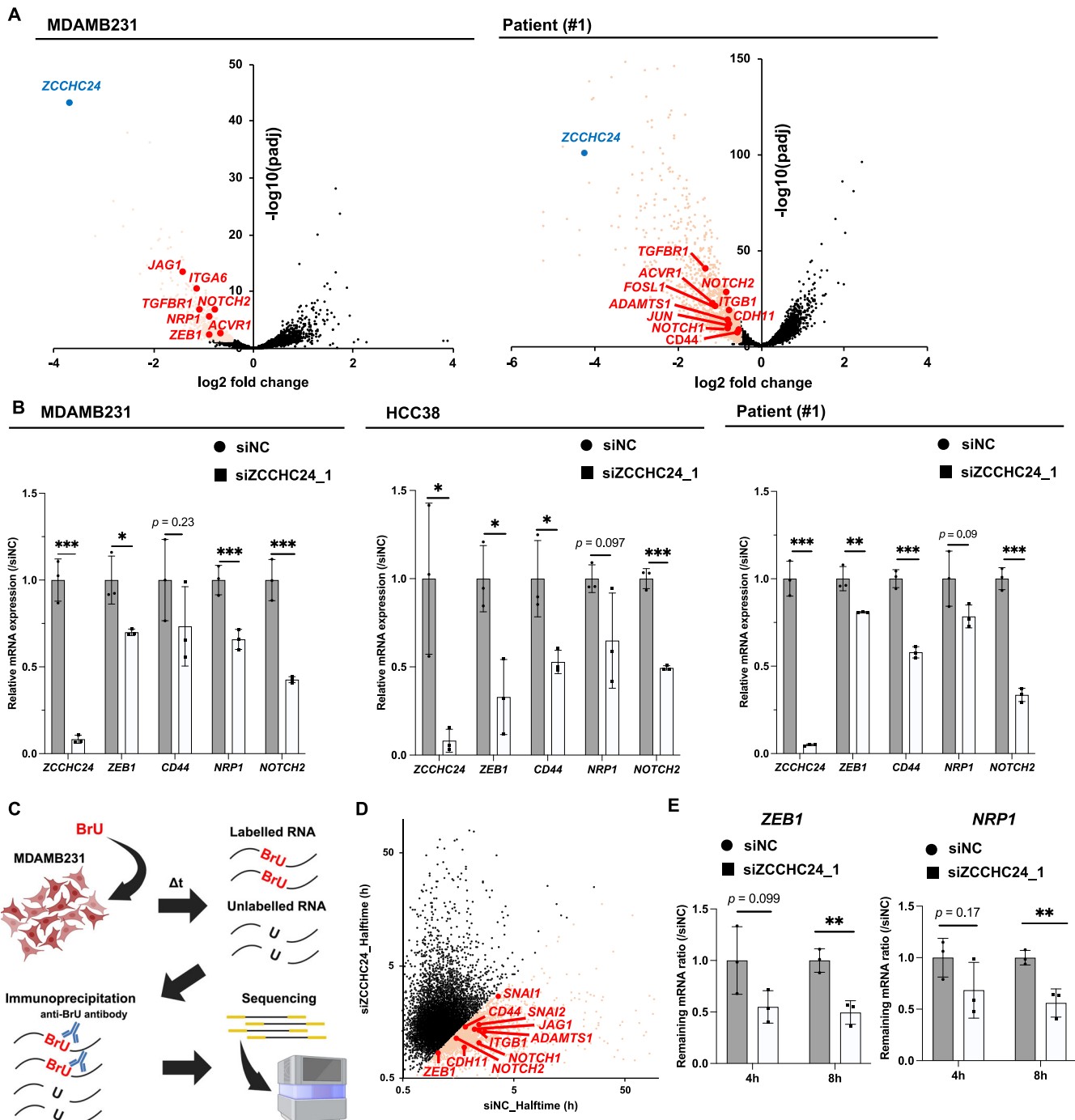

chemotherapy (Hatzis et al, 2011; Loibl et al, 2018). High ZCCHC24 expression correlated with high pathological residual disease rates (Fig. EV5A).

To investigate this further, we evaluated the effect of doxorubicin, a first-line chemotherapeutic agent for TNBC. Adding doxorubicin to MDAMB231 or PDX cells increased *ZCCHC24* expression (Fig. EV5B). Furthermore, the combined use of doxorubicin and siRNA against ZCCHC24 had additive effects on suppressing cell viability compared with a single treatment (Fig. EV5C,D).

Moreover, we explored BET inhibitors that elevate ZCCHC24 expression in TNBC cell lines (Lin et al, 2017; Shu et al, 2020). BET inhibitors target bromo- and extra-terminal domain (BBD) proteins (BRD2-BRD4) via acetyl-lysine-competitive binding to displace BBD proteins from chromatin, thereby selectively suppressing oncogenic transcription by inhibiting super-enhancers (Filippakopoulos et al, 2010). BET inhibitors are considered promising for overcoming treatment resistance in TNBC owing to their low toxicity to healthy cells and strong antitumor effects. However, associated clinical trials have been

◀ **Figure 2. ZCCHC24 regulates genes related to tumorigenicity and chemoresistance via the stabilization of mRNAs.**

(A) Distribution plots of RNA-seq analysis for the TNBC cell line MDAMB231 or patient-derived xenografts (patient #1) knocked down with ZCCHC24 siRNA. Differences in gene expression between siNC and siZCCHC24_1 were tested using DESeq2 (https://bioconductor.org/packages/release/bioc/html/DESeq2.html) following the manufacturer's protocol. The cutoff for the determination of differentially expressed genes (DEGs) was as follows: log2 fold change < −0.4 and $P$adj <0.05 for downregulated genes (colored in orange) and log2 fold change >0.4, $P$adj <0.05 for upregulated genes (gene lists are shown in Dataset EV3). (Three biological replicates per group). (B) qPCR analysis of the TNBC cell lines MDAMB231 and HCC38 or patient-derived xenografts (patient #1) knocked down with ZCCHC24 siRNA. Differences in gene expression between the siNC and siZCCHC24_1 cells were tested using an unpaired $t$ test for independent experiments. ($P$ values; MDAMB231; ZCCHC24: $2.1 \times 10^{-4}$, ZEB1: 0.020, CD44: 0.23, NRP1: 0.0045, NOTCH2: 0.0012, HCC38; ZCCHC24: 0.021, ZEB1: 0.015, CD44: 0.022, NRP1: 0.097, NOTCH2: $1.1 \times 10^{-4}$, PDX (Patient#1); ZCCHC24: $8.0 \times 10^{-5}$, ZEB1: 0.0087, CD44: 0.0027, NRP1: 0.094, NOTCH2: $9.8 \times 10^{-5}$) ($N = 3$ biological replicates each, *$P < 0.05$, **$P < 0.01$, ***$P < 0.005$). (C) Schema of BRIC-Seq. MDAMB231 cells were knocked down with siRNA against ZCCHC24 or the negative control, labeled with bromouridine (BrU) overnight. The medium was then discarded and replaced with fresh medium. At each time point, RNA was collected from the cells and labeled RNA was isolated by immunoprecipitation for BrU. Libraries were prepared and sequenced from the immunoprecipitated RNAs. (D) BRIC-seq results. Transcripts with less than 0.8-fold half-life are indicated in orange. (E) RNA levels of the indicated genes upon actinomycin D treatment of the TNBC cell line MDA-MB-231 knocked down with siRNA against ZCCHC24, compared to the negative control (siRNA negative control (NC)). Differences in the proportions of the remaining RNA were assessed using unpaired $t$ tests. ($P$ values; ZEB1; 4 h: 0.099, 8 h: 0.0057; NRP1; 4 h: 0.17, 8 h: 0.0077) ($N = 3$ biological replicates each, **$P < 0.01$). Data information: Data are presented as the mean ± SD (B, E). Source data are available online for this figure.

disappointing (Piha-Paul et al, 2019; Postel-Vinay et al, 2019). We also observed that adding JQ1 to MDAMB231 or PDX cells increased ZCCHC24 expression (Fig. EV5E). Based on these data, we hypothesized that targeting ZCCHC24 would increase the efficacy of TNBC treatment with JQ1. To test this in vivo, we added siRNAs against ZCCHC24 together with JQ1 treatment to subcutaneously grafted MDAMB231 or PDX cells and found that the dual use of JQ1 and siRNA against ZCCHC24 had additive effects in suppressing tumor growth in both models compared with single treatments (Fig. 6A).

## Discussion

Here, we identified ZCCHC24 as a critical RBP that directly controls a specific set of target mRNAs that govern tumorigenicity and treatment resistance and is expressed specifically in the TNBC mesenchymal-like population. Several RNA-binding proteins, including LIN28, Musashi, and YTHDF2, have been reported to regulate cancer progression. The upregulation of LIN28 accelerates the progression of various cancers via reciprocal regulation of the tumor suppressor miRNA, let-7, which suppresses oncogenes such as HMGA2, MYCN, and KRAS (Jeong et al, 2009; Molenaar et al, 2012). Musashi is involved in leukemia and the progression of various cancers by directly regulating tumor-suppressive and tumor-progressive genes (Ito et al, 2010; Kharas et al, 2010; Kudinov et al, 2016). YTHDF2, which recognizes and regulates RNA m6A (N6-methyladenosine) modifications, is upregulated in TNBC cell lines and acts as a translational repressor of EMT-related genes (Einstein et al, 2021). However, in the context of TNBC, an RBP that is uniquely expressed and essential for maintaining CSC traits has yet to be identified.

ZCCHC24 possesses a unique function in TNBC, where it is strongly expressed and specifically stabilizes important mRNAs such as ZEB1, CD44, and NRP1 in the mesenchymal-like population by recognizing specific *cis*-elements. Simultaneously, it engages in auto-amplification of this gene expression network within the cell, thus maintaining a cancer stem-like status through positive feedback mediated by ZEB1.

Although the exact mechanism by which ZCCHC24 is involved in maintaining mRNA stability is yet to be fully elucidated, targeted motif analyses have provided some insights. ZCCHC24 controls the

stability of this set of mRNAs in TNBC CSCs by directly binding to the motif "UGUWHWWA." This motif sequence is similar to the "UGUAHAUA" motif that another RBP, Pumilio, binds. Pumilio binding to this motif recruits the CCR4-NOT complex, thereby suppressing the translation of target mRNAs (Enwerem et al, 2021; Hafner et al, 2010). In this context, we anticipate that ZCCHC24 will stabilize mRNAs that characterize tumor initiation and treatment resistance by counteracting some of the target mRNAs in Pumilio.

The fundamental strategy for reducing the number of CSCs that exhibit chemotherapy resistance in solid tumors often involves targeting their niche or surface antigens. Approaches that therapeutically target this niche have gained attention, such as treatment strategies targeting Wnt signaling in colorectal and pancreatic cancers and semaphorin signaling via MICAL3 in breast cancer (Barker et al, 2009; Schepers et al, 2012; Shimokawa et al, 2017; Steinhart et al, 2017; Tominaga et al, 2019). However, CSCs, especially those in advanced stages, may exhibit reduced dependency on the niche (Batlle and Clevers, 2017; Fujii et al, 2016; Fumagalli et al, 2017). In addition, gene expression within the niche differs among primary sites, metastatic sites, and metastatic organs (Berthelet et al, 2021; Malanchi et al, 2011; Oskarsson et al, 2011).

Another therapeutic strategy for CSCs involves targeting cell-surface antigens specific to these CSCs. For instance, targeting Lgr5 in colorectal cancer, DLL3 in pulmonary neuroendocrine tumors (Morgensztern et al, 2019; Saunders et al, 2015), and CXCR1 in breast cancer has been reported to reduce the CSC population (Ginestier et al, 2010). These strategies can neutralize ligands before they bind to their receptors, indirectly targeting all downstream signaling cascades dependent on specific receptors.

In addition, intracellular gene expression networks, such as the intracellular ubiquitin ligase subunit Fbw7 (Takeishi et al, 2013), and histone deacetylase enzymes (HDAC) (Pattabiraman and Weinberg, 2014) serve as therapeutic targets for CSCs by regulating their differentiation or cell cycle. Furthermore, RBPs, such as Musashi mentioned above (Minuesa et al, 2019), are expected to be therapeutic targets because they control the expression of a set of critical oncogenic genes at the RNA level. However, reports on therapeutic strategies targeting RBPs in solid tumors, especially TNBC CSCs, are scarce.

The advantages of targeting ZCCHC24 in mesenchymal-like cells include not only direct inhibition of the expression of a set of

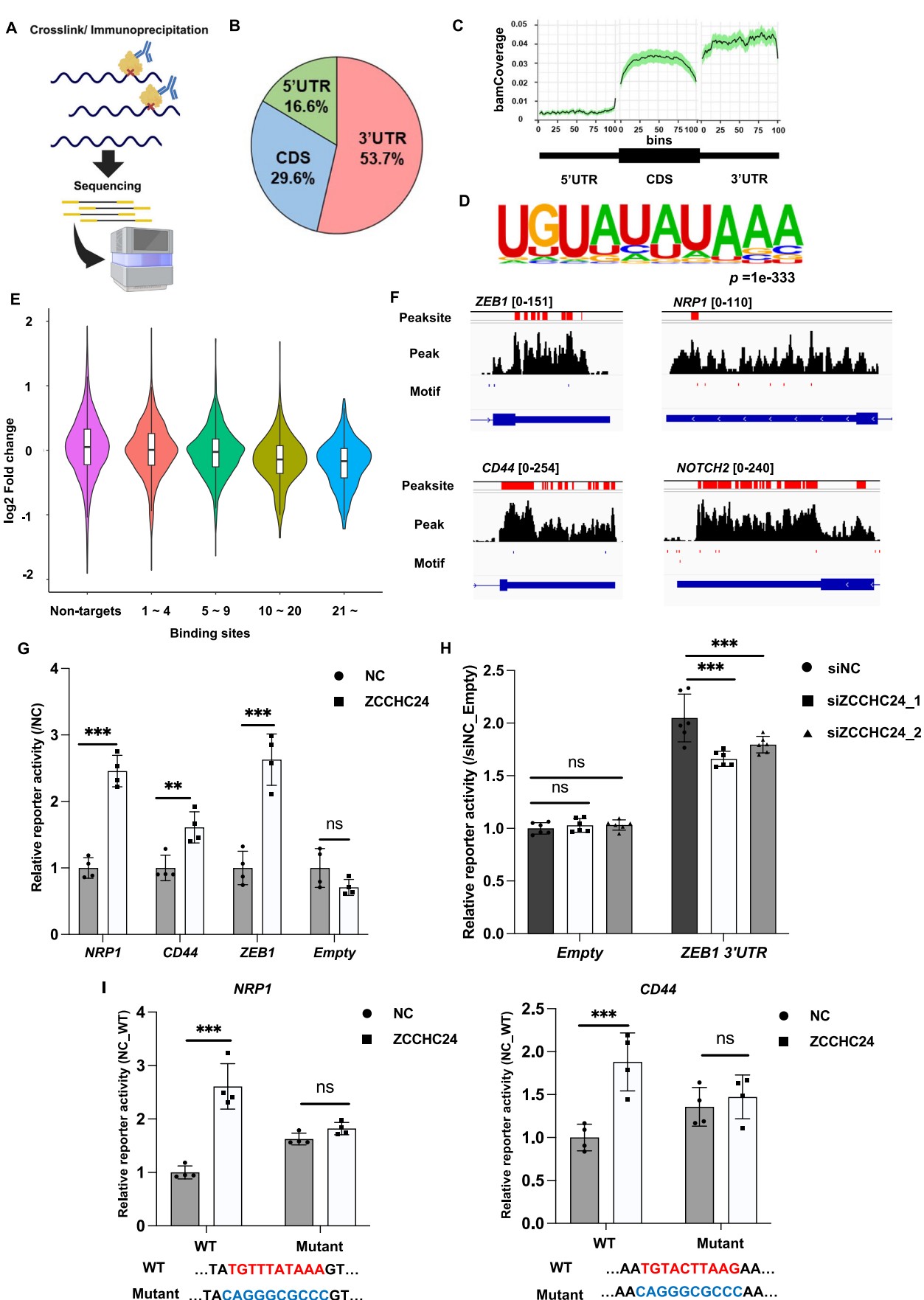

**Figure 3. ZCCHC24 stabilizes the mRNA of genes essential for breast cancer stemness by binding to *cis*-elements on 3'UTR.**

(A) Schema of PAR-CLIP. MDAMB231 cells expressing ZCCHC24 were incubated with 4-thiouridine (s4U) and cross-linked under ultraviolet. RNA-protein complexes were immunoprecipitated and fragmented, and SDS-PAGE purified RNA. Libraries were created and sequenced using these RNAs. (B) Pie chart showing the proportions of ZCCHC24-binding sites identified using PAR-CLIP. (C) Coverage plot showing the distribution of peak sites among 5'UTR, CDS, and 3'UTR. (D) Results of motif analysis using HOMER for the peak sites of PAR-CLIP ($P$ value $= 1 \times 10^{-133}$). (E) Violin plot showing the connection between the number of binding sites of PAR-CLIP in 3'UTR of target genes and the change in expression levels measured by RNA-Seq upon the knockdown of ZCCHC24 compared to the negative control. Box plots are shown with whiskers, medians, and lower and upper 25th percentiles of RNA-seq expression changes for each group. (Number of genes: non-targets: 3045; binding sites 1–4: 2515; binding sites 5–9: 1020; binding sites 10–20: 685; and binding sites >21: 296). (F) BigWig files, peak sites, and motif sites of PAR-CLIP for *NOTCH2*, *NRP1*, *CD44*, and *ZEB1*. (G) Luciferase reporter assay for reporter containing 3'UTR of *NRP1*, *CD44*, and *ZEB1* upon expressing ZCCHC24 or empty vector as a control. Differences in reporter activity were tested using the unpaired $t$ test as an independent test for each reporter. ($P$ values; *NRP1*: $4.9 \times 10^{-5}$, *CD44*: 0.0069, *ZEB1*: $4.0 \times 10^{-4}$, Empty: 0.11) ($N = 4$ biological replicates each, **$P < 0.01$, ***$P < 0.005$, ns: not significant). (H) Luciferase reporter assays using reporters with *ZEB1* 3'UTR and siRNA against endogenous ZCCHC24. Differences in reporter activity were tested using ANOVA and Tukey's post hoc test. ($P$ values; Empty; siZCCHC24_1: 0.998, siZCCHC24_2: 0.995; *ZEB1* 3'UTR; siZCCHC24_1; $1.3 \times 10^{-5}$, siZCCHC24_2; $4.5 \times 10^{-3}$) ($N = 6$, biological replicate each, ***$P < 0.005$, ns: not significant). (I) Luciferase reporter assays using reporters with *cis*-elements from *NRP1* (left) and *CD44* mutated to the ZCCHC24-binding site and evaluating the effect of ZCCHC24 expression or an empty vector as a control on reporter activity. Differences in reporter activity were tested using ANOVA and Tukey's post hoc test. ($P$ values; *NRP1* WT: $2.6 \times 10^{-6}$, *NRP1* Mutant: 0.652; *CD44* WT: 0.0016, *CD44* Mutant: 0.915) ($N = 4$ biological replicates each, ***$P < 0.005$, ns: not significant). Data information: Data are presented as mean ± SD (G–I). Source data are available online for this figure.

genes related to cancer stemness and EMT, including SNAI and NOTCH but also disruption of the auto-amplified gene expression network formed by ZCCHC24 and ZEB1 to maintain CSC traits. The therapeutic efficacy of targeting ZCCHC24 was demonstrated by a significant and specific reduction in the mesenchymal-like population when ZCCHC24-knockdown cells were transplanted into tumor tissue (Fig. 5C–E).

Although previous reports on RBPs (LIN28A, YTHDFs, and Musashi) have shown that a single inhibition through shRNA, siRNA, or small molecules can suppress cancer stemness, these strategies have not yet been adopted in combination with existing chemotherapies to enhance the effects of chemotherapy. In this regard, the upregulation of ZCCHC24 expression by doxorubicin, which is the first-line chemical compound for the treatment of TNBC, or by BET inhibitors, which are considered promising for TNBC treatment, may contribute to the development of treatment resistance when used as a monotherapy (Piha-Paul et al, 2019; Postel-Vinay et al, 2019; Shu et al, 2016; Shu et al, 2020). Our study demonstrates that combining therapies targeting ZCCHC24 in the mesenchymal state with standard treatments, such as conventional cytotoxic therapy, leads to enhanced antitumor effects. ZCCHC24 knockdown specifically reduced the mesenchymal-like population with ZCCHC24 expression, allowing targeting of proliferative cell populations with molecular targeted therapy using ZCCHC24-targeting siRNA in combination, thereby enhancing the effects of molecular targeted therapy in vivo (Fig. 6B). In this study, ZCCHC24-deficient tumors showed an increased apoptotic rate and decreased ZEB1 expression in dissected tumors. Analysis of the lineage of ZEB1-positive cells in PDX models may help reveal the underlying mechanisms.

One limitation of this study is that the half-lives of the mRNAs differed between the BRIC-Seq and actinomycin D assays. Owing to differences in the methods used to evaluate mRNA half-life and drug toxicity, it is highly possible that the measured half-lives differed between the two tests. In addition, identifying target transcripts of endogenous ZCCHC24 by immunoprecipitation of endogenous ZCCHC24 is important for elucidating the roles of ZCCHC24. However, we did not have suitable antibodies for the immunoprecipitation of endogenous ZCCHC24 and could not

perform the experiment. The identification of the endogenous targets of ZCCHC24 is important for future studies. In addition, we could not generate ZCCHC24 knockout (KO) cells because of their importance in cell survival. Analysis of ZCCHC24 KO cells would help to characterize the roles of ZCCHC24 in CSCs better. Furthermore, we identified the candidate enhancer site in ZCCHC24 using Hi-C and ChIP-Seq analyses, followed by reporter assays and ChIP-qPCR validation. However, other candidate enhancer sites may be important for the regulation of ZCCHC24 expression, which should be an essential future question for the validation of the ZEB1-ZCCHC24 axis.

For pathological analyses, TMA was performed to evaluate the co-expression of ZCCHC24 and ZEB1 objectively. However, because of the small number of ZEB1-positive cells, we could not perform TMA analysis, which remains an important topic for future research. In addition, we did not analyze the TCGA-BRCA samples separately for each subtype, and whether the strong correlation is specific to TNBC or general to all breast cancer subtypes remains unknown, as well as the critical issues to be analyzed.

Our findings suggest a niche-independent phenotypic maintenance in the CSC population, indicative of a positive loop between the nucleus and cytoplasm. This offers fundamental insights into cancer treatment and presents a novel gene expression network that regulates cellular status beyond genetics and epigenetics. Understanding this reciprocal gene expression control mechanism may provide new perspectives on pathology and pave the way for developing novel therapeutic options.

## Methods

### Animal experiments

All animal experiments were conducted according to the Guidelines for Proper Conduct of Animal Experiments (Science Council of Japan) and approved by the Center for Experimental Animals of Tokyo Medical and Dental University (Approval No. A2022-075C3). All experimental animals were bred under conventional conditions according to the guidelines of each institution.

### Reagents and tools table

| Reagent/resource | Reference or source | Identifier or catalog number |
| --- | --- | --- |
| **Experimental models** | | |
| Balb/c nu (*M. Musculus*) | CLEA Japan | CAnN.Cg-*Foxn1^{nu}*/CrlCrlj |
| NOG (*M. Musculus*) | CLEA Japan | NOD.Cg-*Prkdc^{skid}Il2rg^{tm1Sug}*/ShiJic |
| **Recombinant DNA** | | |
| pcDNA3.1 (+) | Thermofisher Scientific | V79020 |
| pCSII-CMV-IRES-Venus | RIKEN | RDB04383 |
| Tet-responsive promoter | Clontech | N/A |
| pLuc2-KAP-MCS | Ito et al, 2017 | N/A |
| pGL4.10 | Promega | N/A |
| **Antibodies** | | |
| Mouse monoclonal anti-ACTB | Sigma Aldrich | Cat#A2228; RRID: AB_476743 |
| Mouse monoclonal anti-BrdU | MBL | Cat#MI-11-3; RRID: AB_590678 |
| Mouse monoclonal anti-FLAG | Sigma Aldrich | Cat#F3165; RRID: AB_2637089 |
| Mouse monoclonal anti-FLAG | MBL | Cat#M185-3L; RRID: AB_11123930 |
| Mouse monoclonal anti-HiBiT | Promega | Cat#N7200; RRID: AB_2924793 |
| Rabbit monoclonal anti-NOTCH2 | Cell Signaling Technology | Cat#5732; RRID: AB_10694634 |
| Rabbit monoclonal anti-NRP1 | Cell Signaling Technology | Cat#3725; RRID: AB_2155231 |
| Rabbit monoclonal anti-ZEB1 | Cell Signaling Technology | Cat#3396; RRID: AB1904164 |
| Rabbit monoclonal anti-H3K27Ac | Cell Signaling Technology | Cat#8173; RRID: AB10949503 |
| Rabbit polyclonal anti-ZCCHC24 | abcam | Cat#ab243699 |
| Rabbit polyclonal anti-ZCCHC24 | LS-bio | Cat#LS-C160747-400 |
| Mouse monoclonal anti-ZEB1 | R and D systems | Cat#MAB6708; RRID: AB_10972647 |
| Alexa 488 goat anti-rabbit | Thermo Fisher Scientific | Cat#A32731; RRID: AB_2633280 |
| Alexa 594 donkey anti-mouse | Thermo Fisher Scientific | Cat#A21203; RRID: AB_141633 |
| Phycoerythrin-conjugated anti-CD44 | Biolegend | Cat#50-0441; RRID: AB_2621762 |
| Allophycocyanin-conjugated anti-NRP1 | R and D Systems | Cat#FAB3870A; RRID: AB_1241850 |
| Anti-rabbit IgG | WAKO | Cat#111-001-008 |
| Anti-mouse IgG | WAKO | Cat#115-001-003 |
| **Oligonucleotides and other sequence-based reagents** | | |
| qPCR primer: *PSMB2* forward TTCCCTCTG AGGTGCTGTCT | This paper | N/A |
| qPCR primer: *PSMB2* reverse ACATCTTGTCAT GATCGTCCTTCA | This paper | N/A |
| qPCR primer: *ZCCHC24* forward CTAGCGCCTTC GATGCCTT | This paper | N/A |
| qPCR primer: *ZCCHC24* reverse AGGTCTGAGAA GTGCTCGGT | This paper | N/A |
| qPCR primer: *NRP1* forward CGGGACCCA TTCAGGATCAC | This paper | N/A |
| qPCR primer: *NRP1* reverse GCTGATCGTAC TCCTCTGGC | This paper | N/A |
| qPCR primer: *CD44* forward AACTGGAACCC AGAAGCACA | This paper | N/A |
| qPCR primer: *CD44* reverse CAGCTGTCCCT GTTGTCGAA | This paper | N/A |
| qPCR primer: *ZEB1* forward GATGACCTGCC AACAGACCA | This paper | N/A |

| Reagent/resource | Reference or source | Identifier or catalog number |
|---|---|---|
| qPCR primer: *ZEB1* reverse GTCCTCTTCA GGTGCCTCAG | This paper | N/A |
| qPCR primer: *NOTCH2* forward CTACAGTTGTC GCTGCTTGC | This paper | N/A |
| qPCR primer: *NOTCH2* reverse AGGCATGTTAC TGGCCACAG | This paper | N/A |
| qPCR primer: *YAP1* forward CCCTCGTTTT GCCATGAACC | This paper | N/A |
| qPCR primer: *YAP1* reverse CATCCTGCTCC AGTGTTGGT | This paper | N/A |
| qPCR primer: *JUN* forward TATGACGATG CCCTCAACGC | This paper | N/A |
| qPCR primer: *JUN* reverse CCCGTTGCT GGACTGGATTA | This paper | N/A |
| qPCR primer: *B2M* forward ACTCTCTCTTT CTGGCCTGG | This paper | N/A |
| qPCR primer: *B2M* reverse CGTGAGTAACCT GAATCTTTGG | This paper | N/A |
| qPCR primer: *ACTB* forward CTGACTACCTCA TGAAGATCCTC | This paper | N/A |
| qPCR primer: *ACTB* reverse CATTGCACCTGG TGATGACCTG | This paper | N/A |
| qPCR primer: ChIP enhancer region forward TTTCAGGTCCC TCCAAGGGG | This paper | N/A |
| qPCR primer: ChIP enhancer region reverse TTTCAGGTCCC TCCAAGGGG | This paper | N/A |
| qPCR primer: ChIP *GAPDH* promoter region forward TACTAGCGG TTTTACGGGCG | This paper | N/A |
| qPCR primer: ChIP *GAPDH* promoter region reverse TCGAACAGGAGGA GCAGAGAGCGA | This paper | N/A |
| AllStars negative control siRNA | QIAGEN | Cat#1027281 |
| siRNA for JUN (siJUN_1) | IDT | Cat#hs.Ri.JUN.13.1 |
| siRNA for JUN (siJUN_2) | IDT | Cat#hs.Ri.JUN.13.2 |
| siRNA for YAP1 (siYAP1_1) | IDT | Cat#hs.Ri.YAP1.13.2 |
| siRNA for YAP1 (siYAP1_2) | Thermo Fisher Scientific | Cat#hs.HSS155944 |
| siRNA for ZCCHC24 (siZCCHC24_1) | Thermo Fisher Scientific | Cat#HSS137253 |
| siRNA for ZCCHC24 (siZCCHC24_2) | Thermo Fisher Scientific | Cat#HSS176719 |
| siRNA for ZEB1 (siZEB1_1) | Thermo Fisher Scientific | Cat#HSS110549 |
| siRNA for ZEB1 (siZEB1_2) | Thermo Fisher Scientific | Cat#HSS110548 |
| stealth RNAi siRNA negative control Hi | Thermo Fisher Scientific | Cat#12935-400 |
| RNA Sequence for 3′RNA linker of PAR-CLIP: /5Phos/ AUAUAGGNNNNNAGAUCGGAAGAG CGUCGUGUAG/3SpC3, /5Phos/AAUAGCA NNNNNAGAUCGGAAGAGCGUC GUGUAG/3SpC3 | This paper | N/A |
| Reverse transcribe RNA for PAR-CLIP: ACACGACGC TCTTCCGA | This paper | N/A |
| crRNA for tagging: ACTGCC GTCGCGTGC AGTGA | This paper | N/A |

| Reagent/resource | Reference or source | Identifier or catalog number |
|---|---|---|
| Oligo DNA for tagging AGAGCAAGGAGC ACCCGCAGCACCTCTGCG AGAAGTGCAAGGTCCTGGGCTACTACTG CCGTCGCGTGCAGGACTACAAGGACGA CGACGACAAGGTGAGCGGCTGGCGGCT GTTCAAGAAGATTAGCTGACGGGCTGC CCGCCCGCACCCAGAGCCACCCCCCG CCAGCCCGAGGAGACGCTGCTTCCCT GTGCTACTC | This paper | N/A |
| **Chemicals, enzymes, and other reagents** | | |
| B27 | WAKO | Cat#400-160 |
| Doxorubicin | Selleck | Cat#E2516 |
| EGF | R and D Systems | Cat#236-EG |
| FGF | WAKO | Cat#062-06041 |
| (+)-JQ1 | Selleck | Cat#S7110 |
| (+)-JQ1 | Med Chem Express | Cat#HY-13030 |
| 10X Chromium Single Cell Capture Chip | 10X Genomics | Cat#1000127 |
| Acid phenol/chloroform/isoamyl alcohol | Nippon Gene | Cat#311-90151 |
| Actinomycin D | Thermofisher Scientific | Cat#A7592 |
| Alkaline phosphatase | Thermo Fisher Scientific | Cat#EF0651 |
| Ampure XP beads | Beckman Coulter | Cat#A63881 |
| ALT-R Cas9 Nuclease V3 | IDT | Cat#1081059 |
| Collagenase | WAKO | Cat#032-22364 |
| DNaseI | Thermo Fisher Scientific | Cat#18047019 |
| Dulbecco's modified Eagle's medium | Corning | Cat#10-017-CV |
| Dynabeads Protein A | Thermo Fisher Scientific | Cat#10001D |
| Dynabeads Protein G | Thermo Fisher Scientific | Cat#10009D |
| Epicult-C Human Medium | STEM-CELL | Cat#ST-05630 |
| ExoSAP-IT | Thermo Fisher Scientific | Cat#78201 |
| Hoechst | Dojindo | Cat#346-07951 |
| Lipofectamine RNA iMax | Thermofisher Scientific | Cat#13778150 |
| Matrigel | Corning | Cat#354234 |
| Nano Glo HiBiT Lytic Detection System | Promega | Cat#N3050 |
| NEBNext rRNA depletion kit v2 | NEB | Cat#E7400 |
| NEBNext Ultra II RNA library prep kit for Illumina | NEB | Cat#E7770 |
| NuPAGE sample buffer | Invitrogen | Cat#NP0007 |
| NucleoSpin Gel and PCR-clean up kit | Takara | Cat#740609 |
| PEG300 | Selleck | Cat#S6704 |
| Prime Script | Takara | Cat#2680A |
| Proteinase K | NEB | Cat#P8107 |
| Q5 PCR enzymes | NEB | Cat#M0491S |
| RBC lysis buffer | Invitrogen | Cat#00-4333-57 |
| RealTime-Glo MT Cell Viability Assay kit | Promega | Cat#G9711 |
| Relia Prep RNA Miniprep system | Promega | Cat#Z6010 |
| RNA ligase high-concentration | NEB | Cat#M0437S |
| RNase I | Thermo Fisher Scientific | Cat#EN0601 |
| T4 polynucleotide kinase | NEB | Cat#M0201S |
| TGIRT-III enzyme | InGex | Cat#TGIRT50 |

| Reagent/resource | Reference or source | Identifier or catalog number |
| --- | --- | --- |
| Tyramide SuperBoost kit, goat anti-mouse IgG | Thermo Fisher Scientific | Cat#15611892 |
| Tyramide SuperBoost kit, goat anti-rabbit IgG | Thermo Fisher Scientific | Cat#15631902 |
| Tween 80 | Selleck | Cat#S6702 |
| **Software** | | |
| Prism | GraphPad Software | https://www.graphpad.com |
| STAR | Dobin et al, 2013 | https://github.com/alexdobin/STAR |
| iDEP.96 | | http://bioinformatics.sdstate.edu/idep/ |
| Cell Ranger | 10X Genomics | https://www.10xgenomics.com/support/software/cell-ranger/downloads |
| MANE select | National Institute of Health, USA | https://www.ncbi.nlm.nih.gov/refseq/MANE/ |
| GGGenome | https://github.com/meso-cacase/GGGenome | https://gggenome.dbcls.jp/en/help.html |
| Cuffnorm | Trapnell et al, 2010 | https://cole-trapnell-lab.github.io/cufflinks/cuffnorm/ |
| Imaris Viewer | Oxford Instruments | N/A |
| R v.4.1.2 | R Foundation for Statistical Computing | https://www.r-project.org/ |
| Seurat v. 4.0.6 | Butler et al, 2018 | https://satijalab.org/seurat/ |
| Trim-Galore | The Babraham Institute | https://github.com/FelixKrueger/TrimGalore |
| **Other** | | |
| Analyzed data | This paper | DRA016408 |
| Confocal microscope BC43 | Andor | Cat#BC43 |
| Patient-derived xenografts (PDX) | Division of Cancer Cell Biology, Kanazawa University (PI: Noriko Gotoh) | https://bunshibyotai.w3.kanazawa-u.ac.jp/ |
| Human triple-negative breast cancer specimens | Department of Comprehensive Pathology, Tokyo Medical and Dental University | https://www.tmd.ac.jp/med/pth2/index.html |
| Reanalyzed ChIP-Seq data | Beesley et al, 2020; Feldker et al, 2020; He et al, 2021; Zanconato et al, 2015 | EBI datasets: E-MTAB-8258 and E-MTAB-8264, GEO datasets: GSE166941 and GSE66081 |
| Reanalyzed Hi-C data | Beesley et al, 2020 | PRJEB29716 |
| Reanalyzed RNA expression datasets of clinical trials | Hatzis et al, 2011; Loibl et al, 2018 | GEO datasets: GSE164458, GSE25055 |
| RNA-Seq expression profiles for breast cancer samples | National Cancer Institute Center for Cancer Genomics | TCGA-BRCA datasets |

## Plasmid construction

For the overexpression vector, the sequence of the open reading frame of ZCCHC24 or ZEB1 was inserted with a 3XFLAG peptide at the N-terminus between the NheI and NotI restriction sites into the pcDNA3.1 (+) vector (Thermo Fisher Scientific). The pCLT lentiviral vectors for cDNA expression using the tet-on system were created by modifying the pCSII-CMV-IRES-Venus vector (RDB04383; RIKEN). The CMV-IRES-Venus sequence was removed by inverse PCR, and the tet-responsive promoter (Clontech) and PGK promoter-puromycin-N-acetyltransferase-P2A-reverse tetracycline transactivator (rtTA) were cloned into the pCSII vector to create the pCLT vector. A lentiviral expression vector was constructed by inserting the GFP or ZCCHC24 cDNA sequences with 3xFLAG downstream of the tet-on promoter into the lentiviral vectors. For luciferase reporter vectors for post-transcriptional regulation, 3'UTR of CD44, ZEB1, or their partial sequences with mutation in binding sites were inserted downstream of the luciferase gene. The control vector (pLuc2-KAP-MCS) has been described in a previous study (Ito et al, 2017). For the luciferase reporter vectors used for transcriptional regulation, a control reporter vector was designed and generated by inserting the PGK promoter sequence into the multi-cloning site (MCS) of pGL4.10 (Promega). The estimated regulatory sequence was inserted into the MCS of the reporter vectors. The sequences of other reporter and lentiviral vectors expressing the tet-on promoter are shown in Appendix Table S1.

## Cell culture

MDAMB231 cells were cultured in Dulbecco's modified Eagle's medium (DMEM; Corning) supplemented with 10% fetal bovine

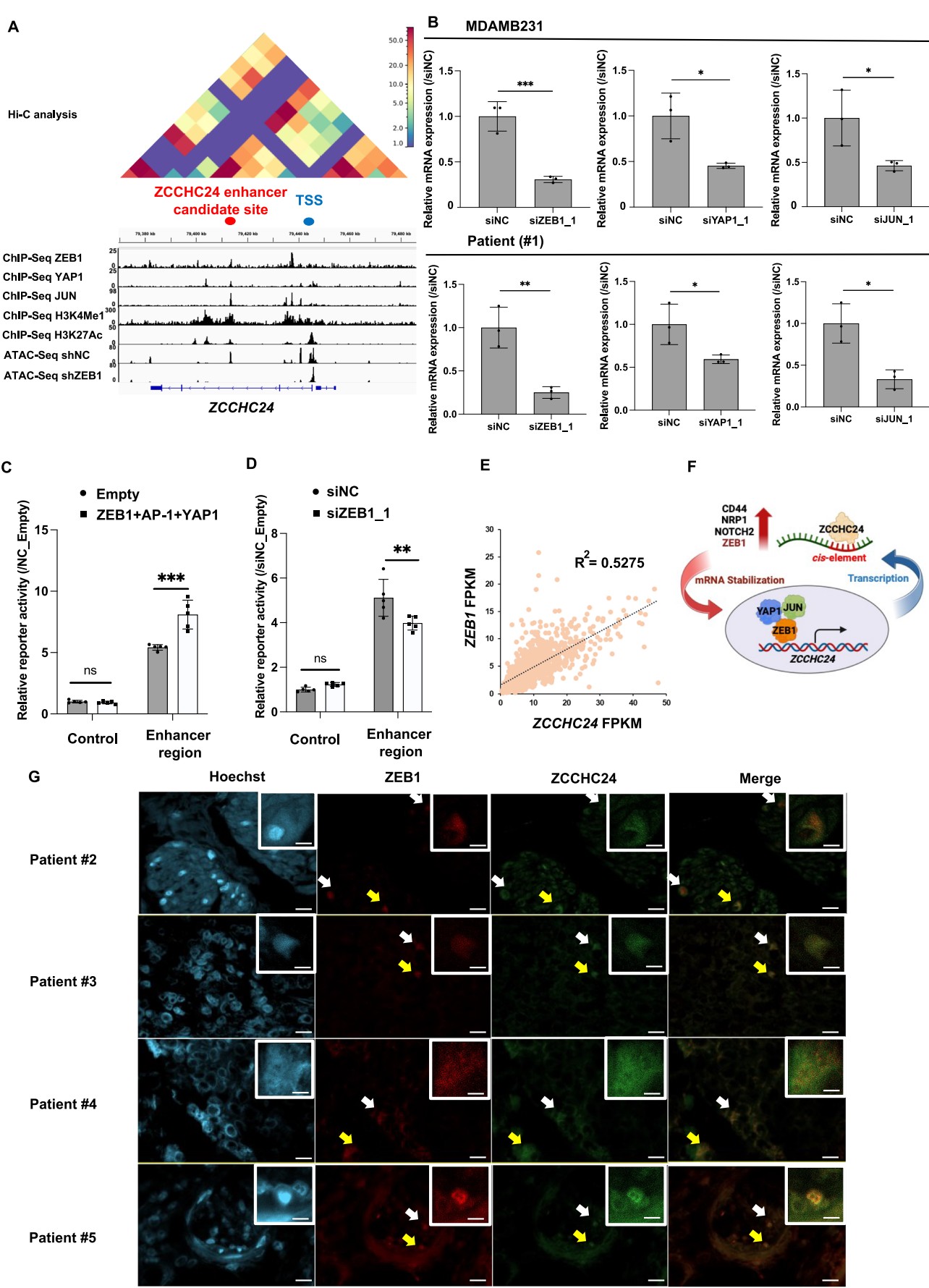

**Figure 4. The transcriptional regulator ZEB1 upregulates the expression of ZCCHC24.**

(A) Topologically associated domains (TADs) for reanalyzed Hi-C of MDAMB231 (Beesley et al, 2020), reanalyzed ChIP-Seq data for ZEB1, YAP1, JUN, H3K4Me1, and H3K27Ac (Zanconato et al, 2015; Feldker et al, 2020, and He et al, 2021), and reanalyzed TAC-Seq data for MDAMB231 cells knocked down with ZEB1 shRNA (Feldker et al, 2020). (B) qPCR analyses of MDAMB231 or PDX (Patient #1) cells knocked down using siRNA targeting ZEB1, JUN, and YAP1. Differences in gene expression between the siNC and siZCCHC24_1 cells were tested using an unpaired $t$ test for independent experiments. ($P$ values; MDAMB231; siZEB1_1: 0.0019, siYAP1_1: 0.020, siJUN_1: 0.044, PDX (Patient #1); siZEB1_1: 0.0061, siYAP1_1: 0.043, siJUN_1: 0.011) ($N = 3$ biological replicates each, *$P < 0.05$, **$P < 0.01$, ***$P < 0.005$). (C) Luciferase reporter assay for the reporter containing the PGK promoter (PGKp) and enhancer candidate region with the expression of AP-1, YAP1, and ZEB1. Differences in reporter activity were tested using ANOVA and Tukey's post hoc test. ($P$ values: Control: 0.998, Enhancer candidate site: $1.6 \times 10^{-5}$) ($N = 5$ biological replicates each, ***$P < 0.005$). (D) Luciferase reporter assay for a reporter containing the PGK promoter (PGKp) and an enhancer candidate region with ZEB1 knockdown. Differences in reporter activity were tested using ANOVA and Tukey's post hoc test. ($P$ values: Empty: 0.849, Enhancer candidate site: 0.0052) ($N = 5$ biological replicates, **$P < 0.01$). (E) Distribution plot of *ZEB1* and *ZCCHC24* mRNA expression from TCGA-BRCA database. Relevance was calculated using the FPKM of each gene as the correlation coefficient. (F) Schematic representation of the model. ZEB1 transcriptionally upregulates ZCCHC24, while ZCCHC24 regulates the expression of target genes, including ZEB1, by directly binding to *cis*-elements in target mRNAs. (G) Immunofluorescence analysis of TNBC pathological samples (patients #2–#5) for ZEB1 and ZCCHC24 expression. White and yellow arrows show cells co-expressing ZEB1 and ZCCHC24. For the cells marked with a yellow arrow, the images captured from a single cell are shown in the upper-right corner of the figure. Data information: Data are presented as mean ± SD (B–D). (G) Scale bar: 10 μm (overview) and 3 μm (captured single cell). Source data are available online for this figure.

serum (FBS) (Gibco) and 1% penicillin–streptomycin (PS) (Wako) at 37 °C and 5% $CO_2$. HCC38 cells were cultured in Roswell Park Medium Institute media (RPMI; Corning) supplemented with 10% FBS (Gibco) and 1% penicillin–streptomycin (Wako) at 37 °C and 5% $CO_2$. Breast cancer patient-derived cells were cultured in Epicult-C Medium (STEMCELL) with 200 mM L-glutamine (−) (Wako), 1% penicillin–streptomycin (Wako) and 0.5 mg/mL hydrocortisone on collagen-coated (Type I collagen) dishes (IWAKI) at 37 °C and 5% $CO_2$.

## Establishment of patient-derived xenografts (PDX)

Fresh breast cancer tissues were obtained from patients who underwent surgical resection or biopsy at Kanazawa University Hospital or University of Tokyo Hospital. The use of the samples was approved by the Institutional Review Boards of the Cancer Research Institute of Kanazawa University, Institute of Medical Science of the University of Tokyo, Minami-Machida Hospital, University of Tokyo Hospital, and Tokyo Medical and Dental University (G2020-040). Informed consent was obtained from all patients. All the experiments were performed in accordance with the Declaration of Helsinki. Female NOD. CG-PRKDC SCID IL2RG TM1WJL /SZJ (NSG) mice (Charles River, Wilmington, MA) were handled according to the guidelines of the Institute for Experimental Animals, Kanazawa University. Animal experiments were approved by the Committee for Animal Research of Kanazawa University. The PDX models were established as previously reported (Tominaga et al, 2019). Briefly, breast cancer tissues obtained from breast cancer patients were cut into 1-mm pieces, and five slices were mixed with Matrigel (Corning, 354234) to create 50 μL of the cell-Matrigel mixture. This five slice-per-site in Matrigel mixture was transplanted onto mammary fat pads of NSG mice. When tumor sizes reached 1000 mm³, the mice were sacrificed. Tumor tissues were used in the experiments.

## Patient information for PDX

The patient information for PDXs is as follows: age: 55, histological type: invasive ductal carcinoma, estrogen receptor: (−), progesterone receptor: (−), HER2: (−), Ki67 index: 50%, clinical subtype: TNBC, BRCA1,2 mutations: (−), Drug resistance: No.

## Single-cell isolation, library preparation, and sequencing for PDX

Tumor tissues derived from PDXs were subcutaneously transplanted into 6–7-week-old female *NOD.Cg-Prkdc^{SCID}IL2rg^{tm1Sug}/Shijc* mice (CIEA, Kawasaki, Japan) using Matrigel (Corning, 354234). Before transplantation, trypan blue staining confirmed high viability (>80%). Distilled water with 2% DMSO, 30% PEG300, and 5% Tween 80 was injected intraperitoneally once every 3 days. Three weeks after transplantation, fresh breast cancer tissues were extracted, and murine tissues were removed. Then, tissues were dissociated into single cells using collagenase (FUJIFILM Wako, 032-22364), and red blood cells were excluded using RBC lysis buffer (Invitrogen, 00-4333-57). Single live cells were loaded onto a 10X Chromium Single Cell Capture Chip, followed by single-cell capture reverse transcription and library preparation according to the manufacturer's protocols. The constructed library was sequenced using NovaSeq ($2 \times 150$ bp).

## Single-cell RNA-Seq analysis

Cell ranger (v. 6.1.1. 10x Genomics) was used to demultiplex the samples, process barcodes, and align the GRCh38 genome. Individual samples were integrated, expression was normalized, and cell populations were clustered based on the matrix files of gene expression using R v.4.1.2 (R Foundation for Statistical Computing, Vienna, Austria) and the R package Seurat v. 4.0.6 (Butler et al, 2018) as described by the developer. The datasets were analyzed using the following protocols: Genes expressed in <3 cells, cells with <2000 unique molecular identifiers (UMIs), and <200 genes were removed from the gene expression matrix for each dataset. Furthermore, the data were also filtered with expression of mitochondrial gene contamination using the criterion of <20%. The data were log-normalized, and the expression of each gene was scaled by regressing the number of UMIs. The gene expression matrix was analyzed using principal component analysis (PCA), and we utilized unsupervised shared nearest neighbor (SNN) clustering of the genes with a resolution of 0.025. They were then visualized using Uniform Manifold Approximation and Projection (UMAP). Uniquely expressed genes in each cluster were analyzed using the Seurat FindConservedMarker function. We set the

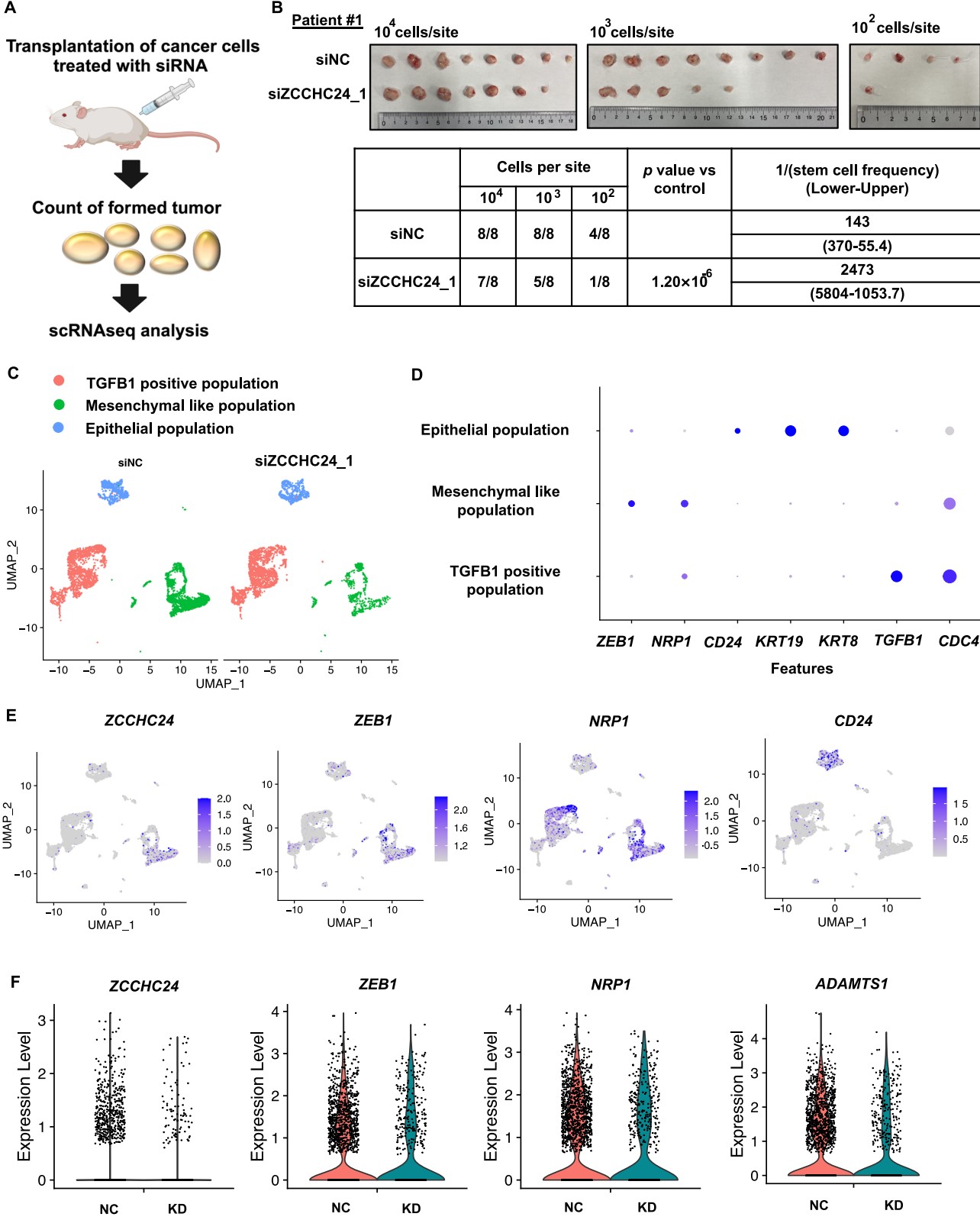

**Figure 5. ZCCHC24 maintains the mesenchymal-like population and contributes to tumor formation.**

(A) Schema of the experiments. PDX transfected with ZCCHC24 siRNA were subcutaneously transplanted, and the number of tumors formed was counted. The resulting tumors were subjected to single-cell RNA sequencing. (B) Comparison of tumor formation ability in vivo for patient-derived xenografts (Patient #1) knocked down with siRNA against ZCCHC24 or the negative control. The cells were transplanted into 7-week-old female NOG mice. Tumor formation ability was tested using the likelihood ratio test of the single-hit model, as shown on the ELDA software website (https://bioinf.wehi.edu.au/software/elda/). ($P$ value = $1.20 \times 10^{-6}$). (C) Dimensional reduction plot for scRNA-seq of transplanted PDX knocked down with siRNA against ZCCHC24 or negative control. (D) Dot plots of representative genes for scRNA-seq of transplanted PDX. (E) Feature plots of scRNA-seq on PDX for genes characterizing mesenchymal-like populations (*ZCCHC24, ZEB1, NRP1, CD24*). (F) Violin plots of *ZCCHC24* and its downstream genes (*ZEB1, NRP1,* and *ADAMTS1*) for the mesenchymal-like population of scRNA-seq transplanted PDX cells (number of cells in the cluster; siNC: 2007, si-ZCCHC24:664). Source data are available online for this figure.

threshold at $P < 1.00$ E-5 as the criterion for the marker genes. Violin plots were visualized using the VlnPlot function, and feature plots were depicted using the FeaturePlot function.

## Single-cell RNA seq reanalysis for deposited data of five triple-negative breast cancer patients

Fastq data provided in a previous report (Wu et al, 2020) were used for the analyses. Cell ranger (v. 6.1.1. 10x Genomics) was used to demultiplex the samples, process barcodes, and align the GRCh38 genome. Individual samples were integrated, expression was normalized, and cell populations were clustered based on the matrix files of gene expression (R v.4.1.2; R Foundation for Statistical Computing, Vienna, Austria) and the R package Seurat v. 4.0.6, as described by the developer. The following protocol for analysis was the same as that for the single-cell analysis of PDX.

## Knockdown with siRNA

To downregulate gene expression, the cells were transfected with siRNA against *ZCCHC24* (si-ZCCHC24_1; stealth RNAi HSS137253, Thermo Fisher Scientific, siZCCHC24_2; stealth RNAi HSS176719 Thermo Fisher Scientific), stealth RNAi siRNA negative control Hi (Thermo Fisher Scientific), siRNA against ZEB1 (ZEB1_1; stealth RNA I HSS110549 Thermo Fisher Scientific, ZEB1_2; stealth RNA i HSS110548 Thermo Fisher Scientific), JUN (JUN_1; IDT; Design ID hs.Ri.JUN.13.2, JUN_2; Design ID hs.Ri.JUN.13.1), YAP1 (YAP1_1; IDT; Design ID hs.Ri.YAP1.13.2, YAP1_2; HSS115944 Thermo Fisher Scientific), AllStars negative control siRNA (5′-AAGGCAAGTGTTGGAGAATAA-3′; QIA-GEN) using Lipofectamine RNA iMax (Thermo Fisher Scientific), following the manufacturer's instructions.

## RNA extraction

RNA was extracted using the Relia Prep RNA Miniprep System (Promega). The extracted RNA was subjected to a quantitative real-time polymerase chain reaction (qRT-PCR) or RNA sequencing (RNA-Seq).

## Establishment of GFP or ZCCHC24-expressing cells

For producing lentivirus, 293FT cells cultured with 10% FBS and 1% PS containing DMEM on 10-cm dishes are transfected with 4 μg of lentiviral vectors and 2 μg pCMV-VSVg, 2 μg pHIV-gp. After 48 h incubation, the viral media were collected, precipitated with polyethylene glycol, and centrifuged at 8000×*g* for 30 min. The

pellet was collected and resuspended with 200 μl of Opti-Mem. MDAMB231 cells were infected with the virus-containing media, and cells were selected with 1 μg/ml of puromycin containing DMEM with 10% FBS and 1% PS.

## qRT-PCR

The extracted RNA was reverse-transcribed into complementary DNA (cDNA) using Prime Script (Takara, Kusatsu, Japan). cDNA was subjected to qRT-PCR using the primers listed in the reagents and tools table.

## RNA-Seq

RNA (500 ng) from MDAMB231 cells or breast cancer patient-derived cells knocked down with ZCCHC24 siRNA was subjected to RNA-Seq analysis. Ribosomal RNA was extracted using the NEBNext rRNA Depletion Kit v2 (NEB, MA, USA). An RNA-Seq library was prepared using the NEBNext Ultra II RNA Library Prep Kit (Illumina). RNA-Seq was performed using a Next-Seq 500 (Illumina, CA, USA).

## RNA-Seq data analysis

Adapters in the RNA-seq data were removed using Trim-Galore (https://www.bioinformatics.babraham.ac.uk/projects/trim_galore/). The trimmed RNA-seq data were mapped to the GRCh38 genome using STAR (https://github.com/alexdobin/STAR). The mapped data were quantified by RSEM. Differentially expressed genes (DEGs) were identified using iDEP 96 (http://bioinformatics.sdstate.edu/idep/).

## Flow cytometric analysis

The assembled cells were stained with phycoerythrin-conjugated anti-CD44 (IM7, BioLegend, CA, USA) and allophycocyanin-conjugated anti-NRP1 (FAB3870A, R&D Systems, MN, USA) antibodies in fluorescence-activated cell sorting buffer (0.5% FBS in PBS) on ice for 30 min. The stained cells were analyzed using a FACS Calibur flow cytometer (BD Biosciences, NJ, USA).

## BRIC-Seq library preparation

MDAMB231 cells were seeded in 10 cm dishes and incubated for 24 h. After incubation, the cells were transfected with siRNA against ZCCHC24 or the negative control and incubated for 48 h. Then, cells were stimulated with 150 μM bromouridine in 10% FBS/ 1% penicillin–streptomycin-containing DMEM and incubated for

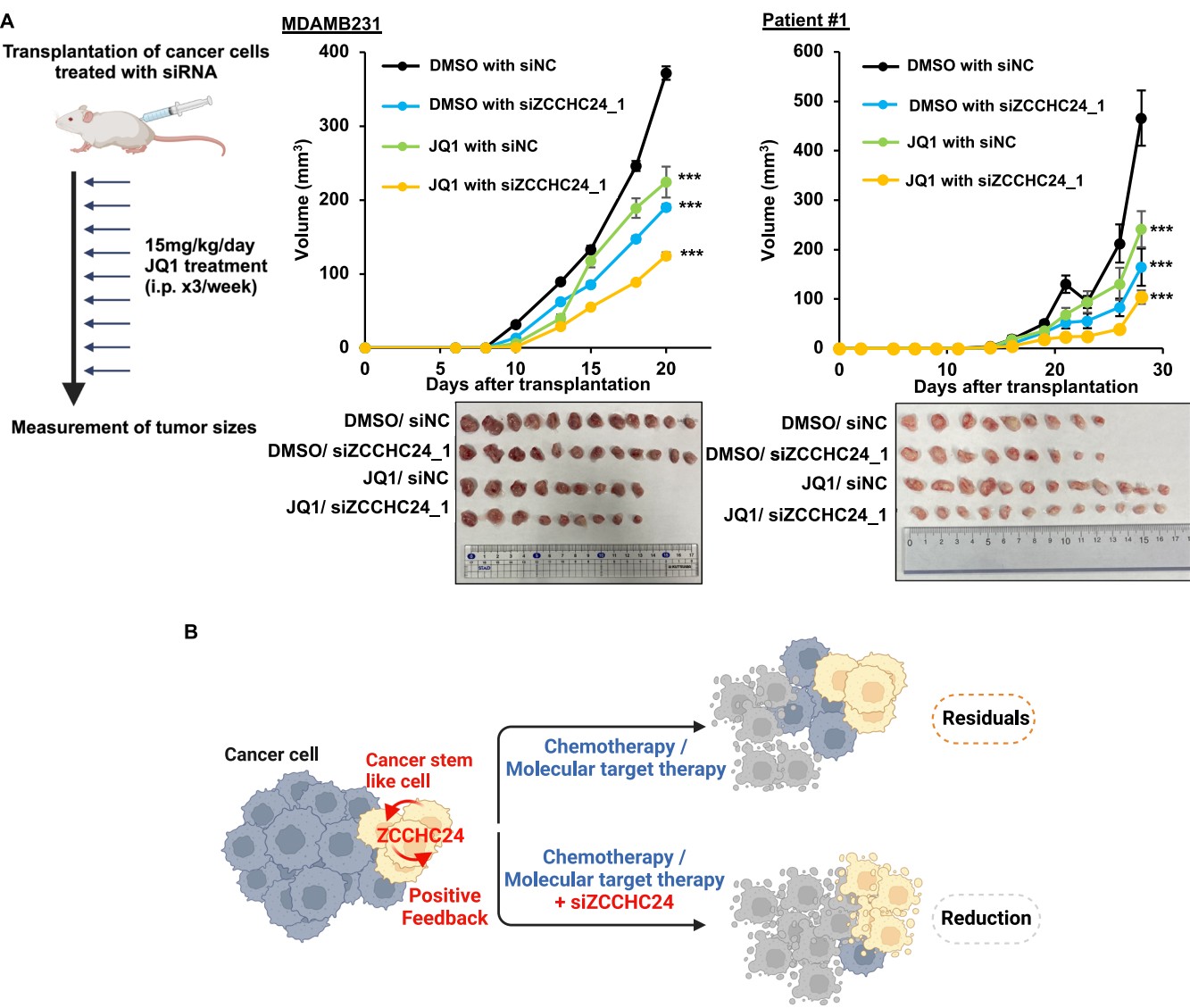

**Figure 6. Combining ZCCHC24 siRNA with a BET inhibitor overcomes chemoresistance.**

(A) In vivo treatment with siRNAs against ZCCHC24 and JQ1. MDAMB231 or PDX (patient #1) knocked down with siRNA for ZCCHC24 or negative control (NC) were transplanted into mice. Mice were intraperitoneally injected with 15 mg/kg/day JQ1 three times per week, and tumor sizes were measured. Differences in tumor size were evaluated using Dunnett's test (DMSO with siNC as the control). (P values: MDAMB231: DMSO with siZCCHC24_1: 0.000189, JQ1 with siNC: 0.0039, JQ1 with siZCCHC24_1: $6.65 \times 10^{-5}$, PDX (Patient #1): DMSO with siZCCHC24_1: $7.69 \times 10^{-4}$, JQ1 with siNC: $3.07 \times 10^{-3}$, JQ1 with siZCCHC24_1: $4.98 \times 10^{-9}$) (MDAMB231: $N = 12$ for DMSO treated samples and $N = 9$ for JQ1 treated samples, PDX: $N = 9$ for DMSO treated samples and $N = 12$ for JQ1 treated samples, ***$P < 0.005$). (B) Schematic representation of the concept. Positive feedback between the transcription factor ZEB1 and RNA-binding protein ZCCHC24 maintains cancer stemness in TNBC and leads to tumor survival. Data information: Data are presented as mean ± SE (A). Source data are available online for this figure.

16 h. Next, cells were washed with PBS twice and incubated with 10% FBS/1% penicillin–streptomycin-containing DMEM. Samples were collected at the time point of 0 h, 1 h, and 2 h. RNA was extracted from cells using TRIZOL (Toyobo). For BRIC-Seq, 30 μg of total RNA from the samples was incubated at 80 °C for 2 min and cooled on ice. Then 5 μg of anti-BrdU antibody (WAKO)-conjugated beads in IP buffer (1% Triton X-100 in PBS) are added to each sample. The samples are incubated at 4 °C for 2 h with rotation and washed with IP buffer four times. Then, the beads are suspended in 400 μL TRIZOL, and RNA is collected. The collected RNA was prepared for next-generation sequencing using polyA

selection. Next-generation sequencing was performed using a NovaSeq 2 × 150 bp.

## BRIC-Seq analysis

For BRIC-Seq samples, adapters were trimmed using Trim-Galore. The trimmed FASTQ files were mapped to the GRCh38 human genome using the STAR software (Dobin et al, 2013). RPKM scores for each sample were calculated with Cuffnorm (Trapnell et al, 2010), and the RPKM scores were used to calculate RNA half-lives with the BridgeR2 package (Imamachi et al, 2014).

## Actinomycin D test

MDAMB231 or HCC38 cells ($5 \times 10^4$ cells/well) were seeded in 24-well plates and cultured for 24 h. The cells were then transfected with the target siRNAs for 48 h. Next, the cells were incubated with 10 μg/mL actinomycin D in DMEM or RPMI. RNA was extracted from the cells at 0, 4, and 8 h after the actinomycin D treatment. *PSMB2* was used as a reference gene for the analysis.

## Photoactivatable ribonucleoside-enhanced crosslinking immune-precipitation (PAR-CLIP)

MDAMB231 cells overexpressing ZCCHC24 in a medium supplemented with doxycycline were UV-cross-linked (365 nm and 300 mJ/cm²) on ice using an ultraviolet crosslinker (UVP, CA, USA). The cells were collected, centrifuged, and incubated with lysis buffer (50 mM Tris-HCl (pH 7.4), 100 mM NaCl, 1% NP-40 (Igepal CA630), 0.1% SDS, 0.5% sodium deoxycholate, and protease inhibitor (1:100)) on ice for 15 min. The sample was sonicated using a bioruptor (Cosmobio, Tokyo, Japan) at a low setting for 5 min at 4 °C with a five 30-s on/off cycle. Next, the sample was incubated with 10 μL RNase I (1:100; Thermo Fisher Scientific, MA, USA) and 2 μL Turbo DNase (Thermo Fisher Scientific) at 37 °C and 1200 rpm for 5 min on a thermomixer. The sample was then treated with 11 μL murine RNase inhibitor and centrifuged at 4 °C for 15 min. Following the manufacturer's instructions, the immunoprecipitation assay was performed using a 1:1000 mouse anti-FLAG antibody (MBL) coupled to Dynabeads Protein G (Thermo Fisher Scientific). The coupled antibodies were washed with lysis buffer and incubated with the whole lysate at 4 °C overnight. Antibody-bound proteins and RNA-protein complexes were isolated using a magnetic stand and washed with cold wash buffer (20 mM Tris-HCl (pH 7.4), 10 mM MgCl₂, 0.2% Tween-20), high-salt wash buffer (50 mM Tris-HCl (pH 7.4), 1 M NaCl, 1 mM ethylenediaminetetraacetic acid, 1% NP-40, 0.1% SDS, and 0.5% sodium deoxycholate), and Fast AP Buffer (10 mM Tris-HCl (pH 7.4), 5 mM MgCl₂, 100 mM KCl, and 0.02% Triton X-100). Bound RNA was incubated with fast alkaline phosphatase (Thermo Fisher Scientific) for 30 min and T4 polynucleotide kinase (PNK) (NEB) for 45 min. Next, the coupled beads were washed with cold wash buffer, high-salt wash buffer, and ligase buffer (50 mM Tris-HCl (pH 7.5) and 10 mM MgCl₂). The bound RNA was ligated with a 3′-RNA linker for 3 h using RNA ligase high-concentration (NEB) and RNA adapters (sequences are shown in oligos). Next, the RNA was washed with cold and high-salt wash buffers. The RNA-protein complex was extracted using NuPAGE sample buffer (Invitrogen) and subjected to SDS-PAGE. The resolved proteins were transferred onto a nitrocellulose membrane (0.2 μm). The blot region with the target protein (30kDa-75kDa) was cut, and RNA was extracted using proteinase K (NEB), acid phenol/chloroform/isoamyl alcohol (Nippon Gene, Tokyo, Japan), and a Quick-RNA Miniprep kit (Zymo Research, CA, USA). The purified RNA was subjected to reverse transcription using the TGIRT-III enzyme (InGex, MO, USA). cDNA was incubated with ExoSAP-IT (Thermo Fisher Scientific). Further, the 5'-end of cDNA was ligated with rand3Tr3 adapter (sequences are shown in supplementary files) using RNA ligase high-concentration (NEB) overnight at room temperature. The adapter-ligated cDNA was PCR-amplified using Q5 PCR enzymes (NEB) for 15 cycles and purified using Ampure XP beads (Beckman Coulter, CA, USA). The samples were then subjected to gel purification. The CLIP-Seq library was sequenced using a Next-Seq 500 (Illumina).

## The analysis of PAR-CLIP data

Adapter sequences in the sequencing reads were removed using Cut-Adapt. The reads were mapped to the GRCh38 genome using the STAR software. Duplicate reads were removed using UMI tools. Bigwig files were created using DeepTools (Ramírez et al, 2016). Bam files from mapping were analyzed with a PARalyzer, and peaks were detected(Corcoran et al, 2011). The binding site regions were analyzed using RCAS (https://www.bioconductor.org/packages/release/bioc/vignettes/RCAS/inst/doc/RCAS.vignette.html). The motif sequences of the binding peaks were calculated using HOMER software (Heinz et al, 2010). Violin plots were constructed by comparing RNA-Seq for MDAMB231 knockdown with ZCCHC24 siRNA.

## Counting the number of motif sites in 3'UTR of genes

Motif sequences "UGUWHWWA" within 3'UTR of mRNAs on all genes were searched on Human Refseq release 215 (Nov, 2022) with GGGenome software (https://gggenome.dbcls.jp/hsnm_refseq215/+/UGUWHWWA). The results were acquired as bed files, and the number of motif sites was calculated for each transcript. Representative transcripts for each gene were selected using MANE software (https://www.ncbi.nlm.nih.gov/refseq/MANE/).

## RNA-immunoprecipitation assay

For the experiment, GFP or ZCCHC24-expressing MDAMB231 cells were seeded in 10-cm dishes are prepared. Cells were treated with 100 ng/mL doxycycline, and after 24 h of incubation, the cells were collected in PBS. In all, 30 μL of Dynabeads Protein G (Thermo Fisher Scientific) per sample was used for the experiment and diluted with PBST and 2 μg per sample of mouse IgG (Wako) or anti-FLAG antibody (M2: Thermo Fisher Scientific). The collected cells were lysed with 1 mL of RIP Lysis Buffer (20 mM Tris-HCl, pH 7.5, 100 mM KCl, 5 mM MgCl₂, 0.5% NP-40, and 0.5% sodium deoxycholate) on ice for 20 min. Samples were centrifuged at 15,000 rpm for 5 min, 2 μL of Turbo DNase (Thermo Fisher Scientific) was added, and samples were incubated at 37 °C for 5 min. Samples were aliquoted into two 1.5-mL tubes, and 20 μL was saved as input samples. 30 μL of normal mouse IgG or anti-FLAG antibodies conjugated beads were added and incubated at 4 °C for 2 h with rotation. Beads were washed with RIP lysis buffer twice, RIP low salt buffer (20 mM Tris-HCl, pH 7.5, 150 mM NaCl, 5 mM MgCl₂, 0.5% Triton X-100) twice, and RIP high-salt buffer (20 mM Tris-HCl, pH 7.5, 300 mM NaCl, 5 mM MgCl₂, 0.5% Triton X-100) twice. The TRI reagent (Cosmo Bio) collected RNA from the beads. Purified RNA was reverse-transcribed using Superscript III reverse transcriptase (Thermo Fisher Scientific) with random primers (Takara) and dNTP (NEB). The enrichment of target transcripts was evaluated by qPCR analyses using the primers shown in the Reagents tools table.

## Reporter assays for post-transcriptional regulation by overexpressed ZCCHC24

In total, $6 \times 10^4$ of 293FT cells were plated into 24-well plates. After 24 h of incubation, 250 ng of the luciferase reporter vector was

transfected with 150 ng of the expression vectors and 50 ng of the Renilla reporter vector. After 48 h of incubation, media was discarded, and 150 µl of luciferase buffer for the Dual-Glo Luciferase Assay System (Promega) was added. After 10 min of incubation in the dark at room temperature, luminescence was measured using a GloMax Discover Microplate Reader (Promega). Then, 150 µl Stop & Glo buffer/substrate mix was added and incubated in the dark at room temperature for 10 min. Luminescence was measured using a GloMax Discover Microplate Reader.

## Reporter assays for the post-transcriptional regulation by knocking down ZCCHC24

Overall, $5 \times 10^4$ cells of MDAMB231 were plated in 24-well plates. After 24 h of incubation, 200 ng of the luciferase reporter vector was transfected with 20 nM siRNA and 400 ng of the Renilla reporter vector. After 48 h of incubation, the media was discarded, and 150 µl of luciferase buffer from the Dual-Glo Luciferase Assay System (Promega) was added. After 10 min of incubation in the dark at room temperature, luminescence was measured using a GloMax Discover Microplate Reader (Promega). Then, 150 µl Stop & Glo buffer/substrate mix was added and incubated in the dark at room temperature for 10 min. Luminescence was measured using a GloMax Discover Microplate Reader.

## Western blotting

Proteins were separated by sodium dodecyl sulfate-polyacrylamide gel electrophoresis, followed by semi-dry transfer to polyvinylidene fluoride membranes. Membranes were blocked with Blocking-One (Nacalai Tesque) for 60 min and incubated with primary antibodies at 4 ℃ overnight. After washing with phosphate-buffered saline containing 0.05% Tween-20 (PBST), the membranes were reacted with ECL mouse or rabbit IgG HRP-conjugated whole antibody (GE Healthcare). Membranes were then washed with PBST three times, and the blot was developed using Pierce ECL Western Blotting Substrate (Thermo Fisher Scientific) and detected using LAS 4000 (GE Healthcare).

## Antibodies for western blotting

Antibodies shown below were used for western blotting; FLAG (M2, Sigma Aldrich), ACTB (AC-74, Sigma Aldrich), ZEB1 (D80D3, Cell Signaling Technology), NRP1 (D62C6, Cell Signaling Technology), CD44 (3578, Cell Signaling Technology), NOTCH2 (D76A6, Cell Signaling Technology), HiBiT (30E5, Promega), and ZCCHC24 (ab243699, Abcam).

## Sphere-formation assay

MDAMB231, HCC38, or PDX (Patient #1) cells ($2 \times 10^5$ cells/well) were seeded in six-well plates and cultured for 24 h. The cells were transfected with si-ZCCHC24 ($N = 8$) and cultured in DMEM F-12 (Gibco) supplemented with 20 ng/mL EGF (R&D Systems), 20 ng/mL FGF (Wako), and B27 supplement for 1 week. The number of spheres was counted for each sample.

## Extreme limiting dilution analysis (ELDA)

MDAMB231, HCC38, or PDX (Patient #1) cells ($2 \times 10^5$ cells/well) were seeded in six-well plates and cultured for 24 h. The cells were transfected with si-ZCCHC24 ($N = 8$) and cultured in DMEM F-12 (Gibco) supplemented with 20 ng/mL EGF (R&D Systems), 20 ng/mL FGF (Wako), and B27 supplement for 1 week at 250, 125, 62, and 31 cells per well. The formation of spheres was also observed. The data were analyzed on a website (http://bioinf.wehi.edu.au/software/elda/).

## In vivo colony-formation assay

MDAMB231 or PDX (Patient #1) cells ($1 \times 10^6$ cells/well) were plated in a 10-cm dish and cultured for 24 h. The cells were transfected with siRNA for 24 h. Next, various densities ($1 \times 10^4$, $1 \times 10^3$, and $1 \times 10^2$ cells) of transfected cells were subcutaneously transplanted into 5-week-old female nude mice (MDAMB231) or 6–7-week-old NOG mice with 50% Matrigel (Corning). Mice were randomized prior to tumor cell injection. After 1 month, colonies were extracted from the mice and examined. The data were analyzed on a website (http://bioinf.wehi.edu.au/software/elda/).

## Single-cell RNA Seq for PDXs knocked down with siRNA for ZCCHC24

PDX (Patient #1) cells ($1 \times 10^6$ cells/well) were plated in a 10-cm dish and cultured for 24 h. Cells were transfected with siRNA for 24 h. Eighty percent cell viability was confirmed by Trypan Blue staining. Next, $1 \times 10^3$ transfected cells were transplanted into 6–7-week-old female NOG mice with 50% Matrigel (Corning). Three weeks after transplantation, fresh breast cancer tissues were extracted, and murine tissues were removed. Then, tissues were dissociated into single cells using collagenase (FUJIFILM Wako, 032-22364), and red blood cells were excluded using RBC lysis buffer (Invitrogen, 00-4333-57). Single live cells were loaded onto a 10X Chromium Single Cell Capture Chip, followed by single-cell capture reverse transcription and library preparation according to the manufacturer's protocols. The constructed library was sequenced using NovaSeq ($2 \times 150$ bp). Analyses of single-cell RNA-seq data followed the protocol for single-cell RNA-seq analyses of PDX sections.

## Reanalysis of Hi-C, chromatin immunoprecipitation (ChIP) and ATAC-Seq

Hi-C for MDAMB231 was obtained from PRJEB29716 (Beesley et al, 2020). The fastq file was mapped to the GRCh38 genome using STAR software. The mapped BAM file was analyzed using Hi-C Explorer (Wolff et al, 2018). ATAC-Seq for MDAMB231 was obtained from the EMBL-EBI database E-MTAB-8264 (Feldker et al, 2020), and ChIP-Seq files for MDAMB231 were obtained from EMBL-EBI database E-MTAB-8264 (ZEB1) (Feldker et al, 2020), GEO database GSE166941 (YAP) (He et al, 2021), and GSE66083 (JUN, H3K4Me1, H3K27Ac) (Zanconato et al, 2015). Using STAR software, Fastq files were mapped onto the GRCh38 genome. Using Deeptools software, all BAM files were converted to Bigwig files.

## Reporter assay for ZEB1, AP-1 and YAP

Overall, $5 \times 10^4$ cells/well MDAMB231 were plated in 24-well plates. After 24 h of incubation, 100 ng of the reporter vector was

transfected with 100 ng of ZEB1, 50 ng of JUN, 50 ng of FOS, 100 ng of YAP, and 50 ng of the Renilla reporter. After 48 h incubation, media was discarded, and 150 µl of luciferase buffer from the Dual-Glo Luciferase Assay System (Promega) was added. After 10 min of incubation in the dark at room temperature, luminescence was measured using a GloMax Discover Microplate Reader (Promega). Then, 150 µl of Stop & Glo buffer/substrate mix was added and incubated in the dark at room temperature for 10 min. Luminescence was measured using a GloMax Discover Microplate Reader.

## Chromatin immunoprecipitation (ChIP) qPCR analyses

MDAMB231 cells in 10 cm dishes were transfected with siRNA against ZEB1 or a negative control. After 48 h of incubation, cells were fixed with 1% formaldehyde for 10 min at room temperature to allow chromatin crosslinking, followed by quenching of the reaction with 0.125 M glycine. The cells were washed in phosphate-buffered saline (PBS) and collected. After centrifugation, the pellet was lysed in ChIP cell swelling buffer (25 mM HEPES-KOH, pH 7.5, 1.5 mM MgCl$_2$, 1 mM EDTA, 0.5% NP-40) and incubated on ice for 10 min. After incubation, the cells were centrifuged, and the pellet was collected. The pellet was lysed with 2 mL of ChIP lysis buffer (50 mM Tris-HCl (pH 8.0), 10 mM EDTA (pH 8.0), 1% SDS) and sheared by sonication with Picoruptor (Diagenode) for 40 cycles. After sonication, the same amount of ChIP dilution buffer (50 mM HEPES-KOH, pH 7.5, 150 mM NaCl, 1 mM EDTA, 1% Triton X-100) was added to the sample, and 50 µl of the samples were collected as input control samples. The lysed samples were then aliquoted to two samples, and 15 µg of normal rabbit IgG (Wako) or anti-H3K27Ac antibody (D5E4, Cell Signaling Technology) were added and incubated for 24 h. Next, 20 µL per sample of Dynabeads Protein A (Thermo Fisher Scientific) was added to samples and incubated for two hours. The beads were washed with ChIP lysis buffer twice, low salt wash buffer (20 mM Tris-HCl, pH 8.0, 150 mM NaCl, 2 mM EDTA, 1% Triton X-100, 0.1% SDS) twice, high-salt wash buffer (20 mM Tris-HCl, pH 8.0, 500 mM NaCl, 2 mM EDTA, 1% Triton X-100, 0.1% SDS) twice, LiClL wash buffer (10 mM Tris-HCl, pH 8.0, 250 mM LiCl, 1 mM EDTA, 1% NP-40, 1% deoxycholate) twice and TE buffer twice. Then the beads were resuspended in 25 µL of ChIP elution buffer (0.1 M NaHCO$_3$, 1 mM EDTA, 1% SDS) twice and incubated at 55 °C for 15 min twice. Finally, 5 µL of 5 M NaCl and 1 µL of RNase I (Thermo Fisher Scientific) were added and incubated at 55 °C overnight. ChIP DNA was collected using the NucleoSpin Gel and PCR Clean-up kit (Takara). The collected DNA was examined by qPCR using the primers listed in the reagents and tools table.

## Establishment of HiBiT tag knock-in cells

HiBiT-tagged MDAMB231 cells were generated using the CRISPR-Cas9 system. ALT-R XT CRISPR RNA (crRNA) (Integrated DNA Technologies, Coralville, IA, USA) was resuspended in a nuclease-free Duplex Buffer (Integrated DNA Technologies) to a final concentration of 100 µM. (The sequence of crRNA: 5′-ACTGCCGTCGCGTGCAGTGA-3′). Equal crRNA and *trans*-activating CRISPR RNA volumes were mixed and heated for 5 min at 95 °C. After heating, the oligonucleotide complexes were cooled gradually to room temperature. The oligo complex was then

incubated at room temperature for 20 min with ALT-R Cas9 Nuclease V3 (61 µM) (Integrated DNA Technologies) to form the Cas9 complex. The single-stranded DNA (ssDNA) oligo (The sequence of oligo: 5′-AGAGCAAGGAGCACCCGCAGCACCTC TGCGAGAAGTGCAAGGTCCTGGGCTACTACTGCCGTCGCG TGCAGGACTACAAGGACGACGACGACAAGGTGAGCGGCT GGCGGCTGTTCAAGAAGATTAGCTGACGGGCTGCCCGCCC GCACCCAGAGCCACCCCCCGCCAGCCCGAGGAGACGCTGC TTCCCTGTGCTACTC -3′), including sequences of HiBiT, a complementary sequence to the C-terminal in the ZCCHC24 genome, and the Cas9 complex were then co-transfected into the MDAMB231 cells using the NEPA21 Super Electroporator (NEPAGENE, Chiba, Japan). After incubating the cells for a few days, single-cell cloning was performed to identify HiBiT-tagged cells. Sequencing analysis of the knock-in regions and western blotting analyses with an anti-HiBiT antibody (Promega) were performed for the knock-in validation. Moreover, to validate the functionality of the tag, we knocked down ZCCHC24 with siRNA, performed the HiBiT lytic assay described below, and confirmed a decrease in luminescence caused by the HiBiT tag.

## HiBiT lytic assay

Before the examination, the media in 96-well plates was discarded. Next, 12.5 µl of the lytic buffer of Nano Glo HiBiT Lytic Detection System (Promega) with 12.5 µl of PBS, 0.25 µl of the substrate, and 0.125 µl of LgBiT of the kit were added, and incubated in the dark for 10 min. The luminescence was measured using a GloMax Discover Microplate Reader (Promega).

## Reanalysis of TCGA-BRCA RNA-Seq data

FPKM data for each gene from RNA-Seq of breast cancer patients were obtained from TCGA-BRCA datasets (https://portal.gdc.cancer.gov/).

## Reanalysis of RNA expression datasets of clinical trials

RNA expression datasets were obtained from GSE25055 and GSE164458 clinical trials. For the GSE25055 dataset, clinical stage III samples were used for analysis. The cutoff for RNA expression of ZCCHC24 was set to 8.5, and the pathologic response (pCR or RD) was calculated for high or low ZCCHC24 samples. For the GSE164458 dataset, samples of the planned arm code C (neoadjuvant therapy with taxane only) were used for the analysis. The cutoff for RNA expression of ZCCHC24 was set to 3.0, and the pathologic response (pCR or RD) was calculated for high or low ZCCHC24 samples.

## In vitro drug treatment assay

In all, $5 \times 10^4$ cells of MDAMB231 or PDX (patient #1) were seeded in 24-well plates. After 24 h, 1 µM of JQ1, doxorubicin (Selleck), or 0.1% DMSO are added. After 24 h of incubation, the RNA was extracted from the cells.

## In vitro cell viability assay

In total, $4 \times 10^3$ MDAMB231 cells or $5 \times 10^3$ PDX (patient #1) cells were plated in 96-well plates. After 24 h, 1 µM (MDAMB231 cells) or

500 nM (patient #1) doxorubicin (Selleck) was added to cells, while siRNA against ZCCHC24 or negative control was transfected with Lipofectamine RNA iMax at the final concentration 20 nM. After 48 h, cell viability in each well was detected using the RealTime-Glo MT Cell Viability Assay kit (Promega). Luminescence was detected using a GloMax Discover Microplate Reader (Promega).

### In vivo treatment assay

Six-to-seven-week-old NOG or nude mice were used for the experiments. In total, $5 \times 10^4$ MDAMB231 or PDX cells transfected with ZCCHC24 siRNA or negative control were subcutaneously transplanted into mammary fat pads of the mice. Mice were randomized prior to tumor cell injection and drug treatments. In all, 15 mg/kg/day of JQ1 (with 2% DMSO, 30% PEG300, 5% Tween 80, and ddH2O) was intraperitoneally injected three times per week. The tumor size was measured using the formula $V = L \times S^2 \times 0.52$. No blinding was done in the measurements of the tumor size.

### Patients and pathological specimens

We examined pathological specimens obtained from four patients with TNBC at the Tokyo Medical and Dental University Hospital, Tokyo, between 2000 and 2021. Pathologists confirmed the pathological diagnosis according to the WHO Health Organization criteria. Specimens were obtained by surgical resection, routinely fixed in 10% neutralized formalin, and embedded in paraffin for conventional histopathological examination. Informed consent was obtained from all patients. This study was approved by the ethics committees of Tokyo Medical and Dental University, and all procedures were performed in accordance with the ethical standards established by these committees (M2021-006).

### Immunohistochemistry

Formalin-fixed, paraffin-embedded (FFPE) tissue was sliced (4-μm thickness), and the sections were placed on silane-coated slides. Post deparaffinization, heat-based antigen retrieval at 95 °C for 20 min in citrate buffer (pH 6.2), and blocking with blocking buffer (containing Alexa Fluor™ 488 Tyramide SuperBoost™ Kit, goat anti-rabbit IgG (B40943)) were performed. The primary antibodies used were anti-ZCCHC24 (LS-bio, c817634, 1:250) and anti-ZEB1 (R and D Systems, MAB6708, 1:250). The primary antibodies were then incubated sequentially. Specimens treated with primary antibodies were incubated overnight at 4 °C. After reacting, the primary antibody, Alexa Fluor™ 488 Tyramide SuperBoost™ Kit, goat anti-rabbit IgG (B40943), and Alexa Fluor™ 594 Tyramide SuperBoost™ Kit, goat anti-mouse IgG (B40942) were also used sequentially to enhance the signal as directed by the manufacturer (Invitrogen). After staining with Hoechst 33342 solution (Dojindo, 1:10,000), the slides were mounted using ProLong Glass Antifade Mountant (Invitrogen). The images were captured using a BC43 confocal microscope (Andor). Images were analyzed using Imaris Viewer software (v10-1-0, Oxford).

### Statistical analysis

The text and figure legends indicate statistical methods and significance values, respectively. Error bars represent the standard error of the mean of independent experiments or samples. No samples nor animals were excluded from the experiments performed in this manuscript. Analyses of images and animal experiments were blinded and randomized where possible.

## Data availability

The datasets produced in this study are available in the following databases: Raw scRNA-seq data, transcriptome analysis, PAR-CLIP, and BRIC-Seq: Sequence Read Archive DRA016408 (https://ddbj.nig.ac.jp/resource/sra-submission/DRA016408).

The source data of this paper are collected in the following database record: biostudies:S-SCDT-10_1038-S44319-024-00282-8.

## Peer review information

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

## Acknowledgements

The authors thank Yuki Naito, Lin Liu, Kana Shishido, Tomomi Kato, Mitsuyo Nakajima, and all the staff of the Department of Systems Biomedicine at Tokyo Medical and Dental University (TMDU) for their support and advice; Tsuyoshi Nakagawa for kindly providing pathological samples; Miori Inoue for preparing pathological samples; Girls Pickersgill for scientific editing; and Editage for language editing. This work was supported by the Japan Society for the Promotion of Science, KAKENHI (Grant No. JP20H05696 and JP21K19403 to HA), and an NIH Grant No (Grant No. AR080127 to HA) and AMED-LEAP from AMED (Grant No. JP23gm0010009 to HA). This work was also partially supported by an Extramural Collaborative Research Grant from the Cancer Research Institute, Kanazawa University.

## Author contributions

**Yutaro Uchida**: Conceptualization; Data curation; Formal analysis; Validation; Investigation; Visualization; Methodology; Writing—original draft. **Ryota Kurimoto**: Resources; Investigation; Visualization; Methodology; Writing—original draft. **Tomoki Chiba**: Validation; Investigation; Visualization; Methodology; Writing—review and editing. **Takahide Matsushima**: Investigation; Visualization; Methodology; Writing—review and editing. **Goshi Oda**: Resources; Writing—review and editing. **Iichiroh Onishi**: Resources;

Writing—review and editing. **Yasuto Takeuchi**: Resources; Methodology; Writing—review and editing. **Noriko Gotoh**: Resources; Methodology; Writing—review and editing. **Hiroshi Asahara**: Conceptualization; Resources; Formal analysis; Supervision; Funding acquisition; Methodology; Project administration; Writing—review and editing.

Source data underlying figure panels in this paper may have individual authorship assigned. Where available, figure panel/source data authorship is listed in the following database record: biostudies:S-SCDT-10_1038-S44319-024-00282-8.

## Disclosure and competing interests statement

The authors declare no competing interests.

# Expanded View Figures

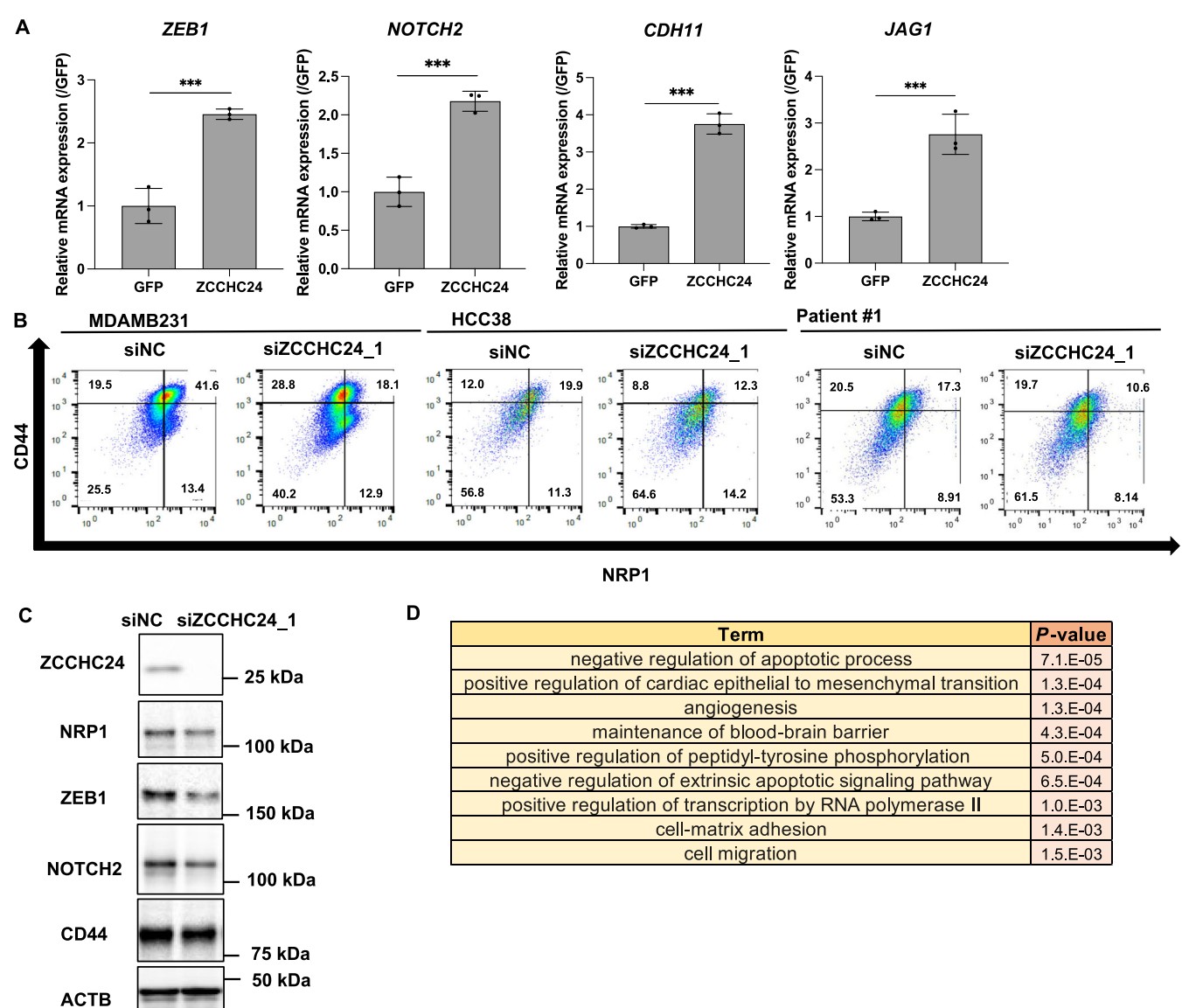

**Figure EV1.  Quantitative PCR for MDAMB231 overexpressing ZCCHC24 and FACS / Western blotting analysis for MDAMB231 or PDX knocked down with siRNA for ZCCHC24.**

(A) qPCR analysis of MDAMB231 cells overexpressing ZCCHC24 or GFP as controls. Changes in expression were assessed using unpaired *t* tests. (*P* values: *NOTCH2*: $9.7 \times 10^{-4}$, *ZEB1*: 0.0023, *CDH11*: $6.68 \times 10^{-5}$, JAG1: $9.0 \times 10^{-4}$) (\*\*\**P* < 0.005, *N* = 3 biological replicates each). (B) FACS analysis (CD44 and NRP1) of TNBC cell lines (MDAMB231 and HCC38) or PDX (Patient #1) knocked down with siRNA for ZCCHC24. (C) Western blotting analysis for MDAMB231 knocked down with siRNA for ZCCHC24. (D) Gene ontology analysis of common differentially expressed genes in the RNA-Seq analysis and destabilized genes in the BRIC-Seq analysis. The specificity of the gene ontology was tested using DAVID software (https://david.ncifcrf.gov/tools.jsp), following the manufacturer's protocol. Data information: Data are presented as mean ± SD (A).

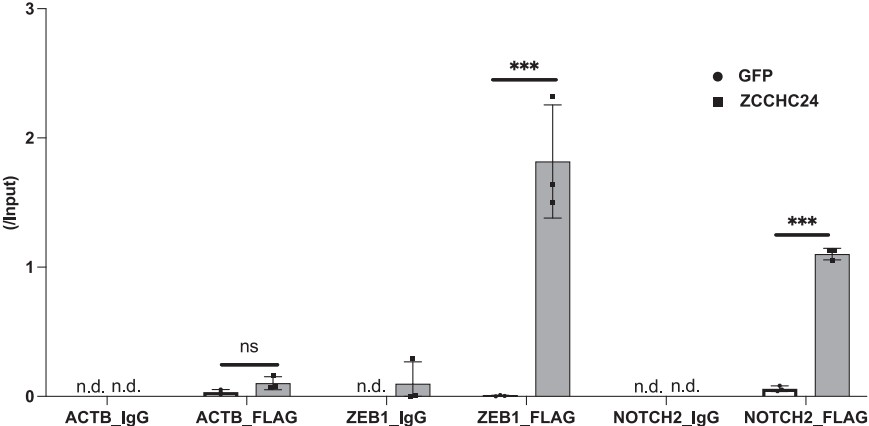

**Figure EV2. ZCCHC24 binds to mRNA of genes important for tumor progression and breast cancer stemness.**

RIP-qPCR analysis of ZCCHC24. Differences in enrichment were tested using analysis of variance, followed by Tukey's post hoc test. ($P$ values: *ACTB* FLAG: 1.0, *ZEB1* FLAG: $1.25 \times 10^{-12}$, *NOTCH2* FLAG: $1.11 \times 10^{-7}$) ($N = 3$ biological replicates each, ***$P < 0.005$, n.d.: not detected, ns: not significant). Data information: Data are presented as mean ± SD.

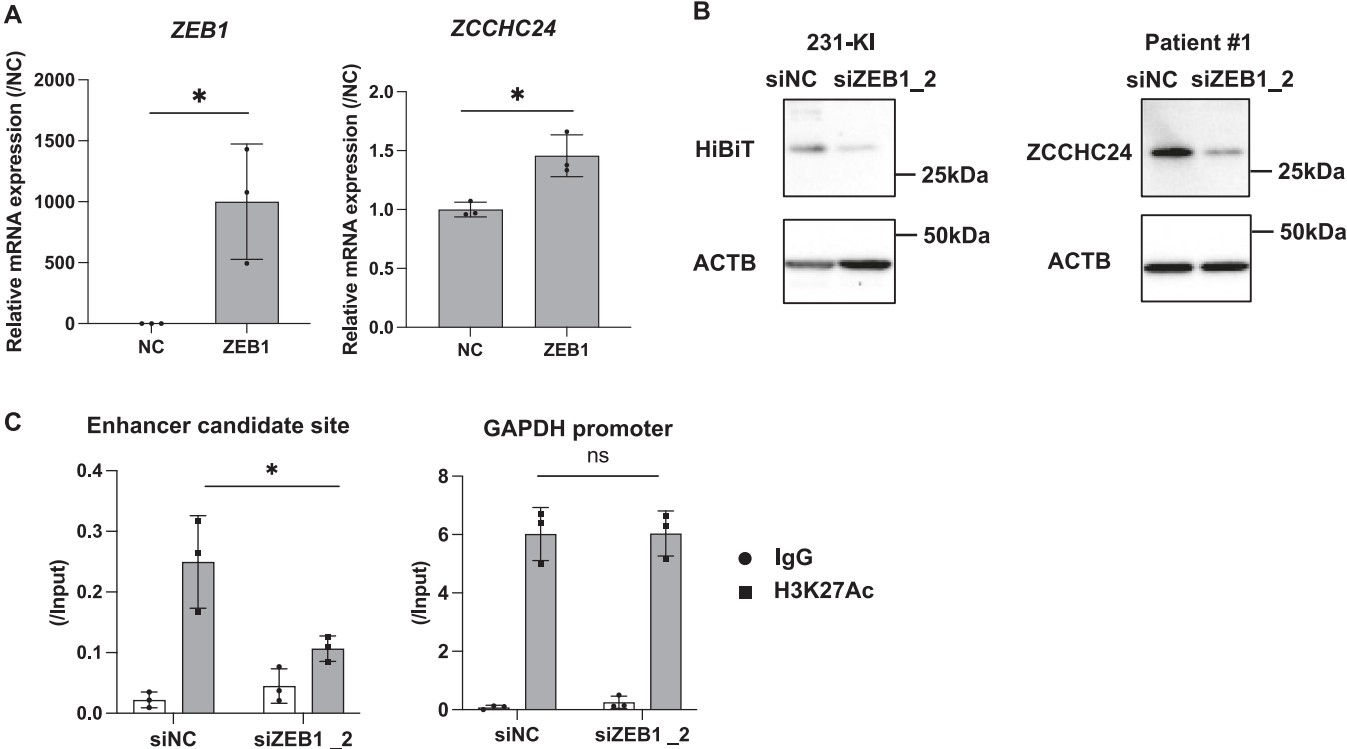

**Figure EV3.  ZEB1, JUN, and YAP transcriptionally regulate ZCCHC24.**

(**A**) qPCR analysis of MDAMB231 overexpressing ZEB1. Changes in gene expression were analyzed using an unpaired t test. (*P* value: *ZEB1*: 0.022, *ZCCHC24*: 0.014) (*N* = 3 biological replicates each, *\*P* < 0.05). (**B**) Western blot analysis of MDAMB231 and PDX (Patient #1) cells knocked down with siRNA for ZEB1 or the negative control (NC). (**C**) Chromatin immunoprecipitation (ChIP) analysis of MDAMB231 knocked down with siRNA against ZEB1 or negative control (NC). Differences in enrichment were tested using analysis of variance, followed by Tukey's post hoc test. (*P* values: Enhancer candidate site: 0.014, *GAPDH* promoter: 1.0) (*N* = 3 biological replicates each, *\*P* < 0.05). Data information: Data are presented as mean ± SD (**A**, **C**).

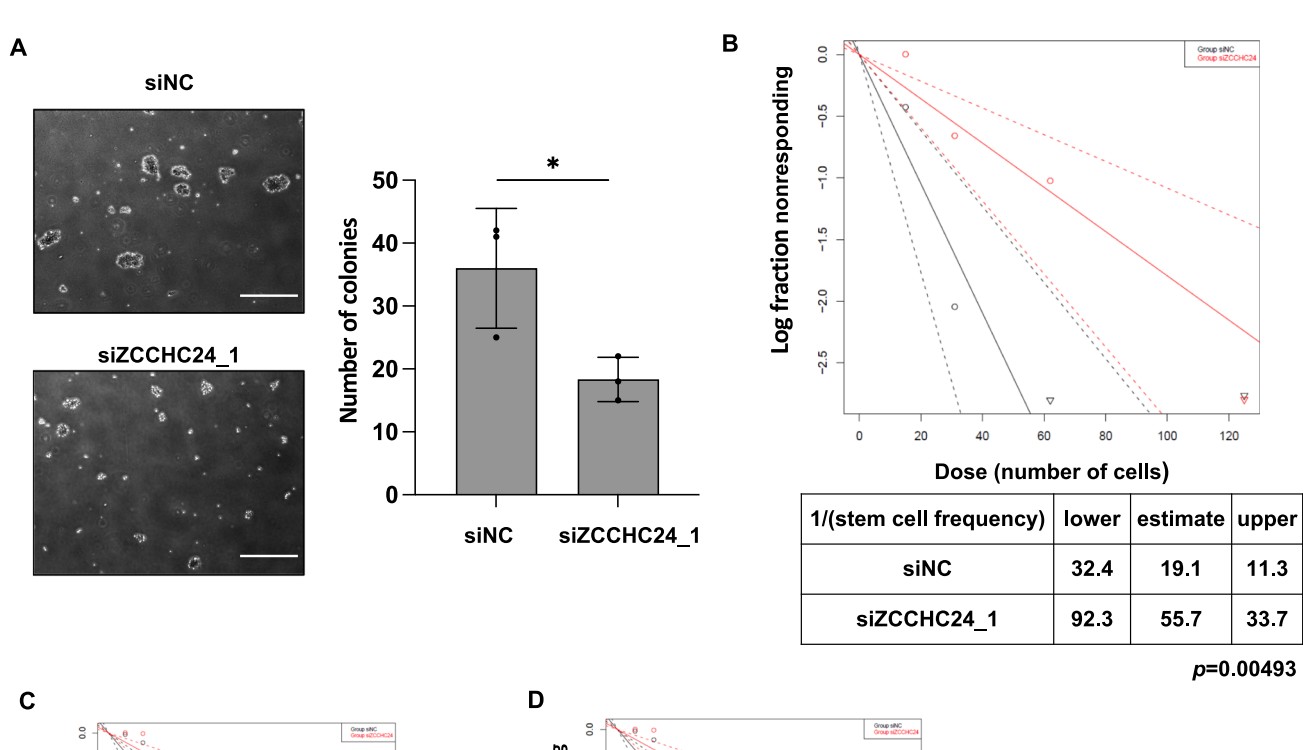

**A**

siNC

siZCCHC24_1

**B**

| 1/(stem cell frequency) | lower | estimate | upper |
|---|---|---|---|
| siNC | 32.4 | 19.1 | 11.3 |
| siZCCHC24_1 | 92.3 | 55.7 | 33.7 |

*p*=0.00493

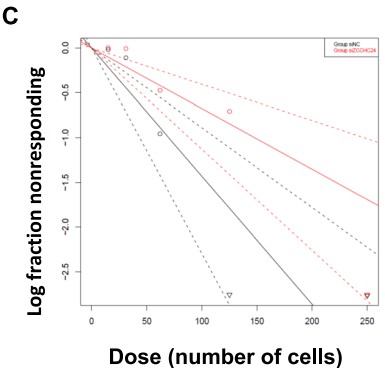

**C**

| Group | Lower | Estimate | Upper |
|---|---|---|---|
| siNC | 112 | 69.9 | 43.5 |
| siZCCHC24_1 | 247 | 147.8 | 88.5 |

*p*=0.0228

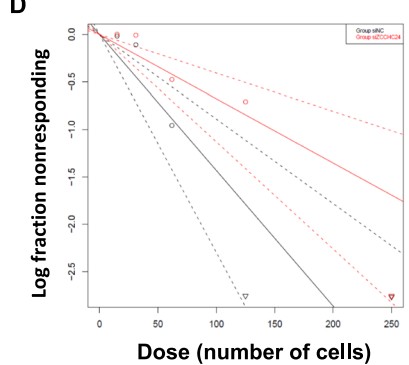

**D**

| Group | Lower | Estimate | Upper |
|---|---|---|---|
| siNC | 112 | 69.9 | 43.5 |
| siZCCHC24_1 | 247 | 147.8 | 88.5 |

*p*=0.045

**E** MDAMB231   $10^4$ cells/site   $10^3$ cells/site   $10^2$ cells/site

siNC
siZCCHC24_1

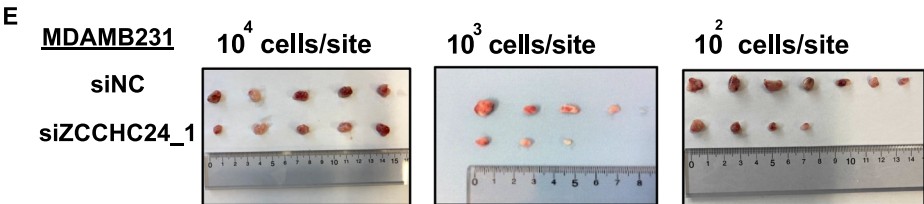

| | Cells per site | | | *p* value vs control | 1/(stem cell frequency) (Lower-Upper) |
|---|---|---|---|---|---|
| | $10^4$ | $10^3$ | $10^2$ | | |
| siNC | 5/5 | 5/5 | 7/8 | | 48.6 (117-20.4) |
| siZCCHC24_1 | 5/8 | 4/5 | 4/8 | $1.37×10^{-4}$ | 520.5 (1302-208.2) |

◀

**Figure EV4. ZCCHC24 knockdown downregulates tumor formation in vitro and in vivo.**

(A) Sphere-formation assay of MDAMB231 cells knocked down with ZCCHC24 siRNA. Differences in the number of spheres formed were tested using unpaired $t$ tests. ($P$ value = 0.040) ($N = 3$ biological replicates each; *$P < 0.05$). (B) In vitro extremely limited dilution assay (ELDA) of MDAMB231 knocked down with siRNA for ZCCHC24. Tumor formation ability was tested using the likelihood ratio test of the single-hit model, as shown on the ELDA software website (https://bioinf.wehi.edu.au/software/elda/) by the manufacturer. ($P$ value = 0.00493). (C) In vitro extremely limited dilution assay (ELDA) of HCC38 knocked down with siRNA for ZCCHC24. Tumor formation ability was tested using the likelihood ratio test of the single-hit model, as shown on the ELDA software website by the manufacturer. ($P$ value = 0.0228). (D) In vitro extremely limited dilution assay (ELDA) of PDX (patient #1) knocked down with siRNA for ZCCHC24. Tumor formation ability was tested using the likelihood ratio test of the single-hit model, as shown on the ELDA software website by the manufacturer. ($P$ value = 0.045). (E) Comparison of tumor formation ability in vivo of MDAMB231 knocked down with siRNA for ZCCHC24 or negative control. The cells were transplanted into seven-week-old female nude mice. Tumor formation ability was tested using the likelihood ratio test of the single-hit model, as shown on the manufacturer's ELDA software website ($P$ value: $1.37 \times 10^{-4}$). Data information: Data are presented as mean ± SD (A). (A) Scale bar 50 μm.

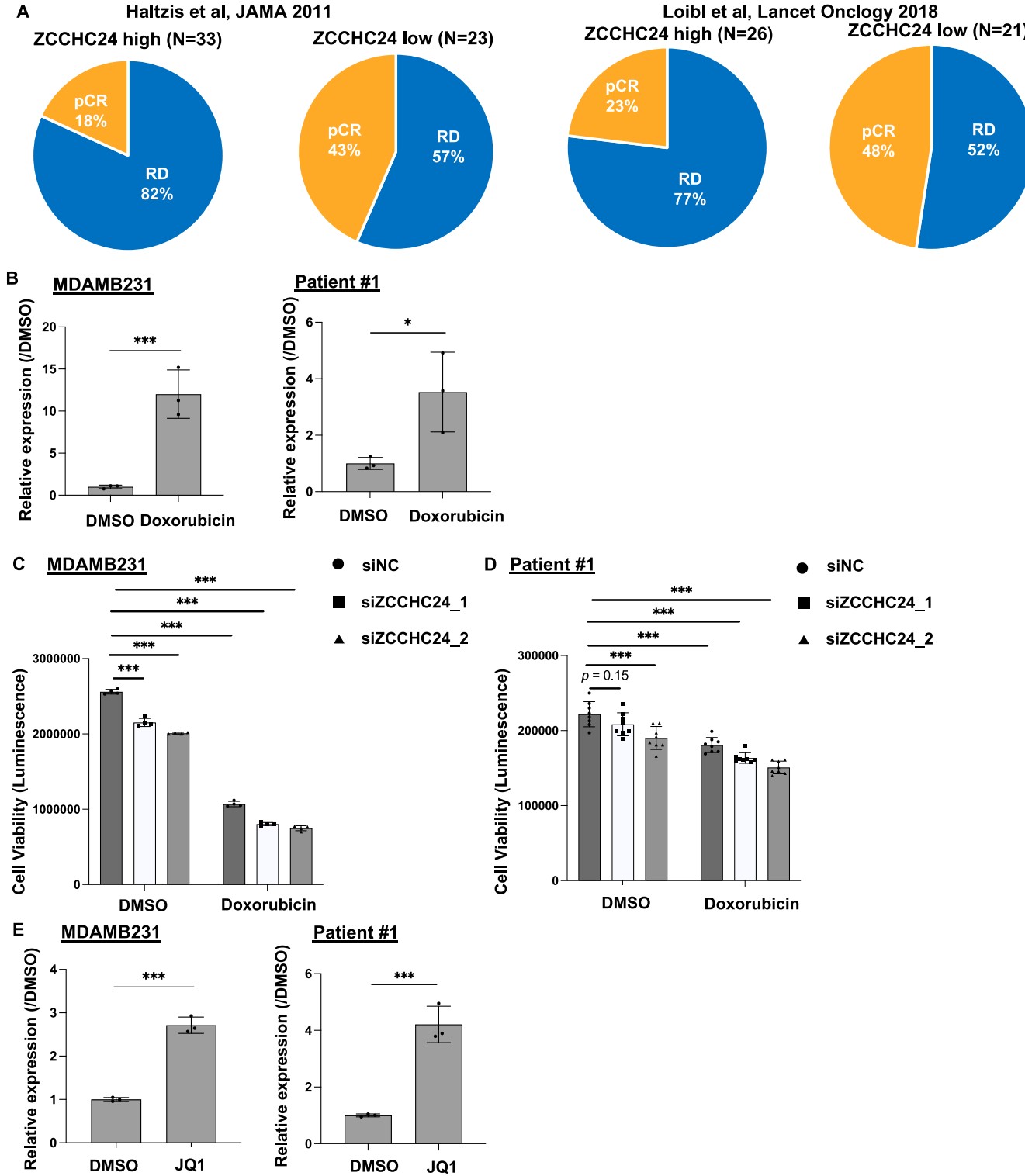

◀

**Figure EV5.  ZCCHC24 expression and clinical prognosis in clinical trials and combined use of a chemotherapy drug and siRNA against ZCCHC24.**

(**A**) Clinical prognosis (pathologic complete response (pCR) and residual disease (RD)) and RNA expression of ZCCHC24 in clinical trials (Hatzis et al, 2011; Loibl et al, 2018) with neoadjuvant chemotherapy. (**B**) qPCR analyses of *ZCCHC24* for MDAMB231 or PDX (Patient #1) with the addition of 0.1% DMSO or 1 μM doxorubicin. Changes in expression were assessed using unpaired *t* tests. (*P* values: MDAMB231:0.0027, PDX (Patient #1): 0.037) (*N* = 3 biological replicates each, *$P < 0.05$, ***$P < 0.005$). (**C**) Cell viability assay of MDAMB231 with siRNA transfection and treated with 0.1% DMSO or 500 nM doxorubicin. Differences in cell viability were tested using Dunnett's test (DMSO with siNC as the control). (*P* values: DMSO with siZCCHC24_1: $P < 2.22 \times 10^{-16}$, DMSO with siZCCHC24_2: $P < 2.22 \times 10^{-16}$, doxorubicin with siNC: $P < 2.22 \times 10^{-16}$, doxorubicin with siZCCHC24_1: $P < 2.22 \times 10^{-16}$, doxorubicin with siZCCHC24_2: $P < 2.22 \times 10^{-16}$) (*N* = 4 biological replicates each, ***$P < 0.005$). (**D**) Cell viability assay for PDX (Patient #1) with siRNA transfection and addition of 0.1% DMSO or 1 μM doxorubicin. Differences in cell viability were tested using Dunnett's test. (DMSO with siNC was used as a control). (*P* values: DMSO with siZCCHC24_1: 0.145, DMSO with siZCCHC24_2: $5.61 \times 10^{-5}$, doxorubicin with siNC: $9.77 \times 10^{-7}$, doxorubicin with siZCCHC24_1: $5.43 \times 10^{-12}$, doxorubicin with siZCCHC24_2: $1.67 \times 10^{-14}$) (*N* = 8, biological replicates each, ***$P < 0.005$). (**E**) qPCR analyses of *ZCCHC24* for MDAMB231 or PDX (Patient #1) with the addition of 0.1% DMSO or 1 μM JQ1. Changes in expression were assessed using unpaired *t* tests. (*P* alues: MDAMB231: $1.1 \times 10^{-4}$, PDX (Patient #1): $9.9 \times 10^{-4}$) (*N* = 3 biological replicates each, ***$P < 0.005$). Data information: Data are presented as mean ± SD (**B**–**E**).

