## [Peer Review File · EMBO Reports]

RNA binding protein ZCCHC24 promotes tumorigenicity in triple-negative breast cancer

Yutaro Uchida, Ryota Kurimoto, Tomoki Chiba, Takahide Matsushima, Goshi Oda, Ichiroh Onishi, Yasuto Takeuchi, Noriko Gotoh, and Hiroshi Asahara

Corresponding author(s): Hiroshi Asahara (asahara.syst@tmd.ac.jp)

Review Timeline:

Submission Date:	26th Feb 24
Editorial Decision:	28th Mar 24
Revision Received:	27th Jun 24
Editorial Decision:	18th Jul 24
Revision Received:	26th Jul 24
Editorial Decision:	23rd Aug 24
Revision Received:	28th Aug 24
Accepted:	26th Sep 24

Editor: Achim Breiling

Transaction Report:

Dear Prof. Asahara,

Thank you for the submission of your research manuscript to EMBO reports. I have now received the reports from the three referees that were asked to evaluate your study, which can be found at the end of this email.

As you will see, the referees think that the findings are of interest. However, referees #1 and #3 have several comments, concerns, and suggestions, indicating that a major revision of the manuscript is necessary to allow publication of the study in EMBO reports. From the analysis of the referee comments it is clear that a significant revision is required before publication can be considered, and I would also understand your decision if you chose to rather seek rapid publication elsewhere at this stage.

However, also considering the evaluation of referee #2, I would like to give you the opportunity to address the concerns and would be willing to consider a revised manuscript with the understanding that all referee concerns must be addressed in the revised manuscript and in a detailed point-by-point response. Acceptance of your manuscript will depend on a positive outcome of a second round of review. It is EMBO reports policy to allow a single round of revision only and acceptance of the manuscript will therefore depend on the completeness of your responses included in the next, final version of the manuscript.

- 1) a .docx formatted version of the final manuscript text (including legends for main figures, EV figures and tables), but without the figures included. Figure legends should be compiled at the end of the manuscript text.
- 2) individual production quality figure files as .eps, .tif, .jpg (one file per figure), of main figures and EV figures. Please upload these as separate, individual files upon re-submission.

- 4) a complete author checklist, which you can download from our author guidelines

(<https://www.embopress.org/page/journal/14693178/authorguide>). Please insert page numbers in the checklist to indicate where the requested information can be found in the manuscript. The completed author checklist will also be part of the RPF.

5) that primary datasets produced in this study (e.g. RNA-seq, CHIP-seq, structural and array data) are deposited in an appropriate public database. If no primary datasets have been deposited, please also state this in a dedicated section (e.g. 'No primary datasets have been generated and deposited'), see below.

The accession numbers and database should be listed in a formal "Data Availability" section (placed after Materials & Methods) that follows the model below. This is now mandatory (like the COI statement). Please note that the Data Availability Section is restricted to new primary data that are part of this study. This section is mandatory. As indicated above, if no primary datasets have been deposited, please state this in this section

Data availability

8) Regarding data quantification and statistics, please make sure that the number "n" for how many independent experiments were performed, their nature (biological versus technical replicates), the bars and error bars (e.g. SEM, SD) and the test used to calculate p-values is indicated in the respective figure legends (also for EV figures and all those in an Appendix). Please also check that all the p-values are explained in the legend, and that these fit to those shown in the figure. Please provide statistical testing where applicable. Please avoid the phrase 'independent experiment', but clearly state if these were biological or technical replicates. Please also indicate (e.g. with n.s.) if testing was performed, but the differences are not significant. In case n=2, please show the data as separate datapoints without error bars and statistics. See also: <http://www.embopress.org/page/journal/14693178/authorguide#statisticalanalysis>

9) Please add scale bars of similar style and thickness to microscopic images, using clearly visible black or white bars (depending on the background). Please place these in the lower right corner of the images themselves. Please do not write on or near the bars in the image but define the size in the respective figure legend.

10) Please also note our reference format:

12) We now use CRediT to specify the contributions of each author in the journal submission system. CRediT replaces the author contribution section. Please use the free text box to provide more detailed descriptions and do not provide your final manuscript text file with an author contributions section. See also our guide to authors: <https://www.embopress.org/page/journal/14693178/authorguide#authorshipguidelines>

13) We would encourage you to use 'Structured Methods', our new Materials and Methods format. According to this format, the Materials and Methods section should include a Reagents and Tools Table (listing key reagents, experimental models, software, and relevant equipment and including their sources and relevant identifiers), uploaded as separate file, followed by a Methods and Protocols section in which we encourage the authors to describe their methods using a step-by-step protocol format with bullet points, to facilitate the adoption of the methodologies across labs. More information on how to adhere to this format as well as downloadable templates (.doc or .xls) for the Reagents and Tools Table can be found in our author guidelines (section 'Structured Methods'):

14) Please shorten the abstract to not more than 175 words and order the manuscript sections like this, using these names: Title page - Abstract - Keywords - Introduction - Results - Discussion - Materials and Methods - Data availability section - Acknowledgements - Disclosure and Competing Interests Statement - References - Figure legends - Expanded View Figure legends

I look forward to seeing a revised version of your manuscript when it is ready. Please let me know if you have questions or comments regarding the revision.

Yours sincerely,

Referee #1:

In the current manuscript Uchida and colleagues investigated the expression of RNA-binding proteins (RBPs) in mesenchymal-like cells within a triple negative breast cancer (TNBC) patient-derived xenograft (PDX) using single-cell RNA-Sequencing (scRNA-Seq). Intersection of these data with additional published scRNA-Seq data identified a total of 240 genes commonly expressed in the mesenchymal-like population characterized by the expression of CD44, NRP1, ZEB1 and low levels of CD24. This population is thought to contain cancer stem cells (CSCs), which are responsible for tumor initiation and therapy resistance. In order to treat this sub-population of TNBC more efficiently the authors suggest to target RBPs that are enriched in CSCs. Thus, the authors turned to ZCCHC24, which was among the 240 enriched genes and one out of 17 genes annotated as "RNA binding". The authors performed a broad range of cellular and molecular in vitro and in vivo experiments and ultimately propose a function of ZCCHC24 in regulating the mRNA stability of a selected set of genes, including ZEB1, thereby regulating tumorigenicity and treatment resistance.

Overall, the manuscript follows a logical flow and contains a large amount of data. However, some of the described results are not really convincing due to the small effect sizes and the biased data presentation. Also, important regulatory feedback loops are suggested, yet not validated at the protein level.

These are my concerns:

Major:

1) All loss-of-function analyses were performed with one siRNA only. Given the well-known off-target effects of si/shRNAs the authors need to include a second, independent siRNA and repeat key experiments.

2) The authors identified 17 putative/known RBPs (Figure 1D) that are enriched in the mesenchymal-like population. Why was ZCCHC24 selected? Please explain!

3) The knockdown of ZCCHC24 affects several genes, yet the authors only highlighted a small amount of hand-selected ones (Figure 2). Moreover, the selected genes (ZEB1, CD44, NRP1 and NOTCH2), whose ZCCHC24-dependent expression was validated via qPCR also in HCC38 (Figure 2B), show only minor expression changes, some were not even significant. This raises severe doubts about the validity and relevance of the suggested function of ZCCHC24!

A more unbiased data analysis should have been performed! For example, what pathways and/or gene sets are affected by ZCCHC24? What are the top de-regulated genes? What genes are commonly deregulated in both MDA-MB-231 and PDX1? Does the overexpression of ZCCHC24 lead to opposite expression changes?

4) In line with the statement above, the authors performed BRIC-Seq in order to identify transcripts that are altered in their stability upon ZCCHC24 depletion (Figure 2C). The most interesting targets would lie far away from the centreline of the plot, i.e.

transcripts with a long half-life in siZCCHC24, but a short half-life in siNC (upper left quadrant) as well as long half-life in siNC and short half-life in siZCCHC24 (lower right quadrant). Instead, the authors focused their attention on transcripts whose half-lives were nearly unchanged (close to the centreline). Moreover, the RNA stability in the BRIC-Seq experiment was analyzed after 0 h, 1 h and 2 h, whereas the Actinomycin D validation experiment analyzed expression levels of selected mRNAs after 0 h, 4 h and 8 h (Figure 2E). Why was the BRIC-Seq experiment stopped shortly after the BrdU washout and why is only the 1 h time point of the BRIC-Seq shown? What is the identity of the other transcripts that display more significant/relevant differences between siZCCHC24 and siNC? Why does NOTCH2 level increase after Actinomycin D treatment (Fig. 2E)? Furthermore, the authors need to explain why the decay functions in Figure 2E are linear.

5) PAR-CLIP was performed to enrich for direct RNA targets of ZCCHC24 using FLAG-tagged and overexpressed ZCCHC24 in MDA-MB-231 cells and a consensus binding motif was identified (Figure 3A-D). However, the PAR-CLIP experiment is poorly documented. Please show specific enrichment of the transgene and the protein gel with the area that was cut out for RNA isolation.

Only two targets (NRP1 and CD44) were validated via luciferase reporter assay (Figure 3G, H) in ZCCHC24 overexpressing 293FT cells and the most important target, ZEB1, was not included in this analysis. This needs to be done. Also, RIP experiments should be performed, ideally pulling down endogenous ZCCHC24, and enrichment of selected target transcripts (as well as negative controls) should be analyzed via RT-qPCR.

6) Moreover, the authors need to validate the regulation of endogenous targets in relevant cell lines after gain-/loss-of-function of ZCCHC24. Here, western blot analyses need to be shown, especially for ZEB1. In addition, the regulation of endogenous ZCCHC24 on protein level by ZEB1 needs to be shown.

7) In Figure 4G the authors claim to provide proof of co-expression of ZCCHC24 and ZEB1 in multiple TNBC samples. Multiple equals three and the co-expression is not really visible. The authors might want to perform a TMA analysis to perform a more relevant and robust analysis.

8) The authors show that depletion of YAP and JUN also affects ZCCHC24 transcript abundance (Figure 4B). Are these factors also part of a feedback loop like it is suggested for ZEB1?

9) In Figure 6 the authors present JQ1 in vivo treatment data in combination with ZCCHC24 depletion. The biological effect is not very impressive and the experiment per se not really relevant, as JQ1 is not used to treat patients. While the rationale of this experiment was explained in the text, the clinical relevance is missing. It would have been more relevant to combine clinically approved drugs / drug combinations (see e.g. cited studies by Haltzis et al., JAMA, 2011; and Loibl et al., Lancet Oncology, 2018) with depletion of ZCCHC24 in order to support the concept shown in Figure 6B.

Minor:

a) For multiple group comparisons ANOVA has to be used, not T-Test (e.g. Fig.2B; Fig.3G)

b) In Figure 2A, right graph: x-axis label should be corrected ("-10" should be "-1")

c) The scheme in Figure 4F is not supported by data, as the authors did not investigate translation rates and protein levels.

d) Line 157: "high relevance" should be replaced by "strong correlation". Is the correlation stronger in TNBC samples only, i.e. is it subtype-dependent?

e) Figure 5B: p value in the table should be corrected (1.20×10^{-6})

f) Please specify the lentiviral plasmids used for cloning and overexpression of GFP and ZCCHC24.

Referee #2:

In this manuscript, Yutaro Uchida et al identified the RNA binding protein ZCCHC24 in the mesenchymal fraction of TNBC, demonstrated that it stabilizes mRNA for several genes including ZEB1, YAP, JUN and NOTCH2, and that its knockdown in TNBC cell lines and PDX reduces tumor burden in xenotransplantation assays. The authors employed state-of-the-art analysis and the results are of top quality, interest and importance.

My only concern is whether ZCCHC24 targets CSC per se as suggested by the authors or the mesenchymal-like state. Unless the authors show that CSC population (defined by flow cytometry - e.g. markers below) is directly targeted - could the effects in Figs.5-6 be due to oncogene/non-oncogene addiction that curtails the growth of the mesenchymal population? This should either be addressed experimentally - better explained - - or the text may be modified from "targeting CSC" to "targeting the

mesenchymal state".

Minor issues:

The authors should mention in the introduction the different TNBC subtypes including basal-like and mesenchymal like lesions and indicate for each cell line used in this study, which subtype it represents -e.g. mda-mb-231 - mesenchymal; HCC38 basal-like.

What's CDC42? What does it represent? - what's the reference for its use to sort CSC?

Why did the authors use the markers they used rather than known markers for CSC such as CD24, CD44 and CD49f or ALDH?

The reason for selecting ZCCHC24 out of the 17 other RNA binding proteins should be explained /rationalized better vis-à-vis Fig 1E.

Fig 4A - the range scale of 1-50 on the right overlaps with the TAS pyramid - please move it a bit

Fig 4G - very difficult to see the details. Perhaps show a smaller area at much higher magnification.

Fig 5 - is this siRNA (transient) or shRNA (stable) knockdown? Are these lenti-shRNA clones? The clones should be characterized - e.g. showing the level of ZCCHC24 knocked-down.

Referee #3:

In this manuscript, Uchida et al. identified the RNA-binding protein ZCCHC24 sustaining the expression of important tumor-supportive factors that potentiate the aggressive behavior of triple-negative breast cancer. Particularly, the authors uncovered that ZCCHC24 can bind to the 3'UTR of mRNAs encoding for several mesenchymal- and stemness-associated factors, thereby maintaining their RNA stability half-life and basal protein expression. Moreover, among the ZCCHC24-targets, e.g ZEB1, authors demonstrated that it could exert a positive gene-activating role in cooperation with YAP and JUN on ZCCHC24 gene expression, thereby, highlighting a so far uncharacterized positive feedback circuit that can support the aggressive features of TNBC. To draw this novel conclusion, authors leveraged single-cell RNA-seq approaches applied on TNBC cell lines and PDXs, BRIC-seq to uncover the gene targets reliant on ZCCHC24 for their RNA stability, PAR-CLIP to uncover its RNA-binding sites and in vitro functional assays as well as in vivo mouse models to unveil the clinical/therapeutic value of ZCCHC24 in TNBC.

Nevertheless, major concerns have been raised to substantially optimize their work, which should be addressed in the best possible way.

Major concerns:

Figure 1

1) The authors do not explain why they used the expression of CDC42 as a marker for clustering TNBC cells from the PDX of patient #1. What is the rationale for using this marker? I strongly recommend authors use alternative standard cancer stem cell- and mesenchymal-associated markers that better reflect the different subpopulations of epithelial and mesenchymal/dedifferentiated/stem-like cancer cells e.g rather based on a combined gene expression score of high CD44, low CD24/KRT8/KRT19, high ALDH1A1/ALDH1A2/ALDH1A3, high ZEB1 and if possible any additional mesenchymal marker such as VIM, or SNAI1. Two typical examples that the authors' clustering gene selection criteria is insufficient are 1) that the CDC42+ population encompasses a fraction of "mesenchymal" cells and 2) that several cells in the "mesenchymal population" include several CDC42 positive cells, raising doubts regarding the appropriateness of the used genes for clustering. Moreover, it is doubtful if the "mesenchymal population" in their UMAP encompasses cells with high CD44 expression, something that is also not shown here.

Also, did the authors perform any "iteration analysis" that could provide the information for the possible number of cell clusters that best represent the number of existing cell populations in dissected tumors from PDX and MDAMB231? If yes, authors are strongly encouraged to show such an analysis that explains the generation of e.g. 3 clusters in their UMAPs.

And in general, they should provide a better-detailed protocol for their scRNAseq analyses in the method section.

2) Regarding the violin plot in Fig.1E, it is problematic the fact that the authors did not remove cells with 0 counts for their gene of interest. As a result, the shape of the violin plot does not reliably depict the true distribution of cells according to their expression levels of ZCCHC24. Therefore, authors ought to reperform such an analysis on the revised cell clusters according to the previous comment as well as adding a statistic regarding the median expression differences.

Figure 2

3) In Fig.2D, the authors' choice to select several genes whose RNA half-life is marginally changed upon ZCCHC24 loss (very close to the borderline) is rather a weak decision since several other hits (in grey) seem to have a more robust change. Authors

should first highlight what kind of genes do represent these hits and most importantly identify/select at least a couple of these hits and perform key experiments similar to what has been done for ZEB1 in Fig.2 and subsequent figures.

4) Since authors have at their disposal a very nice collection of homemade datasets, why do not they proceed with an overlap between highly ZCCHC24-dependent genes (based on RNAseq) with strong hits from the BRICseq to identify the most promising targets? That would provide a solid basis for selecting the right key genes whose expression is reliant on ZCCHC24 and pivotal for the aggressiveness of TNBC, at least this approach is better than just "cherry-picking" some genes from Fig.2D.

5) Probably one of the most "interesting" concerns is that authors utilized one single siRNA against ZCCHC24 throughout their whole work, where mentioned. "Interesting" because the use of one siRNA is highly raising the possibility of off-target effects. For this reason, I would strongly suggest them make use of a "smart pool" of different ZCCHC24-specific siRNA in equimolar amounts and perform several important key experiments (similar to Fig. 2B, Fig.2D) to validate their findings from RNAseq and BRICseq and if possible for some in vivo experiments using PDX or MDAMB231 cells (Fig. 5A-B or EV5E) and in vitro functional assays (like in EV5A). For the RT-qPCR, performing a single-siRNA silencing out of the smart pool showing similar effects to the "smart pool" would be highly suggested.

6) If I am not mistaken, in Fig. 2E authors did not mention in the method section to which "housekeeping" gene the expression of the several tested genes was normalized. Also, the use of actinomycin D at certain concentrations may affect the expression of broadly used "housekeeping" genes such as 18S transcribed by RNAPol-I or even more several known like ACTB, RPLP0, GAPDH etc that are transcribed by RNAPol-II. This renders the normalization to any of the abovementioned genes quite inappropriate. Therefore, authors are highly recommended to use an alternative inhibitor such as a-amanitin that can specifically RNAPol-II and -III and preferentially use an RNAPol-I-transcribed gene such as 18S to which they can normalize and in general.

7) In Fig.EV2, effects in FACS are slightly marginal. It is highly recommended to perform a western blot for the same stemness markers in control and ZCCHC24-silenced TNBC cells.

Figure 3

1) The identified binding motif of ZCCHC24 shown in Fig. 3D, is also a well-known poly-A motif that recruits several other cap-proteins of the 3'UTR such as the poly-A binding protein PABP1. How specific can this motif be for ZCCHC24? The fact that authors have designed a reporter construct with the identified binding site of their favorite factor still does not preclude the possibility that mutation of this binding site and subsequently reported activity decay is exclusively explained by the loss of ZCCHC24 binding to it but could be also explained by the other proteins such as PABP1 that share the same binding motif. The setting that authors used of overexpressing ZCCHC24 provides only part of the needed information to ensure dependency of mRNA with the respective binding motif on this factor. Therefore, equally important is to follow a loss-of-function approach for ZCCHC24 (similar to the previous experiments) to address this question. For example, a naïve suggestion would be to perform a silencing of a dominant candidate equally binding to the WT reporter e.g. PABP1 alone or with ZCCHC24 silencing, and perform a luciferase reporter assay. At least silencing of the latter should have an exclusive or a strong additive effect on the reporter activity.

2) What is the cumulate fraction represented in Fig.3E? As it will be hard to interpret this graph for several readers, authors are highly recommended to draw a violin plot with the different groups of genes depending on the number of ZCCHC24-binding sites that bear and subsequently use the log2FC as a variable on the y-axis. This should visually better demonstrate the differences in gene expression upon ZCCHC24 loss in genes with >10 binding sites. Also, does this latter gene group with >10 binding sites encompass key genes promoting stemness and a mesenchymal state? Because it seems that ZEB1 and CD44 are not good examples based on the number of motif sites they show in Fig.3F.

Figure 4

1) Authors should perform a ChIP-qPCR for H3K27ac on the identified enhancer region of ZCCHC24 in control and ZEB1-, JUN- and YAP-silenced cells. Also, based on the HiC interaction network, it is not convincing if the TSS of ZCCHC24 is indeed interacting with the identified enhancer region located in the gene body. Possibly, authors should seek some additional enhancer regions based on your HiC analyses and perform a similar ChIP-qPCR.

2) Authors should quantify their immunofluorescence stainings and make a cell-based correlation analysis for ZCCHC24 and ZEB1 staining. Also, it is highly recommended to show zoom-in perspectives of their stainings.

Figure 5

1) Authors performed a siRNA-mediated loss of ZCCHC24, therefore they should explain if they used a trypan-blue exclusion approach to surely inject only viable cells in Fig.5A-B and Fig.EV5E. Also, after 3 weeks of subcutaneous injection, most probably the silencing effect vanishes. However, the ZCCHC24-downstream effect could be still not reversed thereby still bearing a minor expression of e.g. ZEB1, JUN, etc and ultimately demonstrating a high apoptotic rate. Therefore, authors should consider performing an IHC or IF for the abovementioned markers or at least making use of their scRNAseq data and addressing these concerns.

2) If authors have a knockout clone of ZCCHC24, authors should consider performing a similar in vivo experiment.

3) Also, since chemotherapy is a standard-of-care therapeutic approach in TNBC, authors should consider introducing e.g. a platin-based chemotherapeutic and check for a stronger sensitiveness in ZCCHC24-silenced cells. Also, important is to corroborate their findings with publicly available patient-derived transcriptomic and follow-up data.

Figure 6

1) Authors made an interesting observation where ZCCHC24-deficient tumors were sensitized to JQ1 treatment. Authors may consider examining if ZCCHC24-deficient tumors presented an augmented apoptotic rate and more decreased expression of ZEB1 in their dissected tumors.

Minor concerns:

- 1) Statistics and used thresholds are missing in Fig. 2D-E
- 2) Authors should explain why NOTCH2 stability increased with time upon actinomycin D treatment (Fig.2E).
- 3) In Fig.3C, the y-axis does not provide the unit for bamcoverage (is it RPKM?). Also, what is the scale for the x-axis? Did they perform a "scale-region" aggregate profiling for the binding of ZCCHC24? They should mention this in the legend or/and methods.
- 4) Fig.3H: If the mutant reporter impedes the binding of ZCCHC24 on it, authors should better normalize the differences to the WT-NC condition and not normalizing to the internal control each time. The same applies to Fig.4C-D and normalizing to the control.
- 5) Authors should explain why the track with ZCCHC24 binding sites, at least the summits of those does not align with the motif sites identified by PAR-CLIP. Also, the peak site track with the peak track (bigwig file I guess) of NRP1 is not matching to each other and should be corrected.
- 6) Statistic for Fig.4E.
- 7) In the graphical abstract (Fig.6B), authors should correct the "total kill" and "chemotherapy" as JQ1 is not a chemotherapy and most importantly, based on their in vivo studies, combination of ZCCHC24 and JQ1 does not "totally" reduce the tumor burden of mice.
- 8) The molecular weight of proteins in western blots is missing and should be added.

Summary:

The conception and findings are of enlightening significance for TNBC therapy, especially given the lack of targeted therapies in this particularly aggressive breast cancer subtype. Hence, I posit this work as highly relevant for the respected readership of EMBO Reports.

Referee #1:

In the current manuscript Uchida and colleagues investigated the expression of RNA-binding proteins (RBPs) in mesenchymal-like cells within a triple negative breast cancer (TNBC) patient-derived xenograft (PDX) using single-cell RNA-Sequencing (scRNA-Seq). Intersection of these data with additional published scRNA-Seq data identified a total of 240 genes commonly expressed in the mesenchymal-like population characterized by the expression of CD44, NRPI, ZEB1 and low levels of CD24. This population is thought to contain cancer stem cells (CSCs), population of TNBC more efficiently the authors suggest to target RBPs that are enriched in CSCs. Thus, the authors turned to ZCCHC24, which was among the 240 enriched genes and one out of 17 genes annotated as "RNA binding". The authors performed a broad range of cellular and molecular *in vitro* and *in vivo* experiments and ultimately propose a function of ZCCHC24 in regulating the mRNA stability of a selected set of genes, including ZEB1, thereby regulating tumorigenicity and treatment resistance. Overall, the manuscript follows a logical flow and contains a large amount of data. However, some of the described results are not really convincing due to the small effect sizes and the biased data presentation. Also, important regulatory feedback loops are suggested, yet not validated at the protein level.

We appreciated your kindful evaluation for our manuscript and constructive advices.

Major:

1) All loss-of-function analyses were performed with one siRNA only. Given the well-known off-target effects of si/shRNAs the authors need to include a second, independent siRNA and repeat key experiments.

Thank you for your comments. Using another siRNA targeting ZCCHC24, we performed qPCR, reporter assays, and *in vitro* cell viability assays. qPCR analysis showed that the expression of the target genes of ZCCHC24 (*ZEB1*, *CD44*, *NRPI*, *NOTCH2*, and *ADAMTS1*) was similarly downregulated by siZCCHC24_2 (Appendix Fig 3). In addition, we performed reporter assays with siRNAs against ZCCHC24 for the reporter with 3'UTR of *ZEB1* and confirmed the decrease in reporter activities with two types of siRNAs against ZCCHC24 (Fig 3H). Finally, through the dual use of siRNA against ZCCHC24 and Doxorubicin, we confirmed the additive effects compared to the single use of the two types of siRNA (Fig EV5D).

Moreover, using another siRNA against ZEB1, JUN, and YAP1, we performed qPCR analysis and confirmed a decrease in the mRNAs expression of *ZCCHC24* (Appendix Fig 8C-D).

Based on these results, we changed the manuscript as follows.

Line 108-112

Quantitative PCR analyses of MDAMB231 cells, the claudin-low TNBC cell line HCC38, and PDX knockdown with siRNA against *ZCCHC24* also showed the downregulation of *CD44*, *NRPI*, *ZEB1*, *NOTCH2*, and *ADAMTS1* expression whereas overexpression of *ZCCHC24* in MDAMB231 cells increased the expression of *NOTCH2*, *ZEB1*, *CDH11*, and *ZEB1* (Fig 2B, Fig EV1A, Appendix Fig 3).

Line 154-156

We also performed a reporter assay using 3'UTR of *ZEB1* with knockdown of endogenous *ZCCHC24* of MDAMB231 and showed that suppression of *ZCCHC24* expression decreased reporter activities (Fig 3H).

Line 174-177

Furthermore, knockdown of ZEB1, YAP1, or JUN in MDAMB231 cells and PDX downregulated ZCCHC24 expression, whereas overexpression of ZEB1 upregulated ZCCHC24 expression, as evaluated through quantitative PCR (Fig 4B, Fig EV3A, Appendix Fig 8).

Line 226-230

To further evaluate this, we explored doxorubicin, which is used as a first-line compound in chemotherapy for TNBC. The addition of Doxorubicin to MDAMB231 or PDX cells increased the expression of ZCCHC24 (Fig EV5B). Furthermore, the dual use of doxorubicin with siRNA against ZCCHC24 had additive effects on suppressing cell viability compared to a single treatment (Fig EV5C-D).

2) The authors identified 17 putative/known RBPs (Figure 1D) that are enriched in the mesenchymal-like population. Why was ZCCHC24 selected? Please explain!

Thank you for your comments. Among the 17 “RNA-binding” genes enriched in the mesenchymal-like population, some RBPs such as RBFOX2 and NOVA1 are known for their roles in mesenchymal cells, and VIM, COL14A1, and DCN are known for their roles as extracellular matrices rather than as RNA-binding proteins. Among these genes, we chose ZCCHC24 as an RBP whose roles are unknown. Accordingly, we have revised the manuscript as follows:

Line 90-95

Among these 17 “RNA binding” genes, *RBFOX2* (Braeutigam *et al*, 2014; Maurin *et al*, 2023) and *NOVA1* (Qu *et al*, 2022) have been reported to be essential for the maintenance of cancer stemness and are known for their roles in epithelial-mesenchymal transition (EMT). In addition, *VIM*, *DCN*, and *COL14A1* are known for their roles as extracellular matrices, rather than as RBPs. Notably, the function of ZCCHC24 has not been well characterized, and its function in cancer remains unknown.

3) The knockdown of ZCCHC24 affects several genes, yet the authors only highlighted a small amount of hand-selected ones (Figure 2). Moreover, the selected genes (ZEB1, CD44, NRP1 and NOTCH2), whose ZCCHC24-dependent expression was validated via qPCR also in HCC38 (Figure 2B), show only minor expression changes, some were not even significant. This raises severe doubts about the validity and relevance of the suggested function of ZCCHC24!

A more unbiased data analysis should have been performed! For example, what pathways and/or gene sets are affected by ZCCHC24? What are the top de-regulated genes? What genes are commonly deregulated in both MDA-MB-231 and PDX1? Does the overexpression of ZCCHC24 lead to opposite expression changes?

Thank you for your comments. Following your comments, we have identified 124 common genes between the differently expressed genes (DEGs) of RNA-Seq for MDAMB231 cells and destabilized transcripts in BRIC-Seq, as shown in Appendix Table 3. Among these genes, those important for cancer stemness, such as *CD44*, *ZEB1*, and *NOTCH2*, were included. Moreover, we also performed gene ontology analysis for these 124 genes, and showed that genes with function of “Negative regulation of apoptotic process,” “Cell matrix adhesion,” “Cell migration” are specifically enriched (Fig EV1D). These functions are essential for the stemness, invasion, and survival of breast cancer cells. As representative genes characterizing these functions, *ADAMTS1*, *CDH11* and *JAG1* are identified.

As common genes were downregulated in both PDX and MDAMB231 cells, 376 genes, represented by *ADAMTS1*, *ACVR1*, *CDH11*, and *CD44* were identified and are listed in Appendix Table 2. These genes are considered the main targets of ZCCHC24 in TNBC. Regarding the scRNA-seq data for PDX, *ACVR1*, *ADAMTS1*, *CDH11*, and *JAG1* were relatively highly expressed in the mesenchymal-like population, and knockdown of ZCCHC24 *in vivo* decreased these genes in the mesenchymal population (Appendix Fig 1,

Fig 5F). These results demonstrated the important roles of ZCCHC24 in maintaining the expression of these genes. Moreover, we showed in the Actinomycin D test that knockdown of ZCCHC24 decreased the stability of the mRNAs of these genes (Appendix Fig 5).

In addition, the overexpression of ZCCHC24 led to the upregulation of *CDH11*, *JAG1*, *ZEB1* and *NOTCH2* expression (Fig EV1A). These results support our hypothesis that ZCCHC24 is essential for the maintenance of breast cancer stemness and tumor invasion/survival without bias.

Following these results, we changed the manuscript as follows.

Line 104-112

This revealed that expressions of a set of genes critical for the characterization of CSC populations, such as *CD44*, *NRP1*, *ZEB1*, and *NOTCH2*, and a set of genes important for cancer progression and invasion, such as *JAG1*, *ADAMTS1*, *ACVR1* were specifically and significantly downregulated (Fig 2A, Appendix Table 2). Quantitative PCR analyses of MDAMB231 cells, the claudin-low TNBC cell line HCC38, and PDX knockdown with siRNA against ZCCHC24 also showed the downregulation of *CD44*, *NRP1*, *ZEB1*, *NOTCH2*, and *ADAMTS1* expression whereas overexpression of ZCCHC24 in MDAMB231 cells increased the expression of *NOTCH2*, *ZEB1*, *CDH11*, and *ZEB1* (Fig 2B, Fig EV1A, Appendix Fig 3).

Line 123-134

We observed reduced stability of mRNAs derived from genes involved in tumorigenicity and treatment resistance (Fig 2D), including *NOTCH2*, *CD44*, *ZEB1*, *CDH11*, and *ADAMTS1*. Based on the RNA-Seq results, 124 genes were identified as differently expressed genes with destabilized transcripts (Appendix Table 3). Gene ontology analysis of these 124 genes with DAVID showed that genes function with “Negative regulation of apoptotic process,” “Cell matrix adhesion,” “Cell migration,” and “cell-cell adhesion,” which are important for cancer stemness were regulated with ZCCHC24 (Fig EV1D). To confirm these results, we analyzed the effect of Actinomycin D addition on RNA stability after transcription was stopped. The results showed that knockdown of ZCCHC24 decreased the stability of mRNAs derived from genes important for tumorigenicity, tumor progression, and treatment resistance, such as *ZEB1*, *NRP1*, *CD44*, *NOTCH2*, *CDH11*, *ADAMTS1* and *JAG1* (Fig 2E, Appendix Fig 5).

4) In line with the statement above, the authors performed BRIC-Seq in order to identify transcripts that are altered in their stability upon ZCCHC24 depletion (Figure 2C). The most interesting targets would lie far away from the centreline of the plot, i.e. transcripts with a long half-life in siZCCHC24, but a short half-life in siNC (upper left quadrant) as well as long half-life in siNC and short half-life in siZCCHC24 (lower right quadrant). Instead, the authors focused their attention on transcripts whose half-lives were nearly unchanged (close to the centreline). Moreover, the RNA stability in the BRIC-Seq experiment was analyzed after 0 h, 1 h and 2 h, whereas the Actinomycin D validation experiment analyzed expression levels of selected mRNAs after 0 h, 4 h and 8 h (Figure 2E). Why was the BRIC-Seq experiment stopped shortly after the BrdU washout and why is only the 1 h time point of the BRIC-Seq shown? What is the identity of the other transcripts that display more significant/relevant differences between siZCCHC24 and siNC? Why does NOTCH2 level increase after Actinomycin D treatment (Fig. 2E)? Furthermore, the authors need to explain why the decay functions in Figure 2E are linear.

Thank you for your valuable comments. In BRIC-Seq, we immunoprecipitated nucleic acid analogs to evaluate the stability of mRNAs, while we stopped the transcription of the cells to evaluate mRNA stability using Actinomycin D assays. Owing to the differences in the methods for the evaluation of mRNAs half-life detection and drug toxicity, it is highly possible that the half-lives of transcripts are different between these two tests. Due to these backgrounds, we concluded that while it is reasonable to compare the relative half-lives within the same test, it is difficult to compare the half-lives of transcripts between different tests for the measurement of transcript half-lives. We note this as a limitation of this discussion.

As shown in Fig 5E, *NOTCH2* levels increased after Actinomycin D treatment. This was due to the slower decay of the mRNAs of *NOTCH2* than that of *PSMB2* as the internal control gene. As you suggested, we concluded that this was an error in our elucidation. Therefore, we have corrected the evaluation as follows: we calculated the relative remaining mRNA percentages for samples with siZCCHC24 compared to samples with siNC at each time point (Fig 2E, Appendix Fig 5).

In our analysis, the decay function appeared to be linear for some genes. Ideally, the decay should be exponential. Because we analyzed only two time points, some genes seemed to degrade linearly. For the exact measurement of the half-life of transcripts, obtaining more time points to measure transcript degradation is needed. We noted this limitation in the Discussion.

Following them, we changed the manuscript as follows.

Line 324-330

As a limitation, the half-lives of the mRNAs were different between the BRIC-Seq and Actinomycin D tests. Owing to the differences in the methods for the evaluation of mRNAs half-life detection and drug toxicity, it is highly possible that the measured half-lives are different between these two tests. Moreover, for some genes, the degradation of transcripts appears to be linear in Actinomycin D assays, whereas the decay of mRNAs is ideally exponential. Measuring the degradation of transcripts at more time points would be helpful for the exact calculation of mRNAs half-life.

5) PAR-CLIP was performed to enrich for direct RNA targets of ZCCHC24 using FLAG-tagged and overexpressed ZCCHC24 in MDA-MB-231 cells and a consensus binding motif was identified (Figure 3A-D). However, the PAR-CLIP experiment is poorly documented. Please show specific enrichment of the transgene and the protein gel with the area that was cut out for RNA isolation. Only two targets (NRP1 and CD44) were validated via luciferase reporter assay (Figure 3G, H) in ZCCHC24 overexpressing 293FT cells and the most important target, ZEB1, was not included in this analysis. This needs to be done. Also, RIP experiments should be performed, ideally pulling down endogenous ZCCHC24, and enrichment of selected target transcripts (as well as negative controls) should be analyzed via RT-qPCR.

Thank you for your comments. As the molecular weight of the ZCCHC24 protein is approximately 27kDa, we extracted a 37-75kDa area of the membrane, as shown in Appendix Fig 6. We have also added a description of PAR-CLIP in the Materials and Methods section.

We also performed reporter assay for the reporter with ZEB1 3'UTR, and showed that the overexpression of ZCCHC24 upregulated the reporter activity (Fig 3G). Moreover, we also showed that the knockdown of the endogenous ZCCHC24 of MDAMB231 decreased the activity of the reporter with ZEB1 3'UTR (Fig 3H).

For the immunoprecipitation of endogenous ZCCHC24, we tagged the HiBiT tag on the C-terminus of ZCCHC24 in the MDAMB231 cell line and created a 231-KI cell line (Appendix Fig 9). Although this cell line was effective for the detection of endogenous ZCCHC24, we could not immunoprecipitate endogenous ZCCHC24 because we did not have suitable antibodies for immunoprecipitation. We also performed immunoprecipitation using a polyclonal antibody against ZCCHC24; however, this did not work.

As an alternative experiment, we performed RNA-IP (RIP) assays on MDAMB231 cells overexpressing ZCCHC24 or GFP as a negative control. ZEB1 and NOTCH2 mRNA levels were enriched in ZCCHC24 overexpressing cells compared in GFP-expressing cells. Furthermore, as a negative transcript, ACTB was not enriched in ZCCHC24 expressing cells (Fig EV2).

Following these results and backgrounds, we changed the manuscript as follows.

Line 147-156

We also confirmed the binding of ZCCHC24 to mRNAs of *ZEB1* and *NOTCH2* using the RNA-immunoprecipitation (RIP) assay, while mRNAs of ACTB, used as negative control transcripts, were not

enriched in *ZCCHC24* expressing cells (Fig EV2). We next performed reporter assays using the 3'UTRs of *CD44*, *NRP1*, and *ZEB1* to evaluate whether these binding sites function as *cis*-elements to regulate target genes upon *ZCCHC24* binding. The results showed that *ZCCHC24* expression led to the upregulation of reporter activity (Fig 3G, Fig EV3A). We also performed reporter assay using 3'UTR of *ZEB1* with knockdown of endogenous *ZCCHC24* of MDAMB231, and showed that suppression of *ZCCHC24* expression decreased reporter activities (Fig 3H).

Line 178-182

For the detection of endogenous *ZCCHC24* protein on MDAMB231, we tagged the HiBiT tag, which is a small part of the nano-luciferase, on the C-terminus of *ZCCHC24* of MDAMB231 (231-KI) for easy detection of endogenous *ZCCHC24* protein expression (Appendix Fig 9A) (Uchida *et al.*, 2021). We confirmed its validity by western blotting and luminescence of nano-luciferase (Appendix Fig 9B-C).

Line 331-335

In addition, the identification of target transcripts of endogenous *ZCCHC24* by immunoprecipitation of endogenous *ZCCHC24* is important for elucidating the roles of *ZCCHC24*. However, we did not have suitable antibodies for the immunoprecipitation of endogenous *ZCCHC24* and could not perform the experiment. The identification of the endogenous targets of *ZCCHC24* is an important subject for future studies.

Line 559-564

The resolved proteins were transferred onto nitrocellulose membrane (0.2µm). The region with the target protein on the blot (30kDa-75kDa) was cut and RNA was extracted using Proteinase K (NEB), acid phenol/chloroform/isoamyl alcohol (Nippon Gene, Tokyo, Japan), and Quick-RNA Miniprep kit (Zymo Research, CA, USA). The purified RNA was subjected to reverse transcription using the TGIRT-III enzyme (InGex, MO, USA).

6) Moreover, the authors need to validate the regulation of endogenous targets in relevant cell lines after gain-/loss-of-function of *ZCCHC24*. Here, western blot analyses need to be shown, especially for *ZEB1*. In addition, the regulation of endogenous *ZCCHC24* on protein level by *ZEB1* needs to be shown.

Thank you for the suggestion. We performed western blotting for MDAMB231 knockdown of *ZCCHC24* with siRNA and confirmed a decrease in *ZEB1*, *NOTCH2*, *CD44*, and *NRP1* protein levels (Fig EV1C). In addition, we also confirmed that the overexpression of *ZCCHC24* led to the upregulation of *ZEB1* (Fig EV1A).

Next, for MDAMB231 and PDX, we knocked down *ZEB1* and performed western blotting for *ZCCHC24*. We confirmed the downregulation of *ZCCHC24* expression by knocking down *ZEB1* (Fig EV3B). In contrast, when we overexpressed *ZEB1*, the expression of *ZCCHC24* was upregulated (Fig EV3A).

Following these results, we changed the manuscript as follows.

Line 108-112

Quantitative PCR analyses of MDAMB231 cells, the claudin-low TNBC cell line HCC38, and PDX knockdown with siRNA against *ZCCHC24* also showed the downregulation of *CD44*, *NRP1*, *ZEB1*, *NOTCH2*, and *ADAMTS1* expression whereas overexpression of *ZCCHC24* in MDAMB231 cells increased the expression of *NOTCH2*, *ZEB1*, *CDH11*, and *ZEB1* (Fig 2B, Fig EV1A, Appendix Fig 3).

Line 115-116

Downregulation of target genes of *ZCCHC24* was also confirmed at the protein level with the western blotting (Fig EV1C).

Line 174-185

Furthermore, knockdown of ZEB1, YAP, or JUN in MDAMB231 cells and PDX downregulated ZCCHC24 expression, whereas overexpression of ZEB1 upregulated ZCCHC24 expression, as evaluated by quantitative PCR (Fig 4B, Fig EV3A, Appendix Fig 8).

For the detection of endogenous ZCCHC24 protein in MDAMB231, we tagged the HiBiT tag, which is a small part of the nano-luciferase, on the C-terminus of ZCCHC24 of MDAMB231 (231-KI) for easy detection of endogenous ZCCHC24 protein expression (Appendix Fig 9A)(Uchida *et al*, 2021). We confirmed its validity by western blotting and luminescence of nano-luciferase (Appendix Fig 9B-C). Western blotting analyses showed that ZCCHC24 was downregulated at the protein level by knockdown of ZEB1 in 231-KI and PDX (Fig EV3B). ZCCHC24 was downregulated at the protein level by knockdown of ZEB1 in MDAMB231 and PDX cells (Fig EV3B).

7) In Figure 4G the authors claim to provide proof of co-expression of ZCCHC24 and ZEB1 in multiple TNBC samples. Multiple equals three and the co-expression is not really visible. The authors might want to perform a TMA analysis to perform a more relevant and robust analysis.

Thank you for the suggestion. Following your request, we performed an immunofluorescence assay for ZCCHC24 and ZEB1 in 10 TNBC pathological specimens. We identified five cases as ZEB1 positive cases and analyzed them. We took pictures with a higher resolution, as shown in Fig. 4G. Moreover, following your suggestion, we attempted to perform a TMA. However, because of the small proportion of ZEB1 positive cells in TNBC specimens, we were unable to perform TMA analysis. We reasoned that this was due to the sensitivity of ZEB1 antibodies and the low expression of transcription factors, and we regarded this question as an important topic for future work. We note this as a limitation of this study. Following these results, we changed the manuscript as follows.

Line 343-346

For pathological analyses, TMA should ideally be performed to objectively evaluate the co-expression of ZCCHC24 and ZEB1. However, because of the small number of ZEB1 positive cells, we could not perform TMA analyses, which is an important topic for future research.

8) The authors show that depletion of YAP and JUN also affects ZCCHC24 transcript abundance (Figure 4B). Are these factors also part of a feedback loop like it is suggested for ZEB1?

Thank you for your comments. We also regarded this as an important question and re-checked the RNA-Seq results. As a result, the expression of YAP1 did not change by ZCCHC24 knockdown. As for JUN, its expression was downregulated on PDX by 0.6 folds and was detected as a differentially expressed gene, while its expression was not changed in MDAMB231 cells. From these results, we concluded that ZEB1 is the main factor in the feedback loop with ZCCHC24, and YAP1 or JUN are not involved, or otherwise may be sub-factors of the feedback loop.

9) In Figure 6 the authors present JQ1 in vivo treatment data in combination with ZCCHC24 depletion. The biological effect is not very impressive and the experiment per se not really relevant, as JQ1 is not used to treat patients. While the rationale of this experiment was explained in the text, the clinical relevance is missing. It would have been more relevant to combine clinically approved drugs / drug combinations (see e.g. cited studies by Haltziz *et al.*, JAMA, 2011; and Loibl *et al.*, Lancet Oncology, 2018) with depletion of ZCCHC24 in order to support the concept shown in Figure 6B.

Thank you for your comments. Following your suggestions, we also performed experiments with doxorubicin, which is clinically used for TNBC chemotherapy. We observed the upregulation of ZCCHC24 in MDAMB231 and PDX cells (Fig EV5B). Next, we transfected doxorubicin siRNA against ZCCHC24 into MDAMB231 and PDX cells and measured cell viability. The dual use of doxorubicin and

siRNAs showed additive effects on cell viability suppression (Fig EV5D). These results suggest the validity of simultaneous targeting of ZCCHC24 for the use of doxorubicin as a chemotherapy against TNBC.

Following these results, we changed the manuscript as follows.

Line 226-230

To further evaluate this, we explored doxorubicin, which is used as a first-line compound in chemotherapy for TNBC. The addition of Doxorubicin to MDAMB231 or PDX cells increased the expression of ZCCHC24 (Fig EV5B). Furthermore, the dual use of doxorubicin with siRNA against ZCCHC24 had additive effects on suppressing cell viability compared to a single treatment (Fig EV5C-D).

Minor:

a) For multiple group comparisons ANOVA has to be used, not T-Test (e.g. Fig.2B; Fig.3G)

Thank you for your comments. Following your suggestions, we re-analyzed the data using ANOVA and Tukey's test (Fig 3H, 3I, 4C, 4D, EV3C EV5C-D, Appendix Fig 9C) as post-hoc tests for multiple group comparisons. As for Fig 2B and Fig 3G, we regarded the qPCR analyses and reporter assays for each gene as independent experiments and did not consider them as repeated tests. Therefore, the *t*-test was used to test for significance. We believe that the explanation of the statistical methods used in the experiment was insufficient, which caused this misunderstanding. Thanks to your suggestions, we have added the explanation for all statistical methods used in the legend as follows.

Legend for Fig 2

B, qPCR analyses of the TNBC cell lines MDAMB231 and HCC38 or patient-derived xenografts (Patient #1) knocked down with ZCCHC24 siRNA. Significance of differences of gene expression between siNC and siZCCHC24_1 was tested using an unpaired t-test for independent experiments. (N=3, biological replicates each, * $p < 0.05$, ** $p < 0.01$, *** $p < 0.005$)

Legend for Fig 3

G, Luciferase reporter assay for reporter containing 3'UTR of NRP1, CD44 and ZEB1 upon expression of ZCCHC24 or empty vector as control. Differences of the reporter activity were tested using the unpaired t-test as an independent test for each reporter. (N = 4, biological replicates each, ** $p < 0.01$, *** $p < 0.005$)

b) In Figure 2A, right graph: x-axis label should be corrected ("-10" should be "-1")

Thank you very much for your comments. We changed the label of Fig 2A.

c) The scheme in Figure 4F is not supported by data, as the authors did not investigate translation rates and protein levels.

Thank you for your comments. Following your suggestion, we reached the same conclusion at the protein level using western blotting analyses. However, as you mentioned, we did not evaluate translation rates in this study, although we did show mRNA stabilization in the presence of ZCCHC24. Therefore, we changed the terms of "translation" in Fig 4F for "mRNA stabilization."

d) Line 157: "high relevance" should be replaced by "strong correlation". Is the correlation stronger in

TNBC samples only, i.e. is it subtype-dependent?

Thank you for your comments. In the current analyses, we utilized data from TCGA-BRCA, but we did not perform the analysis with the classification of each subtype, which should be an important question for future work. We have also changed the term “high relevance” to “strong correlation.” Following them, we changed the manuscript as follows.

Line 191-193

Supporting these results, the RNA expression levels of ZEB1 and ZCCHC24 in the Cancer Genome Atlas Breast Invasive Carcinoma (TCGA-BRCA) dataset (1178 samples) were **strongly correlated (Fig 4E)**.

Line 346-348

In addition, we did not analyze the TCGA-BRCA sample separately with each subtype, and whether the strong correlation is specific to TNBC or general in all breast cancer subtypes is still unknown, and also the critical issues to be analyzed.

e) Figure 5B: *p* value in the table should be corrected (1.20×10^{-6})

Thank you for your comments. We changed *p* value as 1.20×10^{-6} .

f) Please specify the lentiviral plasmids used for cloning and overexpression of GFP and ZCCHC24.

We appreciate your comments. Following your suggestions, we have shown the sequences of the lentiviral plasmids in Appendix Table 5.

Referee #2:

In this manuscript, Yutaro Uchida et al identified the RNA binding protein ZCCHC24 in the mesenchymal fraction of TNBC, demonstrated that it stabilizes mRNA for several genes including ZEB1, YAP, JUN and NOTCH2, and that its knockdown in TNBC cell lines and PDX reduces tumor burden in xenotransplantation assays. The authors employed state-of-the-art analysis and the results are of top quality, interest and importance.

Thank you very much for your evaluation.

My only concern is whether ZCCHC24 targets CSC per se as suggested by the authors or the mesenchymal-like state. Unless the authors show that CSC population (defined by flow cytometry - e.g. markers below) is directly targeted - could the effects in Figs.5-6 be due to oncogene/non-oncogene addiction that curtails the growth of the mesenchymal population? This should either be addressed experimentally - better explained - - or the text may be modified from "targeting CSC" to "targeting the mesenchymal state".

Thank you for your comments. As you mentioned, we supposed that targeting ZCCHC24 decreased the mesenchymal state population, including CSCs, thereby reducing its function as a CSC. In our opinion, this mechanism leads to the loss of tumorigenicity and drug resistance. Following your suggestion, we modified several parts of the manuscript from “targeting CSC” to “targeting the mesenchymal state including CSC” as follows;

Line 300-303

The advantages of targeting ZCCHC24 in **mesenchymal like cells with CSCs** include not only the direct inhibition of the expression of a set of genes related to cancer stemness and EMT, including SNAI and NOTCH, but also the disruption of an auto-amplified gene expression network formed by ZCCHC24 and ZEB1 to maintain CSC traits.

Line 314-320

Our study demonstrated that combining therapies targeting **the mesenchymal state including CSCs** with standard treatments such as conventional cytotoxic therapy leads to an enhanced anti-tumor effect. ZCCHC24 knockdown specifically reduces the mesenchymal like population including CSCs, allowing for the targeting of proliferative cell populations with **molecular target therapy** while using ZCCHC24-targeting siRNA in combination, **thereby enhancing the effects of molecular target therapy *in vivo* (Fig 6B)**.

Minor issues:

The authors should mention in the introduction the different TNBC subtypes including basal-like and mesenchymal like lesions and indicate for each cell line used in this study, which subtype it represents - e.g. mda-mb-231 - mesenchymal; HCC38 basal-like.

Thank you for your comments. MDAMB231 is a mesenchymal-like basal TNBC cell line and HCC38 is Claudin-low TNBC cell line. Accordingly, we revised the manuscript as follows:

Line 102-104

First, to examine the function of ZCCHC24 in regulating gene expression in TNBC-related cells, we knocked down ZCCHC24 with siRNA in the **mesenchymal-like basal** TNBC cell line MDAMB231 or PDX and performed RNA-Seq.

Line 108-112

Quantitative PCR analyses for MDAMB231, the **claudin-low** TNBC cell line HCC38 and PDX knockdown with siRNA for *ZCCHC24* also showed downregulation of *CD44*, *NRP1*, *ZEB1*, *NOTCH2*, and *ADAMTS1* while overexpression of *ZCCHC24* in MDAMB231 cells increased the expression of *NOTCH2*, *ZEB1*, *CDH11*, and *ZEB1* (Fig 2B, Fig EV1A, Appendix Fig 3).

What's CDC42? What does it represent? - what's the reference for its use to sort CSC?

Why did the authors use the markers they used rather than known markers for CSC such as CD24, CD44 and CD49f or ALDH?

Thank you for your comments. Following your suggestions, we searched for other genes specifically expressed in the population (Fig 1A-C, Appendix Fig1). As a result, we showed that this population shows high expression of *TGFBI*, *TGFBR1* and *ITGA6*. Also, middle expression of VIM and NRP1 were detected from the analysis. These analyses suggested that the cell population helps tumor EMT with the expression of TGFbeta1.

We created feature plots for *CD24*, *CD44*, *CD49f* (*ITGA6*), and showed plots on Appendix Fig 1 and Fig 1C. *CD44* was expressed in all cells. The expression of *CD24* is relatively specific to epithelial cells. *ITGA6* is highly expressed in TGFBI positive population. As for ALDH1, *ALDH1A3* is specifically expressed in mesenchymal-like cells. In addition, *VIM* or *ZCCHC24* target genes such as *JAG1*, *CDH11*, and *ADAMTS1* showed high or localized expression in the mesenchymal-like population. Based on previous studies by Tominaga *et al.* (PNAS 2019) and Huimin *et al.* (Nat. Commun. 2020), we focused on analyses of NRP1 high and ZEB1 high cell population.

Following these results, we changed the manuscript as follows.

Line 79-83

As a result, PDX was roughly divided into three population: an epithelial-like population characterized by the expression of *CD24*, *KRT8*, and *KRT19*; a mesenchymal like population characterized by the expression of *NRP1*, *ZEB1*, *JUN*, *VIM*, *ALDH1A3*, and low expression of *CD24*; and a **TGFBI** positive population, characterized by the expression of *TGFBI*, *CDC42*, and *ITGA6* (Fig 1A-C, Appendix Fig 1).

The reason for selecting *ZCCHC24* out of the 17 other RNA binding proteins should be explained /rationalized better vis-à-vis Fig 1E.

Thank you for your comments. Among the 17 “RNA-binding” genes enriched in the mesenchymal-like population, some RBPs such as RBFOX2 and NOVA1 are known for their roles in mesenchymal cells, and VIM, COL14A1, and DCN are known for their roles as extracellular matrices rather than as RNA binding proteins. Among these genes, we chose *ZCCHC24* as an RBP whose roles are unknown. Accordingly, we have revised the manuscript as follows:

Line 90-95

Among these 17 “RNA binding” genes, *RBFOX2* (Baeutigam *et al.*, *Oncogene* 2014, Maurin *et al.*, *Nature Communications* 2023) and *NOVA1* (ref.) have been reported to be essential for the maintenance of cancer stemness and are known for their roles in epithelial-mesenchymal transition (EMT). In addition, *VIM*, *DCN*, and *COL14A1* are known for their roles as extracellular matrices, rather than as RBPs. Notably, the function of *ZCCHC24* has not been well characterized, and its function in cancer remains unknown.

Fig 4A - the range scale of 1-50 on the right overlaps with the TAS pyramid - please move it a bit

Thank you for your comments. We moved the range scale from to 1-50 bit, as suggested in Fig4A.

Fig 4G - very difficult to see the details. Perhaps show a smaller area at much higher magnification.

Thank you for the suggestion. Following your request, we performed an immunofluorescence assay for ZCCHC24 and ZEB1 in 10 TNBC pathological specimens. We identified five cases as ZEB1 positive cases and analyzed them. We took pictures with a higher resolution, as shown in Fig 4G.

Fig 5 - is this siRNA (transient) or shRNA (stable) knockdown? Are these lenti-shRNA clones? The clones should be characterized - e.g. showing the level of ZCCHC24 knocked-down.

Thank you for your comments. We utilized special siRNAs (stealth RNA), which showed prolonged effects compared to other normal siRNAs, to knock down ZCCHC24. Nevertheless, it is possible that the effects of the ZCCHC24 knockdown were transient. Therefore, we generated a violin plot for the expression of ZCCHC24 by splitting the samples into siNC and siZCCHC24 as shown in Fig 5F. We showed that the expression of ZCCHC24 decreased and that ZCCHC24 positive cells also decreased. Moreover, the expression of genes downstream of ZCCHC24, such as *ZEB1*, *NRP1*, and *ADAMTS1*, was also decreased (Fig 5F, Appendix Fig 10). Based on these results, we revised the manuscript as follows:

Line 217-219

Moreover, downstream genes of ZCCHC24 such as *ZEB1*, *NRP1*, and *ADAMTS1* were downregulated in this population, indicating the importance of ZCCHC24 for the maintenance of mesenchymal like populations (Fig 5F, Appendix Fig 10).

Referee #3

In this manuscript, Uchida et al. identified the RNA-binding protein ZCCHC24 sustaining the expression of important tumor-supportive factors that potentiate the aggressive behavior of triple-negative breast cancer. Particularly, the authors uncovered that ZCCHC24 can bind to the 3'UTR of mRNAs encoding for several mesenchymal- and stemness-associated factors, thereby maintaining their RNA stability half-life and basal protein expression. Moreover, among the ZCCHC24-targets, e.g ZEB1, authors demonstrated that it could exert a positive gene-activating role in cooperation with YAP and JUN on ZCCHC24 gene expression, thereby, highlighting a so far uncharacterized positive feedback circuit that can support the aggressive features of TNBC. To draw this novel conclusion, authors leveraged single-cell RNA-seq approaches applied on TNBC cell lines and PDXs, BRIC-seq to uncover the gene targets reliant on ZCCHC24 for their RNA stability, PAR-CLIP to uncover its RNA-binding sites and in vitro functional assays as well as in vivo mouse models to unveil the clinical/therapeutic value of ZCCHC24 in TNBC. Nevertheless, major concerns have been raised to substantially optimize their work, which should be addressed in the best possible way.

We appreciated your thorough evaluation for our manuscript and are grateful for your advice.

Major concerns:

Figure 1

1) The authors do not explain why they used the expression of CDC42 as a marker for clustering TNBC cells from the PDX of patient #1. What is the rationale for using this marker? I strongly recommend authors use alternative standard cancer stem cell- and mesenchymal-associated markers that better reflect the different subpopulations of epithelial and mesenchymal/dedifferentiated/stem-like cancer cells e.g rather based on a combined gene expression score of high CD44, low CD24/KRT8/KRT19, high ALDH1A1/ALDH1A2/ALDH1A3, high ZEB1 and if possible any additional mesenchymal marker such as VIM, or SNAI1. Two typical examples that the authors' clustering gene selection criteria is insufficient are 1) that the CDC42+ population encompasses a fraction of "mesenchymal" cells and 2) that several cells in the "mesenchymal population" include several CDC42 positive cells, raising doubts regarding the appropriateness of the used genes for clustering.

Moreover, it is doubtful if the "mesenchymal population" in their UMAP encompasses cells with high CD44 expression, something that is also not shown here.

Also, did the authors perform any "iteration analysis" that could provide the information for the possible number of cell clusters that best represent the number of existing cell populations in dissected tumors from PDX and MDAMB231? If yes, authors are strongly encouraged to show such an analysis that explains the generation of e.g. 3 clusters in their UMAPs.

And in general, they should provide a better-detailed protocol for their scRNAseq analyses in the method section.

Thank you for your comments. Following your suggestions, we searched for other genes specifically expressed in the population (Fig 1A-C, Appendix Fig1). As a result, we showed that this population shows high expression of *TGFB1*, *TGFBRI* and *ITGA6*. Also, middle expression of VIM and NRP1 were detected from the analysis. These analyses suggested that the cell population helps tumor EMT with the expression of TGFbeta1.

We created feature plots for *CD24*, *CD44*, *CD49f* (*ITGA6*), and showed plots on Appendix Fig 1 and Fig 1C. *CD44* was expressed in all cells. The expression of *CD24* is relatively specific to epithelial cells. *ITGA6* is highly expressed in TGFB1 positive population. As for ALDH1, *ALDH1A3* is specifically expressed in mesenchymal-like cells. In addition, *VIM* or ZCCHC24 target genes such as *JAG1*, *CDH11*, and *ADAMTS1* showed high or localized expression in the mesenchymal-like population. Based on previous studies by Tominaga *et al.* (PNAS 2019) and Huimin *et al.* (Nat. Commun. 2020), we focused on analyses of *NRP1* high and *ZEB1* high cell population. We performed "iteration analysis" by changing the resolution of clustering. Among the several conditions, we chose the current resolution as the most

suitable for our analysis. For the precise analysis of scRNAseq, we have added a description in the Methods section.

Following them, we changed the manuscript as follows.

Line 79-83

As a result, PDX were roughly divided into three populations: an epithelial-like population characterized by the expression of *CD24*, *KRT8*, and *KRT19*; a mesenchymal-like population characterized by the expression of *NRP1*, *ZEB1*, *JUN*, *VIM*, *ALDH1A3*, and low expression of *CD24*; and a *TGFBI* positive population, characterized by the expression of *TGFBI*, *CDC42*, and *ITGA6* (Fig 1A-C, Appendix Fig 1).

Line 422-438

Single Cell RNA-Seq analysis

Cell ranger (v. 6.1.1. 10x Genomics) was used to demultiplex samples, process barcodes, and alignment on GRCh38 genome. Individual samples were integrated, expression was normalized, and cell populations were clustered based on the matrix files of gene expression with R v.4.1.2 (R Foundation for Statistical Computing, Vienna, Austria) and the R package Seurat v. 4.0.6 (Butler *et al*, 2018) as described by the developer. Datasets were analyzed by following protocols. Genes expressed in < 3 cells, cells with < 2000 unique molecular identifiers (UMIs) and < 200 genes were removed from the gene expression matrix for each dataset. Furthermore, the data were also filtered with expression of mitochondrial gene contamination in the criteria of < 20%. The data were log-normalized, and the expression of each gene was scaled by regressing the number of UMIs. Gene expression matrix was analyzed with principal component analysis (PCA) and we utilized unsupervised share nearest neighbor (SNN) clustering to the genes with a resolution of 0.025 and divided into 3 clusters. Then they are visualized with Uniform Manifold Approximation and Projection (UMAP). Uniquely expressed genes in each cluster were analyzed with the Seurat FindConservedMarkers function. As the criteria for marker genes, we set threshold as $p < 1.00 \text{ E-}5$. Violin plots were visualized with VlnPlot function and Feature plots were depicted with FeaturePlots function.

2) Regarding the violin plot in Fig.1E, it is problematic the fact that the authors did not remove cells with 0 counts for their gene of interest. As a result, the shape of the violin plot does not reliably depict the true distribution of cells according to their expression levels of *ZCCHC24*. Therefore, authors ought to reperform such an analysis on the revised cell clusters according to the previous comment as well as adding a statistic regarding the median expression differences.

Thank you for your comments. As mentioned above, we have added a description of the scRNA-seq analysis. For the expression of *ZCCHC24*, we described the p -value as a marker gene of the cluster and evaluated the specificity of the expression of *ZCCHC24* compared to other clusters. We also added the explanation for scRNAseq analyses.

Following them, we changed the manuscript as follows.

Line 96-99

Because *ZCCHC24* showed a specific expression pattern (p -value as marker genes: $1.79\text{E-}169$) in the mesenchymal-like population in both scRNA-seq datasets, we evaluated the potential role of *ZCCHC24* in tumorigenicity and treatment resistance in TNBC (Fig 1E, Appendix Fig 2C-D).

Line 432-438

Gene expression matrix was analyzed with principal component analysis (PCA) and we utilized unsupervised share nearest neighbor (SNN) clustering to the genes with a resolution of 0.025. Then they are visualized with Uniform Manifold Approximation and Projection (UMAP). Uniquely expressed genes in each cluster were analyzed with the Seurat FindConservedMarkers function. As the criteria for marker

genes, we set threshold as $p < 1.00 \text{ E-}5$. Violin plots were visualized with VlnPlot function and Feature plots were depicted with FeaturePlot function.

Figure 2

3) In Fig.2D, the authors' choice to select several genes whose RNA half-life is marginally changed upon ZCCHC24 loss (very close to the borderline) is rather a weak decision since several other hits (in grey) seem to have a more robust change. Authors should first highlight what kind of genes do represent these hits and most importantly identify/select at least a couple of these hits and perform key experiments similar to what has been done for ZEB1 in Fig.2 and subsequent figures.

4) Since authors have at their disposal a very nice collection of homemade datasets, why do not they proceed with an overlap between highly ZCCHC24-dependent genes (based on RNAseq) with strong hits from the BRICseq to identify the most promising targets? That would provide a solid basis for selecting the right key genes whose expression is reliant on ZCCHC24 and pivotal for the aggressiveness of TNBC, at least this approach is better than just "cherry-picking" some genes from Fig.2D.

Thank you for your comments. Following your comments, we have identified 124 common genes between the DEGs of RNA-Seq for MDAMB231 cells and destabilized transcripts in BRIC-Seq, as shown in Appendix Table 3. Among these genes, those important for cancer stemness, such as CD44, ZEB1, and NOTCH2, were included. Moreover, we also performed gene ontology analysis for these 124 genes, and showed that genes with function of "Negative regulation of apoptotic process," "Cell matrix adhesion," "Cell migration" are specifically enriched (Fig EV1D). These functions are essential for the stemness, invasion, and survival of breast cancer cells. As representative genes characterizing these functions, *ADAMTS1*, *CDH11* and *JAG1* are identified.

As common genes were downregulated in both PDX and MDAMB231 cells, 376 genes, represented by *ADAMTS1*, *ACVRI*, *CDH11*, and *CD44* were identified and are listed in Appendix Table 2. These genes are considered the main targets of ZCCHC24 in TNBC. Regarding the scRNA-seq data for PDX, *ACVRI*, *ADAMTS1*, *CDH11*, and *JAG1* were relatively highly expressed in the mesenchymal-like population, and knockdown of ZCCHC24 in vivo decreased these genes in the mesenchymal population (Appendix Fig 1, Fig 5F). These results demonstrated the important roles of ZCCHC24 in maintaining the expression of these genes. Moreover, we showed in the Actinomycin D test that knockdown of ZCCHC24 decreased the stability of the mRNAs of these genes (Appendix Fig 5).

In addition, the overexpression of ZCCHC24 led to the upregulation of *CDH11*, *JAG1*, *ZEB1* and *NOTCH2* (Fig EV1A). These results support our hypothesis that ZCCHC24 is essential for the maintenance of breast cancer stemness and tumor invasion/survival without bias.

Following these results, we changed the manuscript as follows.

Line 104-112

This revealed that expressions of a set of genes critical for the characterization of CSC populations, such as *CD44*, *NRP1*, *ZEB1*, and *NOTCH2*, and a set of genes important for cancer progression and invasion, such as *JAG1*, *ADAMTS1*, *ACVRI* were specifically and significantly downregulated (Fig 2A, Appendix Table 2). Quantitative PCR analyses of MDAMB231 cells, the claudin-low TNBC cell line HCC38, and PDX knockdown with siRNA against ZCCHC24 also showed the downregulation of *CD44*, *NRP1*, *ZEB1*, *NOTCH2*, and *ADAMTS1* expression whereas overexpression of ZCCHC24 in MDAMB231 cells increased the expression of *NOTCH2*, *ZEB1*, *CDH11*, and *ZEB1* (Fig 2B, Fig EV1A, Appendix Fig 3).

Line 123-134

We observed reduced stability of mRNAs derived from genes involved in tumorigenicity and treatment resistance (Fig 2D), including *NOTCH2*, *CD44*, *ZEB1*, *CDH11*, and *ADAMTS1*. Based on the RNA-Seq results, 124 genes were identified as differently expressed genes with destabilized transcripts (Appendix Table 3). Gene ontology analysis of these 124 genes with DAVID showed that genes function with "Negative regulation of apoptotic process," "Cell matrix adhesion," "Cell migration," and "cell-cell

adhesion," which are important for cancer stemness were regulated with *ZCCHC24* (Fig EV1D). To confirm these results, we analyzed the effect of Actinomycin D addition on RNA stability after transcription was stopped. The results showed that knockdown of *ZCCHC24* decreased the stability of mRNAs derived from genes important for tumorigenicity, tumor progression, and treatment resistance, such as *ZEB1*, *NRP1*, *CD44*, *NOTCH2*, *CDH11*, *ADAMTS1* and *JAG1* (Fig 2E, Appendix Fig 5).

5) Probably one of the most "interesting" concerns is that authors utilized one single siRNA against *ZCCHC24* throughout their whole work, where mentioned. "Interesting" because the use of one siRNA is highly raising the possibility of off-target effects. For this reason, I would strongly suggest them make use of a "smart pool" of different *ZCCHC24*-specific siRNA in equimolar amounts and perform several important key experiments (similar to Fig. 2B, Fig.2D) to validate their findings from RNAseq and BRICseq and if possible for some in vivo experiments using PDX or MDAMB231 cells (Fig. 5A-B or EV5E) and in vitro functional assays (like in EV5A). For the RT-qPCR, performing a single-siRNA silencing out of the smart pool showing similar effects to the "smart pool" would be highly suggested.

Thank you for your comments. Using another siRNA targeting *ZCCHC24*, we performed qPCR, reporter assays, and *in vitro* cell viability assays. qPCR analysis showed that the expression of the target genes of *ZCCHC24* (*ZEB1*, *CD44*, *NRP1*, *NOTCH2*, *CDH11*, and *ADAMTS1*) was similarly downregulated by si*ZCCHC24_2* (Appendix Fig 3). In addition, we performed reporter assay with siRNA against *ZCCHC24* for the reporter with 3'UTR of *ZEB1*, and confirmed the decrease of reporter activities with two types of siRNA against *ZCCHC24* (Fig 3H). Finally, by the dual use of siRNA against *ZCCHC24* and Doxorubicin, we confirmed the additive effects compared to the single use of the two types of siRNA (Fig EV5D).

Moreover, using another siRNA against *ZEB1*, *JUN*, and *YAP1*, we performed qPCR analysis and confirmed a decrease in the mRNAs expression of *ZCCHC34* (Appendix Fig 8C-D). Following these results, we changed the manuscript as follows.

Line 108-112

Quantitative PCR analyses for MDAMB231, the **claudin-low** TNBC cell line HCC38 and PDX knockdown with siRNA for *ZCCHC24* also showed downregulation of *CD44*, *NRP1*, *ZEB1*, *NOTCH2*, and *ADAMTS1* while overexpression of *ZCCHC24* in MDAMB231 cells increased the expression of *NOTCH2*, *ZEB1*, *CDH11*, and *ZEB1* (Fig 2B, Fig EV1A, Appendix Fig 3).

Line 154-156

We also performed a reporter assay using 3'UTR of *ZEB1* with knockdown of endogenous *ZCCHC24* of MDAMB231 and showed that suppression of *ZCCHC24* expression decreased reporter activities (Fig 3H).

Line 174-177

Furthermore, knockdown of *ZEB1*, *YAP* or *JUN* in MDAMB231 cells and PDX downregulated *ZCCHC24* expression, whereas overexpression of *ZEB1* upregulated *ZCCHC24* expression as evaluated by quantitative PCR (Fig 4B, Fig EV3A, Appendix Fig 8).

Line 226-230

To further evaluate this, we explored doxorubicin, which is used as a first-line compound in chemotherapy for TNBC. The addition of Doxorubicin to MDAMB231 or PDX cells increased the expression of *ZCCHC24* (Fig EV5B). Furthermore, the dual use of doxorubicin with siRNA against *ZCCHC24* had additive effects on suppressing cell viability compared to a single treatment (Fig EV5C-D).

6) If I am not mistaken, in Fig. 2E authors did not mention in the method section to which "housekeeping" gene the expression of the several tested genes was normalized. Also, the use of actinomycin D at certain concentrations may affect the expression of broadly used "housekeeping" genes such as 18S transcribed by RNAPol-I or even more several known like ACTB, RPLP0, GAPDH etc that are transcribed by RNAPol-II. This renders the normalization to any of the abovementioned genes quite inappropriate. Therefore, authors are highly recommended to use an alternative inhibitor such as alpha-amanitin that can specifically RNAPol-II and -III and preferentially use an RNAPol-I-transcribed gene such as 18S to which they can normalize and in general.

Thank you for your comments. In our Actinomycin D analysis, we utilized *PSMB2* as a reference gene and have added a description in the Methods section as shown below. Evaluation of mRNA stability with Actinomycin D addition is common in many reports, and we followed the methods described in those reports (Kataoka et al, Nature 2016, Coelho et al, Immunity 2018).

Furthermore, we performed experiments with alpha-amanitin to specifically stop the effects of RNA Pol-II. However, owing to its toxicity, the expression of 18S rRNA transcribed by RNA Pol-I also changes, making it difficult to assess mRNAs stability under appropriate conditions.

Line 529-530

PSMB2 gene were utilized as a reference gene for the analysis.

7) In Fig.EV2, effects in FACS are slightly marginal. It is highly recommended to perform a western blot for the same stemness markers in control and ZCCHC24-silenced TNBC cells.

Thank you for your comments. We performed western blot analyses for MDAMB231 knocked down using siRNA against ZCCHC24. We confirmed the decrease in CD44 and NRP1 protein levels, as shown in Fig EV1C.

Following this result, we changed the manuscript as follows.

Line 115-116

Downregulation of the target genes of ZCCHC24 was also confirmed at the protein level using western blotting (Fig EV1C).

Figure 3

1) The identified binding motif of ZCCHC24 shown in Fig. 3D, is also a well-known poly-A motif that recruits several other cap-proteins of the 3'UTR such as the poly-A binding protein PABP1. How specific can this motif be for ZCCHC24? The fact that authors have designed a reporter construct with the identified binding site of their favorite factor still does not preclude the possibility that mutation of this binding site and subsequently reported activity decay is exclusively explained by the loss of ZCCHC24 binding to it but could be also explained by the other proteins such as PABP1 that share the same binding motif. The setting that authors used of overexpressing ZCCHC24 provides only part of the needed information to ensure dependency of mRNA with the respective binding motif on this factor. Therefore, equally important is to follow a loss-of-function approach for ZCCHC24 (similar to the previous experiments) to address this question. For example, a naïve suggestion would be to perform a silencing of a dominant candidate equally binding to the WT reporter e.g. PABP1 alone or with ZCCHC24 silencing, and perform a luciferase reporter assay. At least silencing of the latter should have an exclusive or a strong additive effect on the reporter activity.

Thank you for your comments. As mentioned above, we performed reporter assay with siRNA against ZCCHC24 for the reporter with 3'UTR of ZEB1, and confirmed the decrease of reporter activities with two types of siRNA against ZCCHC24 (Fig 3H). This result suggests that endogenous ZCCHC24 regulates target genes expression via target 3'UTR sequences. Following the result, we changed the manuscript as follows.

Line 154-156

We also performed a reporter assay using 3'UTR of *ZEB1* with knockdown of endogenous ZCCHC24 of MDAMB231 and showed that suppression of ZCCHC24 expression decreased reporter activities (**Fig 3H**).

2) What is the cumulate fraction represented in Fig.3E? As it will be hard to interpret this graph for several readers, authors are highly recommended to draw a violin plot with the different groups of genes depending on the number of ZCCHC24-binding sites that bear and subsequently use the log₂FC as a variable on the y-axis. This should visually better demonstrate the differences in gene expression upon ZCCHC24 loss in genes with >10 binding sites. Also, does this latter gene group with >10 binding sites encompass key genes promoting stemness and a mesenchymal state? Because it seems that ZEB1 and CD44 are not good examples based on the number of motif sites they show in Fig.3F.

Thank you for your helpful comments. As suggested, we have depicted the violin plot as shown in Fig 3E. The relationship between the binding sites of ZCCHC24 and changes in mRNA expression in RNA-Seq analyses has become more understandable.

In the PAR-CLIP analyses, 4-thiouridine (s4U) was incorporated into the cells. After UV-Crosslinking and reverse transcription, the sites bound by RBP were marked with a T > C mutation. With peak caller "PAR-alyzer," we calculated this "T > C mutation" numbers and forms clusters. Finally, the peak caller judges whether the clusters formed are binding sites. Because of this mechanism, peak clusters often overlap, as is the case for ZEB1 and CD44. In our calculations, the number of binding sites for ZEB1 was 15 and that for CD44 was 23. Therefore, these are appropriate as an example of a target of ZCCHC24. The number of binding sites of ZCCHC24 on PAR-CLIP are shown in Appendix Table 4.

Figure 4

1) Authors should perform a ChIP-qPCR for H3K27ac on the identified enhancer region of ZCCHC24 in control and ZEB1-, JUN- and YAP-silenced cells. Also, based on the HiC interaction network, it is not convincing if the TSS of ZCCHC24 is indeed interacting with the identified enhancer region located in the gene body. Possibly, authors should seek some additional enhancer regions based on your HiC analyses and perform a similar ChIP-qPCR.

Thank you for your comments. As you mentioned, we performed ChIP-qPCR against MDAMB231 cells knocked down using siRNA for ZEB1. We showed that the enrichment of H3K27Ac decreased with the knockdown of ZEB1 (Fig EV3C). However, it is also possible that the identified region is one of the candidate enhancer sites and that there may be other sites that function as enhancer sites. We have described the possibility of other candidate enhancer sites in the Discussion section. Following them, we changed the manuscript as follows.

Line 189-191

Moreover, with the knockdown of ZEB1, the decrease in the enrichment of H3K27Ac on enhancer candidate site was confirmed with ChIP-qPCR (**Fig EV3C**).

Line 338-342

Furthermore, we identified the candidate enhancer site of ZCCHC24 from Hi-C and ChIP-Seq analyses, following validation with reporter assays and ChIP-qPCR. However, there may be other candidate

enhancer sites that are important for the regulation of ZCCHC24 expression, and this should be an essential future question for the validation of the ZEB1-ZCCHC24 axis.

2) Authors should quantify their immunofluorescence stainings and make a cell-based correlation analysis for ZCCHC24 and ZEB1 staining. Also, it is highly recommended to show zoom-in perspectives of their stainings.

Thank you for the suggestion. Following your request, we performed an immunofluorescence assay for ZCCHC24 and ZEB1 in 10 TNBC pathological specimens. We identified five cases as ZEB1 positive cases and analyzed them. We took pictures with a higher resolution, as shown in Fig 4G.

Figure 5

1) Authors performed a siRNA-mediated loss of ZCCHC24, therefore they should explain if they used a trypan-blue exclusion approach to surely inject only viable cells in Fig.5A-B and Fig.EV5E. Also, after 3 weeks of subcutaneous injection, most probably the silencing effect vanishes. However, the ZCCHC24-downstream effect could be still not reversed thereby still bearing a minor expression of e.g. ZEB1, JUN, etc and ultimately demonstrating a high apoptotic rate. Therefore, authors should consider performing an IHC or IF for the abovementioned markers or at least making use of their scRNAseq data and addressing these concerns.

Thank you for your comments. In the Methods section, we have added a description of the confirmation of the proportion of viable cells using the trypan blue assay before the subcutaneous transplantation of PDX in NOG mice. Moreover, we utilized special siRNAs (stealth RNA), which showed prolonged effects compared to other normal siRNAs, to knock down ZCCHC24. Nevertheless, it is possible that the effects of the ZCCHC24 knockdown were transient. Therefore, we generated a violin plot for the expression of ZCCHC24 by splitting the samples into siNC and siZCCHC24 as shown in Fig 5F. We showed that the expression of ZCCHC24 decreased and that ZCCHC24 positive cells also decreased. Moreover, the expression of genes downstream of ZCCHC24, such as *ZEB1*, *NRP1*, and *ADAMTS1*, was also decreased (Fig 5F, Appendix Fig 10). These results support our hypothesis that ZCCHC24 plays an important role in the maintenance of mesenchymal-like cell populations, including CSCs, by regulating these genes.

Following these results, we changed the manuscript as follows.

Line 217-219

Moreover, downstream genes of ZCCHC24 such as *ZEB1*, *NRP1*, and *ADAMTS1* were downregulated on this population, indicating the importance of ZCCHC24 for the maintenance of mesenchymal-like populations (Fig 5F, Appendix Fig 10).

Line 667-669

PDX (Patient #1) (1×10^6 cells/well) were plated in a 10-cm dish and cultured for 24 h. The cells were transfected with siRNA for 24 h. The viability of cells were checked with Trypan Blue staining, and we confirmed that the proportion of viable cells was over 80 %.

2) If authors have a knockout clone of ZCCHC24, authors should consider performing a similar in vivo experiment.

Thank you for your comments. As you have noted, ZCCHC24 knockout clone is a strong tool for evaluating the roles of ZCCHC24 *in vitro* and *in vivo*. However, because of its importance in cell survival, we could not establish ZCCHC24 knockout clone for this analysis. Therefore, we have described this as a limitation in the Discussion section as follows:

Line 335-338

Also, we could not pick up ZCCHC24 knocked out (KO) cells due to its importance in cell survival. The analysis with ZCCHC24 KO cells would help to better characterize the roles of ZCCHC24 on CSCs.

3) Also, since chemotherapy is a standard-of-care therapeutic approach in TNBC, authors should consider introducing e.g. a platin-based chemotherapeutic and check for a stronger sensitiveness in ZCCHC24-silenced cells. Also, important is to corroborate their findings with publicly available patient-derived transcriptomic and follow-up data.

Thank you for your comments. Following your suggestions, we also performed experiments with doxorubicin, which is clinically used for TNBC chemotherapy. We observed the upregulation of ZCCHC24 in MDAMB231 and PDX cells (Fig EV5B). Next, we transfected doxorubicin siRNA against ZCCHC24 into MDAMB231 and PDX cells and measured cell viability. The dual use of doxorubicin and siRNAs showed additive effects on cell viability suppression (Fig EV5C-D). These results suggest the validity of simultaneous targeting of ZCCHC24 for the use of doxorubicin as a chemotherapy against TNBC.

In addition, as described in Fig EV5A, reanalysis of the transcriptome data of specimens of TNBC after neo-adjuvant chemotherapy (Haltzis et al, JAMA 2011, Loibl et al, Lancet Oncology 2018), cases with high expression of ZCCHC24 showed high proportion of recurrence of TNBC compared with those with low expression. These data suggest that ZCCHC24 expression is important for predicting recurrence after neoadjuvant chemotherapy.

Following these results, we changed the manuscript as follows.

Line 220-230

CSCs have been reported to show resistance against chemotherapy, and to be a main cause of disease recurrence (Dean *et al*, 2005). To determine whether ZCCHC24 might play a role in this resistance mechanism we reanalyzed clinical samples from two clinical studies that had been treated with neoadjuvant chemotherapy (Hatzis *et al*, 2011; Loibl *et al*, 2018). We found that high expression of ZCCHC24 correlated with high pathological residual disease rates (Fig EV5A).

To further evaluate this, we explored doxorubicin, which is used as a first-line compound in chemotherapy for TNBC. The addition of Doxorubicin to MDAMB231 or PDX cells increased the expression of ZCCHC24 (Fig EV5B). Furthermore, the dual use of doxorubicin with siRNA against ZCCHC24 had additive effects on suppressing cell viability compared to a single treatment (Fig EV5C-D).

Figure 6

1) Authors made an interesting observation where ZCCHC24-deficient tumors were sensitized to JQ1 treatment. Authors may consider examining if ZCCHC24-deficient tumors presented an augmented apoptotic rate and more decreased expression of ZEB1 in their dissected tumors.

Thank you for your comments. Based on the precise observations of the reviewer, as shown in Fig 5, ZCCHC24 increased in response to JQ1 treatment. Based on the current data, knockdown of ZCCHC24 suggests the potential induction of total death. As you have pointed out, it remains to be seen how ZEB1-positive cells may be eliminated through the induction of cell death or transformation into ZEB1-negative cells, thereby losing cancer stemness and becoming targets for chemotherapy. Distinguishing between these scenarios is challenging in our current research strategy; however, analyzing the lineage of ZEB1-positive cells in PDX models may help dissect the underlying mechanisms. We highlighted this critical issue in the Discussion section.

Line 320-323

In the current analysis, ZCCHC24-deficient tumors showed an increased apoptotic rate and decreased ZEB1 expression in dissected tumors. Analysis of the lineage of ZEB1-positive cells in PDX models may help dissect the underlying mechanisms in the future.

Minor concerns:

1) Statistics and used thresholds are missing in Fig. 2D-E

Thank you for your comments. We have added a description of the statistics and thresholds as follows:

Legend of Fig2

D, Results of BRIC-Seq. Transcripts with 0.8 folds of half-lives are marked as orange on the graph.

E, RNA remaining levels of indicated genes upon actinomycin D treatment of TNBC cell line, MDA-MB-231, knocked down with siRNA for ZCCHC24 compared to a negative control (si negative control (NC)) (N=3, biological replicates each, ** $p < 0.01$).

2) Authors should explain why NOTCH2 stability increased with time upon actinomycin D treatment (Fig.2E).

Thank you for your comments. As shown in Fig 5E, *NOTCH2* levels increased after Actinomycin D treatment. This was due to the slower decay of the mRNAs of *NOTCH2* than that of *PSMB2* as the internal control gene. As you suggested, we concluded that this was an error in our elucidation. Therefore, we have corrected the evaluation as follows: we calculated the relative remaining mRNA percentages for samples with siZCCHC24 compared to samples with siNC at each time point (Fig 2E, Appendix Fig 5).

3) In Fig.3C, the y-axis does provide the unit for bamcoverage (is it RPKM?). Also, what is the scale for the x-axis? Did they perform a "scale-region" aggregate profiling for the binding of ZCCHC24? They should mention this in the legend or/and methods.

Thank you very much for your comments. We utilized RCAS for the visualization of binding sites of ZCCHC24. X-axis showed the relative position coordinates within UTRs or CDS, and Y-axis showed mean coverage of the enriched regions. Mean coverage is not neither RPKM nor FPKM. Rather, it is the calculated value after using RCAS reflecting its distribution on its transcripts, as shown in homepage of RCAS software by the manufacturer

(<https://www.bioconductor.org/packages/release/bioc/vignettes/RCAS/inst/doc/RCAS.vignette.html>).

We added the description on the method section as follows.

Line 577-579

The regions of binding sites were analyzed with RCAS

(<https://www.bioconductor.org/packages/release/bioc/vignettes/RCAS/inst/doc/RCAS.vignette.html>).

4) Fig.3H: If the mutant reporter impedes the binding of ZCCHC24 on it, authors should better normalize the differences to the WT-NC condition and not normalizing to the internal control each time. The same applies to Fig.4C-D and normalizing to the control.

Thank you very much for your comments. As you suggested, we normalized the differences to the WT-NC (Fig 3H, 4C) and to the Control-siNC (Fig 4D), and calculated the significance using ANOVA and Tukey's post-hoc test.

5) Authors should explain why the track with ZCCHC24 binding sites, at least the summits of those does not align with the motif sites identified by PAR-CLIP. Also, the peak site track with the peak track (bigwig file I guess) of NRPI is not matching to each other and should be corrected.

Thank you for your comments. We conclude that the current PAR-CLIP data are accurate. Because the peak caller detected the T > C mutation caused by crosslinking of RBPs with UV, sometimes the peak sites, the summits of CLIP peak data, and motif sites did not align well or were slightly misaligned a little bit due to the low proportion of uridine around the binding sites. Regarding the CLIP peak data for the NRPI region, the peak site aligned well with the summits of the CLIP data and motif sites. Therefore, we did not alter the current PAR-CLIP data.

6) Statistic for Fig.4E.

Thank you very much for your comments. We added the description of statistics as follows.

Legend of Fig 4E

E, Distribution plot of ZEB1 mRNA and ZCCHC24 mRNA expression from TCGA-BRCA database. Relevance is calculated from the FPKM of each gene as correlation coefficient.

7) In the graphical abstract (Fig.6B), authors should correct the "total kill" and "chemotherapy" as JQ1 is not a chemotherapy and most importantly, based on their in vivo studies, combination of ZCCHC24 and JQ1 does not "totally" reduce the tumor burden of mice.

Thank you for your comments. As for the description of the treatment with BET inhibitors, we changed the term "chemotherapy" to "Molecular target therapy." Moreover, *in vitro*, we showed the effect of the dual use of doxorubicin with siRNA against ZCCHC24, demonstrating the effectiveness of targeting ZCCHC24 to enhance the effect of chemotherapy. Therefore, we added the term "Molecular target therapy" in addition to "Chemotherapy" in Fig 6B. We also changed the term "total kill" to "Reduction" in Fig 6B.

We also changed the description as follows.

Line 317-320

ZCCHC24 knockdown specifically reduces the mesenchymal like population including CSCs, allowing for the targeting of proliferative cell populations with **molecular target therapy** while using ZCCHC24-targeting siRNA in combination, **thereby enhancing the effects of molecular target therapy *in vivo*** (Fig 6B).

8) The molecular weight of proteins in western blots is missing and should be added.

Thank you for your comments. We have added the molecular weights of the proteins to all western blotting figures (Fig EV1C, EV3B, Appendix Fig 6, Appendix Fig 7, Appendix Fig 9).

Dear Prof. Asahara,

Thank you for the submission of your revised manuscript to our editorial offices. I have now received the reports from the three referees that I asked to re-evaluate the study, you will find below. As you will see, the referees now support the publication of the study in EMBO reports. Referees #1 and #2 have remaining concerns and suggestions to improve the manuscript, I ask you to address in a final revised manuscript. Please also provide a final p-b-p-response regarding the remaining points of the referees.

- Please provide the abstract written in present tense throughout.
- Please order the manuscript sections like this, using these names:
Abstract - Keywords - Introduction - Results - Discussion - Methods - Data availability section - Acknowledgements - Disclosure and Competing Interests Statement - References - Figure legends - Expanded View Figure legends
- The "Data Availability section" is restricted to datasets deposited at external repositories. Thus, please keep the information on the scRNAseq dataset there, but remove the other text.
- Please remove the legends from the EV figure files. These should only be included into the manuscript text file (see above).
- Please make sure that the number "n" for how many independent experiments were performed, their nature (biological versus technical replicates), the bars and error bars (e.g. SEM, SD) and the test used to calculate p-values is indicated in the respective figure legends (also for potential EV figures and all those in the final Appendix). Please also check that all the p-values are explained in the legend, and that these fit to those shown in the figure. Please provide statistical testing where applicable. Please avoid the phrase 'independent experiment', but clearly state if these were biological or technical replicates. Please also indicate (e.g. with n.s.) if testing was performed, but the differences are not significant. In case n=2, please show the data as separate datapoints without error bars and statistics. See also:
<http://www.embopress.org/page/journal/14693178/authorguide#statisticalanalysis>

If n<5, please show single datapoints for diagrams. Moreover:

- It seems that the legend for figures EV 3a-b is interchanged. This needs to be rectified.
- Please note that the exact p values are not provided in the legends of figures 2b, e; 3g-i; 4b-d; 6a; EV 1a; EV 2; EV 3a, c; EV 4a; EV 5b-e. Please provide exact p-values.
- Please indicate the statistical test used for data analysis in the legends of figures 2a; 4b; 5b; EV 1d; EV 4b-d.
- Please note that in figure 4c; there is a mismatch between the annotated p values in the figure legend and the annotated p values in the figure file that should be corrected.
- Please note that the box plot needs to be defined in terms of minima, maxima, centre, bounds of box and whiskers, and percentile in the legend of figure 3e.
- Please note that information related to n is missing in the legends of figures 1e; 3e; 5f.
- Please note that the scale bar needs to be defined for figure 4g.
- Please note that the white arrows are not defined in the legend of figure 4g. This needs to be rectified.
- Please add to each legend (main, and EV figures, where applicable) a 'Data Information' section explaining the statistics used or providing information regarding replicates and scales. See:

- Please add scale bars of similar style and thickness to microscopic images, using clearly visible black or white bars (depending on the background). Please place these in the lower right corner of the images themselves. Please do not write on or near the bars in the image but define the size in the respective figure legend. Presently, scale bars are missing from panel EV4A.
- Please make sure that all figure panels are called out separately and sequentially. It seems, presently a callout to Figure EV6 is missing. Please check.
- Please add a title page ('Appendix for ...') with a table of contents (ToC) and page numbers to the Appendix file. The figures need "S" in their title (Appendix Figure Sx). Please also make sure the figures are called out like this.
- The separately uploaded Appendix tables 1-4 are datasets. Please name these 'Dataset EVx' and upload these as dataset file. Please upload the original excel file and put a legend on the first TAB. Moreover, please change the callouts to these files accordingly ('Dataset EVx').

- Please include the present Appendix Table 5 into the Appendix file (named Appendix Table S1). Please change the callout(s) to this file.

- Please include the information provided in Appendix Table 6 in the Reagents & Tools table (see below).

- Please remove the reagents and tools table from the main manuscript text file. I have attached a template for that in word format. Please upload the filled in table to the manuscript tracking system as 'Reagent Table' file. Please also adjust any callouts to this table. The example linked below shows how the table will display in the published article and includes examples of the type of information that should be provided for the different categories of reagents and tools. Please list your reagents/tools using the categories provided in the template and do not add additional subheadings to the table. Reagents/tools that do not fit in any of the specific categories can be listed under "Other":

https://www.embopress.org/pb%2Dassets/embo-site/msb_177951_sample_FINAL.pdf

Best,

Referee #1:

The authors tried to address my previous comments. However, some responses and newly provided data do not fully resolve my concerns and the authors missed the chance to perform an unbiased data analysis. Instead, one is left with the poor feeling that most of the data are cherry-picked and do not tell the whole story...

These are my open concerns and additional comments to be considered by the authors:

1) The authors now include a list of differentially expressed genes (only downregulated genes) upon siRNA-mediated depletion of ZCCHC24 in MDA-MB-231 and a PDX (new Appendix Table 2). First, the table header is misleading and needs to be more precise! Furthermore, the table does not include the standard information one would expect to find in such a table. Please include fold changes and corrected p-values/FDR per gene and cell line! Moreover, the authors stated that the function of ZCCHC24 is largely unknown. Hence, it is unclear why the authors focused on downregulated genes only, but fully ignored the up-regulated ones. Please provide at least some basic information about the (commonly) up-regulated genes (gene identity, fold change, corrected p-value) as well!

2) In my previous comments I asked the authors to provide more details about the genes whose mRNA stability was altered upon ZCCHC24 and I wanted to know why the authors did not investigate the more interesting extremes but instead focused on (cherry-picked) transcripts whose stability changed only marginally. The authors did not fully address this issue but simply provide a list of genes (Appendix Table 3). This list is again lacking critical information, i.e. fold changes and p-values. The authors should calculate the differences in the mRNA half-lives of the detected genes and provide this information. Alternatively, the authors could just include a table with gene names and the respective half-life in the siNC and siZCCHC24 group, respectively. The Appendix Table 3 in its current form is of low value.

3) I previously asked the authors to provide information about the nature of overexpression and reporter vectors. While the authors provide now the sequence of the vectors in Appendix Table 5, it still remains unclear which vector backbones had been used for cloning. Please add this information as well.

4) The authors discussed my concern regarding the linear decay of the selected transcripts shown in original Figure 2E. Instead of re-drawing the figure using a semi-logarithmic scaling (y-axis in log₂ scale, x-axis in linear scale), the authors changed the figure and include bar graphs now. Hence, the authors should remove the irrelevant sentences from their discussion (lines 327 - 330).

5) An additional, conceptual question that needs to be clarified: Why did the authors decided to distinguish between a mesenchymal and a TGFB1-positive population in the scRNA-Seq analysis (Figure 1)? It is not clear to me why this is necessary / relevant / useful, since TGFB1 is known to induce EMT thereby leading to a mesenchymal population. Please clarify!

6) The authors should carefully revise all their figures and respective figure legends. For example, there seems to be something wrong with Figure EV3.

7) Finally, the authors are advised to perform professional language editing to improve grammar and style.

Referee #2:

The authors made several revisions but did not address some of my concerns directly as outlined below:

1.
In my initial review I noted that whether ZCCHC24 targets CSC per se as suggested by the authors or the mesenchymal-like state. Unless the authors show that CSC population (defined by flow cytometry - e.g. markers below) is directly targeted - could the effects in Figs.5-6 be due to oncogene/non-oncogene addiction that curtails the growth of the mesenchymal population? This should either be addressed experimentally - better explained - - or the text may be modified from "targeting CSC" to "targeting the mesenchymal state".

R2: - the authors changed the text from "targeting CSC" to "targeting the mesenchymal state including CSC". I'm not sure this is acceptable without direct demonstration that CSCs are functionally depleted. The effect on EMT is not simple as there is ample evidence that partial EMT, enabling group migration rather than complete EMT. drive metastatic dissemination. The authors should adhere to the data they have and not over-interpret or assume that blocking CSCs (or even EMT) underlies the effect they see without direct mechanistic evidence.

The authors also did not address my question "What's CDC42? What does it represent? - what's the reference for its use to sort CSC?"

The new analysis with bona fide CSC markers is fine - but the function and rationale for using CDC42 for CSC analysis should be provided.

2.
Fig 4A - the range scale of 1-50 on the right overlaps with the TAS pyramid - please move it a bit

Authors: Thank you for your comments. We moved the range scale from to 1-50 bit, as suggested in Fig4A.

R2: This has not been done yet; the range scale on the right still overlaps with the TAS pyramid

3.
Fig 4G - very difficult to see the details. Perhaps show a smaller area at much higher magnification.

Authors: Thank you for the suggestion. Following your request, we performed an immunofluorescence assay for ZCCHC24 and ZEB1 in 10 TNBC pathological specimens. We identified five cases as ZEB1 positive cases and analyzed them. We took pictures with a higher resolution, as shown in Fig 4G.

R2: The images are still at very low magnification. Please show higher magnification with single stained cells.

4.
Fig 5 - is this siRNA (transient) or shRNA (stable) knockdown? Are these lenti-shRNA clones? The clones should be characterized - e.g. showing the level of ZCCHC24 knocked-down.

Authors: Thank you for your comments. We utilized special siRNAs (stealth RNA), which showed prolonged effects compared to other normal siRNAs, to knock down ZCCHC24. Nevertheless, it is possible that the effects of the ZCCHC24 knockdown were transient. Therefore, we generated a violin plot for the expression of ZCCHC24 by splitting the samples into siNC and siZCCHC24 as shown in Fig 5F. We showed that the expression of ZCCHC24 decreased and that ZCCHC24 positive cells also decreased.

R2: Please show a western blot to demonstrate knockdown of the ZCCHC24 protein - not only mRNA.

Referee #3:

Authors have covered in a satisfactory way all important concerns, therefore, their revised work should be considered for publication. Warm congratulations to all contributing co-authors.

Response to reviewer's comments

Referee #1:

The authors tried to address my previous comments. However, some responses and newly provided data do not fully resolve my concerns and the authors missed the chance to perform an unbiased data analysis. Instead, one is left with the poor feeling that most of the data are cherry-picked and do not tell the whole story...

These are my open concerns and additional comments to be considered by the authors:

1) The authors now include a list of differentially expressed genes (only downregulated genes) upon siRNA-mediated depletion of ZCCHC24 in MDA-MB-231 and a PDX (new Appendix Table 2). First, the table header is misleading and needs to be more precise! Furthermore, the table does not include the standard information one would expect to find in such a table. Please include fold changes and corrected p-values/FDR per gene and cell line! Moreover, the authors stated that the function of ZCCHC24 is largely unknown. Hence, it is unclear why the authors focused on downregulated genes only, but fully ignored the up-regulated ones. Please provide at least some basic information about the (commonly) up-regulated genes (gene identity, fold change, corrected p-value) as well!

Thank you for your comments. Following your suggestions, we have included differentially expressed genes downregulated or upregulated by the knockdown of ZCCHC24 with information regarding their gene name, fold changes, and padj for the transcriptome analysis of both MDAMB231 and PDX cells in Dataset EV2. Moreover, we have also included information on commonly upregulated or downregulated genes in Dataset EV2.

2) In my previous comments I asked the authors to provide more details about the genes whose mRNA stability was altered upon ZCCHC24 and I wanted to know why the authors did not investigate the more interesting extremes but instead focused on (cherry-picked) transcripts whose stability changed only marginally. The authors did not fully address this issue but simply provide a list of genes (Appendix Table 3). This list is again lacking critical information, i.e. fold changes and p-values. The authors should calculate the differences in the mRNA half-lives of the detected genes and provide this information. Alternatively, the authors could just include a table with gene names and the respective half-life in the siNC and siZCCHC24 group, respectively. The Appendix Table 3 in its current form is of low value.

Thank you for your suggestions. We have included information on the gene names, half-lives for both siNC and siZCCHC24, and the ratio for genes with destabilized genes (half-lives ratio < 0.8) of BRIC-Seq in Dataset EV3.

3) I previously asked the authors to provide information about the nature of overexpression and reporter vectors. While the authors provide now the sequence of the vectors in Appendix Table 5, it still remains unclear which vector backbones had been used for cloning. Please add this information as well.

Thank you for your comments. As suggested, we have included information on the vector backbones used for cloning in the "Methods" section, as shown below:

Lines 358-375:

For the overexpression vector, the sequence of the open reading frame of ZCCHC24 or ZEB1 was inserted with a 3XFLAG peptide at the N-terminus between the NheI and NotI restriction enzyme sites on the pcDNA3.1 vector. **The pCLT vectors, lentiviral vectors for cDNA expression using the tet-on system, were created by modifying the pCSII-CMV-IRES-Venus vector (RDB04383; RIKEN). The CMV-IRES-**

Venus sequence was removed by inverse PCR, and the tet-responsive promoter (Clontech) and PGK promoter-puromycin-N-acetyltransferase-P2A-reverse tetracycline transactivator (rtTA) were cloned into the pCSII vector to create the pCLT vector. The lentiviral expression vector was constructed by inserting GFP or ZCCHC24 cDNA sequences with 3xFLAG downstream of the tet-on promoter into the lentiviral vectors. For luciferase reporter vectors for post-transcriptional regulation, 3'UTR of CD44, ZEB1, or their partial sequences with mutation in binding sites were inserted downstream of the luciferase gene. The control vector (pLuc2-KAP-MCS) is as shown in a previous study (Ito *et al.*, 2017). For the luciferase reporter vectors for transcriptional regulation, a control reporter vector was designed and generated by inserting the PGK promoter sequence into the multi-cloning site (MCS) of pGL4.10 (Promega). The estimated regulatory sequence was inserted into the MCS of the reporter vectors. The sequences of other reporter vectors and lentiviral vectors expressing the tet-on promoter are shown in **Appendix Table S1**.

4) The authors discussed my concern regarding the linear decay of the selected transcripts shown in original Figure 2E. Instead of re-drawing the figure using a semi-logarithmic scaling (y-axis in log₂ scale, x-axis in linear scale), the authors changed the figure and include bar graphs now. Hence, the authors should remove the irrelevant sentences from their discussion (lines 327 - 330).

Thank you for your comments. As you suggested, we have removed irrelevant sentences from the “Discussion” section.

5) An additional, conceptual question that needs to be clarified: Why did the authors decided to distinguish between a mesenchymal and a TGFB1-positive population in the scRNA-Seq analysis (Figure 1)? It is not clear to me why this is necessary / relevant / useful, since TGFB1 is known to induce EMT thereby leading to a mesenchymal population. Please clarify!

Thank you for your comments. Initially, we characterized this cell population using CDC42, which has been reported to contribute to cancer progression, as a marker gene. However, following the suggestion of Reviewer #3, we searched for other representative genes to characterize this cell population. As a result, we identified TGFB1, TGFBR1, and ITGA6 as genes that characterize this population (Fig. 1C). As you have pointed out, TGFB1 is well known for its ability to induce EMT. In addition, CDC42 and ITGA6 expressed in this population have been reported to induce EMT and tumor initiation (Keely *et al.*, *Nature* 1997; Azios *et al.*, *Neoplasia* 2007; Zhang *et al.*, *Oncogene* 2014; Vassilopoulos *et al.*, *Oncogene* 2014). In our current analysis, based on previous reports (Al-Hajj *et al.*, *PNAS* 2003; Bianchini *et al.*, *Nat. Rev. Clin. Oncol.* 2016; Ginestier *et al.*, *Cell Stem Cell* 2007; Tominaga *et al.*, *PNAS* 2019; Zhang *et al.*, *Genes Dev.* 2019), we focus on the subset that is double-positive for ZEB1 and NRP1. From this perspective, the relatively low expression of ZEB1 is a significant characteristic of the “TGFB1-positive population” (**Appendix Fig. S1**). Additionally, in the “mesenchymal-like population” that we examined in this study, it is interesting that the level of TGFB1 expression was found to be relatively low (**Fig. 1C**), prompting us to make these distinctions.

Based on these considerations, we have revised the text as follows:

Lines 79-85:

As a result, PDX were roughly divided into three populations: an epithelial-like population characterized by the expression of *CD24*, *KRT8*, and *KRT19*; a mesenchymal-like population characterized by the expression of *NRP1*, *ZEB1*, *JUN*, *VIM*, and *ALDH1A3*, and low expression of *CD24* and *TGFB1*; and a TGFB1 positive population, characterized by the expression of *TGFB1*, *CDC42* (Azios *et al.*, 2007; Keely *et al.*, 1997; Zhang *et al.*, 2014), and *ITGA6* (Vassilopoulos *et al.*, 2014) and low expression of *ZEB1* (**Fig. 1A-C**, **Appendix Fig. S1**).

6) The authors should carefully revise all their figures and respective figure legends. For example, there seems to be something wrong with Figure EV3.

Thank you for the suggestion. We have revised the caption of Fig. EV3 accordingly.

7) Finally, the authors are advised to perform professional language editing to improve grammar and style.

Thank you for your comments. Following your suggestions, we have utilized a professional language editing service, Editage, and have included an acknowledgement of their contribution to the Acknowledgements section, as shown below.

Lines 831-839:

We thank Yuki Naito, Lin Liu, Kana Shishido, Tomomi Kato, Mitsuyo Nakajima, and all the staff of the Department of Systems Biomedicine at Tokyo Medical and Dental University (TMDU) for their support and advice, Tsuyoshi Nakagawa for kindly providing pathological samples, Miori Inoue for preparation of pathological samples, Helen Pickersgill for scientific editing and Editage for language editing. This work was supported by the Japan Society for the Promotion of Science KAKENHI (Grant Nos. JP20H05696, JP21K19403 to H.A.), NIH grant (Grant No. AR080127 to H.A.), and AMED-LEAP from AMED (Grant No. JP23gm0010009 to H.A.). This work was also partly supported by Extramural Collaborative Research Grant of Cancer Research Institute, Kanazawa University.

Referee #2:

The authors made several revisions but did not address some of my concerns directly as outlined below:

1.

In my initial review I noted that whether ZCCHC24 targets CSC per se as suggested by the authors or the mesenchymal-like state. Unless the authors show that CSC population (defined by flow cytometry - e.g. markers below) is directly targeted - could the effects in Figs.5-6 be due to oncogene/non-oncogene addiction that curtails the growth of the mesenchymal population? This should either be addressed experimentally - better explained - - or the text may be modified from "targeting CSC" to "targeting the mesenchymal state".

R2: - the authors changed the text from "targeting CSC" to "targeting the mesenchymal state including CSC". I'm not sure this is acceptable without direct demonstration that CSCs are functionally depleted. The effect on EMT is not simple as there is ample evidence that partial EMT, enabling group migration rather than complete EMT, drive metastatic dissemination. The authors should adhere to the data they have and not over-interpret or assume that blocking CSCs (or even EMT) underlies the effect they see without direct mechanistic evidence.

Thank you for your constructive comments. Following your suggestion and considering the data we have, we changed the term "targeting mesenchymal state including CSC" to "targeting mesenchymal state" as shown below.

Lines 23-25:

Here, we identify an RNA-binding protein, ZCCHC24, that is specifically expressed in the mesenchymal-like population of TNBC.

Lines 62-63:

In this study, we identified ZCCHC24 as an RBP that was predominantly upregulated in the mesenchymal-like population of TNBC.

Lines 198-202:

Taken together, our results highlight the potential role of ZCCHC24 in tumorigenicity and treatment resistance via a positive feedback loop, whereby its expression is induced by stemness-associated transcription factors, specifically in the mesenchymal population. Furthermore, it appears to bind directly to and promote the stability of mRNAs transcribed from genes important for tumorigenicity and treatment resistance (Fig. 4F).

Lines 214-217:

Consistent with our hypothesis that ZCCHC24 is critical for maintaining CSC properties, scRNAseq on the formed tumors showed a large decrease in the mesenchymal-like population with ZCCHC24 expression, characterized by *NRPI*-positive, *ZEB1*-positive, and *CD24*-low expression upon ZCCHC24 knockdown (Fig. 5C-E).

Lines 249-251:

Here, we identified ZCCHC24 as a critical RBP that directly controls a specific set of target mRNAs governing tumorigenicity and treatment resistance, and is expressed specifically in the TNBC mesenchymal-like population.

Lines 262-264:

ZCCHC24 was found to possess a unique function in TNBC, where it is strongly expressed and specifically stabilizes important mRNAs, such as ZEB1, CD44, and NRPI, in the mesenchymal-like population by recognizing specific *cis*-elements.

Lines 299-302:

The advantages of targeting ZCCHC24 in mesenchymal-like cells include not only the direct inhibition of the expression of a set of genes related to cancer stemness and EMT, including SNAI and NOTCH, but also the disruption of the auto-amplified gene expression network formed by ZCCHC24 and ZEB1 to maintain CSC traits.

Lines 313-319:

Our study demonstrates that combining therapies targeting ZCCHC24 in the mesenchymal state with standard treatments such as conventional cytotoxic therapy leads to an enhanced anti-tumor effect. ZCCHC24 knockdown specifically reduced the mesenchymal-like population with ZCCHC24 expression, allowing for the targeting of proliferative cell populations with molecular target therapy using ZCCHC24-targeting siRNA in combination, thereby enhancing the effects of molecular target therapy *in vivo* (Fig. 6B).

The authors also did not address my question "What's CDC42? What does it represent? - what's the reference for its use to sort CSC?"

The new analysis with bona fide CSC markers is fine - but the function and rationale for using CDC42 for CSC analysis should be provided.

Thank you for your comments. We initially focused on CDC42 as a gene specifically expressed in the "TGFB1-positive population." CDC42 is a protein belonging to the Rho GTPase family and has been reported to play a crucial role in controlling the epithelial-mesenchymal transition (EMT), cell motility, and drug resistance in breast cancer initiation (Keely et al., *Nature* 1997; Azios et al., *Neoplasia* 2007; Zhang et al., *Oncogene* 2014; Vassilopoulos et al., *Oncogene* 2014). Following the suggestion of Reviewer #3, we explored a distinct gene set expressed in this population, revealing its co-expression with genes such as *TGFB1*, *TGFBRI*, and *ITGA6*, as illustrated in **Appendix Fig. S1**. This population may promote EMT in breast cancer tissues.

In our study, we specifically analyzed cell subsets expressing *ZEB1*, *ALDH1A3*, and *NRPI* based on previous reports (Al-Hajj et al., *PNAS* 2003; Bianchini et al., *Nat. Rev. Clin. Oncol.* 2016; Ginestier et al., *Cell Stem Cell* 2007; Tominaga et al., *PNAS* 2019; Zhang et al., *Genes Dev.* 2019). Notably, we found relatively low expression of ZEB1 in the "TGFB1-positive population," supporting the existence of the ZCCHC24-ZEB1 axis in this context. In future analyses, evaluating the role of CDC42 in EMT focused on the "TGFB1-positive population" will be a critical task.

Based on these findings, we have added references to the following section:

Lines 79-85:

As a result, PDX were roughly divided into three populations: an epithelial-like population characterized by the expression of *CD24*, *KRT8*, and *KRT19*; a mesenchymal-like population characterized by the expression of *NRPI*, *ZEB1*, *JUN*, *VIM*, and *ALDH1A3*, and low expression of *CD24* and *TGFB1*; and a TGFB1 positive population, characterized by the expression of *TGFB1*, *CDC42* (Azios et al, 2007; Keely et al, 1997; Zhang et al, 2014), and *ITGA6* (Vassilopoulos et al, 2014) and low expression of *ZEB1* (Fig. 1A-C, Appendix Fig. S1).

2.

Fig 4A - the range scale of 1-50 on the right overlaps with the TAS pyramid - please move it a bit

Authors: Thank you for your comments. We moved the range scale from to 1-50 bit, as suggested in Fig4A.

R2: This has not been done yet; the range scale on the right still overlaps with the TAS pyramid

Thank you for your suggestion. We apologize for the misunderstanding. We have moved the range scale to the right, slightly higher than before, ensuring that it no longer overlaps with the TAS pyramid.

3.

Fig 4G - very difficult to see the details. Perhaps show a smaller area at much higher magnification.

Authors: Thank you for the suggestion. Following your request, we performed an immunofluorescence assay for ZCCHC24 and ZEB1 in 10 TNBC pathological specimens. We identified five cases as ZEB1 positive cases and analyzed them. We took pictures with a higher resolution, as shown in Fig 4G.

R2: The images are still at very low magnification. Please show higher magnification with single stained cells.

Thank you for the suggestion. Following your comment, we also captured images at 100× magnification of representative single-stained cells and included them in the upper-right corner of each figure. Owing to the performance of the microscope available and the nature of immunostaining, we believe that the magnification shown represents the optimal choice for our analysis.

4.

Fig 5 - is this siRNA (transient) or shRNA (stable) knockdown? Are these lenti-shRNA clones? The clones should be characterized - e.g. showing the level of ZCCHC24 knocked-down.

Authors: Thank you for your comments. We utilized special siRNAs (stealth RNA), which showed prolonged effects compared to other normal siRNAs, to knock down ZCCHC24. Nevertheless, it is possible that the effects of the ZCCHC24 knockdown were transient. Therefore, we generated a violin plot for the expression of ZCCHC24 by splitting the samples into siNC and siZCCHC24 as shown in Fig 5F. We showed that the expression of ZCCHC24 decreased and that ZCCHC24 positive cells also decreased.

R2: Please show a western blot to demonstrate knockdown of the ZCCHC24 protein - not only mRNA.

Thank you for the suggestion. Following your comment, we performed western blotting for PDX (Patient #1) transfected with siRNAs against ZCCHC24, and validated the decrease in ZCCHC24 protein levels (**Appendix Fig. S10**).

Dear Prof. Asahara,

Thank you for the submission of your further revised manuscript to our editorial offices. I have already forwarded to you the reports from the two referee that I asked to re-evaluate your study, you will find again below. As you will see, the referee now supports the publication of your study in EMBO reports. As you know, referee #1 has remaining concerns and suggestions to improve the study. Going through your further revision plan (preliminary p-b-p-response), I think the remaining points of the referee will be adequately addressed.

I thus invite you to revise the manuscript further as indicated in your letter. Please also provide a final p-b-p-response addressing the remaining points.

Moreover, I have these final editorial requests:

- Please have your final manuscript carefully proofread by a native speaker. The text still contains grammatical errors and typos. See e.g. the legend of panel 4G (it should be 'White and yellow arrows', not 'White and yellow arrays').
- Please provide exact p-values for all diagrams as indicated in my last decision letter.
- Please make sure that the source data (SD) for the main figures is updated accordingly in case you are adding or changing data during the final revision. Please also make sure the SD is uploaded as one folder per main figure (with all files for one figure in one folder and ZIPed).

Yours sincerely,

Referee #1:

During the second round of revision the authors addressed most of my concerns. However, some issues still remain:

1) The authors now provide a list of differentially expressed genes after ZCCHC24 depletion in MDA-MB-231 cells and a PDX model including the respective fold change and p-adj. values per gene (Data Set EV2) as requested. However, the applied cut-offs used to assemble these lists are not clear. While the list of deregulated genes in the PDX contains 1478 downregulated and 915 upregulated genes with a p-adj. below 0.001 (cut-off?), the authors included 761 downregulated and 636 upregulated genes with p-adj. even above 0.05 in the list of MDA-MB-231. In addition, the MDA-MB-231 gene list contains rows that lack the gene name but contain FC and p-values.

The authors should align their data presentation and filtering to common scientific standards, i.e. either include the full list of genes (with FC and p-value) or use standard cut-offs, e.g. p-adj.<0.05. They need to mention these cut-offs so that the readers of the manuscript can understand the data.

However, I need to be quite frank about this: Never before have I experienced something like this as a reviewer (for a prestigious journal like EMBO Reports) and it raises severe doubts about the data quality and validity of the analyses due to the non-

scientific presentation and selection of the data throughout this manuscript!

2) In line with this, the authors still do not provide any explanation why they did not investigate the more interesting extremes, but instead focused on (cherry-picked) transcripts whose stability changed only marginally. At least, the authors now include a list of genes whose stability was decreased upon ZCCHC24 knockdown (Data Set EV3). However, the other side of the coin is missing - the list of genes whose stability is increased upon ZCCHC24! Inclusion of these genes would be important in order to evaluate whether the putative ZCCHC24 binding motif, which was identified via PAR-CLIP is specifically enriched in destabilized transcripts, but not in stabilized ones. Please perform this analysis! Also, the authors should provide a combined list of genes that fulfil the following criteria: a) bound by ZCCHC24, b) altered in their steady-state level (up or down) and c) consistently altered in their stability (increased or decreased, respectively) upon ZCCHC24 depletion. This would help to identify direct, high-confidence ZCCHC24 targets.

3) In line 128 the authors state: "Based on the RNA-Seq results, 124 genes with destabilized transcripts were 129 identified as differently expressed genes with destabilized transcripts (Dataset EV3)."

Where does this number come from? What genes are these? The authors point the reader to Data Set EV3 which contains 1985 genes...

Referee #2:

The authors addressed my concerns but issues of cherry-picking, and others raised by R1 are important and hopefully were satisfactorily addressed in this revised version.

Responses to reviewer

Thank you for considering our manuscript. Following the reviewers' comments, we have reconsidered the manuscript and performed some reanalyses and revisions as follows:

1) The authors now provide a list of differentially expressed genes after ZCCHC24 depletion in MDA-MB-231 cells and a PDX model including the respective fold change and p-adj. values per gene (Data Set EV2) as requested. However, the applied cut-offs used to assemble these lists are not clear. While the list of deregulated genes in the PDX contains 1478 downregulated and 915 upregulated genes with a p-adj. below 0.001 (cut-off?), the authors included 761 downregulated and 636 upregulated genes with p-adj. even above 0.05 in the list of MDA-MB-231. In addition, the MDA-MB-231 gene list contains rows that lack the gene name but contain FC and p-values. The authors should align their data presentation and filtering to common scientific standards, i.e. either include the full list of genes (with FC and p-value) or use standard cut-offs, e.g. p-adj.<0.05. They need to mention these cut-offs so that the readers of the manuscript can understand the data.

However, I need to be quite frank about this: Never before have I experienced something like this as a reviewer (for a prestigious journal like EMBO Reports) and it raises severe doubts about the data quality and validity of the analyses due to the non-scientific presentation and selection of the data throughout this manuscript!

Thank you for this comment. We originally set the threshold in the transcriptome analysis for MDAMB231 and PDX cells such that the approximate number of DEGs would be the same, resulting in different thresholds between the two datasets. In addition, some genes did not attribute the gene name because of the lack of gene names for some transcripts (long non-coding RNA), which we overlooked. We have reanalyzed and modified these issues as follows: Following the reviewer's comment, we standardized the cutoffs for the transcriptome data of both datasets (MDAMB231 and PDX) as follows: downregulated genes with log2 fold change < -0.4 and padj < 0.05 and upregulated genes with log2 fold change > 0.4 and padj < 0.05. The reanalysis revealed that MDAMB231 cells contained 584 downregulated and 438 upregulated genes. For PDX, there were 1,795 downregulated genes and 1,453 upregulated genes. Among the commonly regulated genes, 352 were downregulated, and 192 were upregulated. The lists of DEGs, log2 fold changes, and Padj values are contained in **Dataset EV2**. We also listed the log2 fold changes and Padj values for all genes in **Dataset EV2**. We have also changed **Fig 2A** to reflect the changes in cutoffs and the legend of **Fig 2A** as follows:

Line 1088 - 1095 (Legend for Fig 2A)

A. Distribution plots of RNA-seq analysis for the TNBC cell line MDAMB231 or patient-derived xenografts (patient #1) knocked down with ZCCHC24 siRNA. Differences in gene expression between siNC and siZCCHC24_1 were tested using DESeq2 (<https://bioconductor.org/packages/release/bioc/html/DESeq2.html>) following the manufacturer's protocol. The cutoff for the determination of differentially expressed genes (DEGs) was as follows: log2 fold change < -0.4 and padj < 0.05 for downregulated genes (colored in orange) and log2 fold change > 0.4, padj < 0.05 for upregulated genes (gene lists are shown in **Dataset EV3**). (Three biological replicates per group).

2) In line with this, the authors still do not provide any explanation why they did not investigate the more interesting extremes, but instead focused on (cherry-picked) transcripts whose stability changed only marginally. At least, the authors now include a list of genes whose stability was decreased upon ZCCHC24 knockdown (Data Set EV3). However, the other side of the coin is missing - the list of genes whose stability is increased upon ZCCHC24! Inclusion of these genes would be important in order to evaluate whether the putative ZCCHC24 binding motif, which was identified via PAR-CLIP is specifically enriched in destabilized transcripts, but not in stabilized ones. Please perform this analysis!

Also, the authors should provide a combined list of genes that fulfil the following criteria: a) bound by ZCCHC24, b) altered in their steady-state level (up or down) and c) consistently altered in their stability (increased or decreased, respectively) upon ZCCHC24 depletion. This would help to identify direct, high-confidence ZCCHC24 targets.

Thank you for your comment. Following the reviewer's suggestion, we listed the genes whose stability increased upon ZCCHC24 knockdown in **Dataset EV3**. We also added the half-lives of all genes to **Dataset EV3**. Additionally, we searched for the motif sequence identified by PAR-CLIP (UGUWHWWA) in the 3' UTR of all genes. We investigated its correlation with the results of transcriptome analysis and BRIC-Seq for MDAMB231. As a result, as shown in the figure below (**Appendix Fig S6B**), the motif sequence was significantly enriched in the 3' UTR of downregulated DEGs. In contrast, the number of motif sequences tended to be lower in upregulated DEGs. Thanks to Reviewer #1's insightful comments, we more clearly demonstrated the importance of the "UGUWHWWA" motifs in ZCCHC24-mediated expression regulation. We have now considered this figure in the revised manuscript. We have also listed the transcriptome analysis results and the number of motif sites in the 3' UTR, attached as **Dataset EV5**.

However, as shown in the figure below, it became apparent that there was no correlation between the half-lives identified by the BRIC-Seq and the number of motif sequences. Since the BRIC-Seq measures the half-life of genes with low expression levels, it may be more susceptible to noise, and we believe that this result reflects these effects. However, for the group of genes that we focused upon, we demonstrated that mRNA stability decreases with ZCCHC24 knockdown using experiments with actinomycin D. Therefore, we believe that our conclusions, together with the analysis of the transcriptome data mentioned above, are valid.

Moreover, following the reviewer's comment, we provided a list of genes that fulfilled the following criteria: a) bound by ZCCHC24, b) altered in their steady-state level (up or down), and c) consistently altered in their stability (increased or decreased, respectively) upon ZCCHC24 depletion. As a result, we identified "pure targets" as shown in **Dataset EV6**: 16 genes were negatively regulated, and 31 genes were positively regulated by ZCCHC24 expression. Among these 31 genes, ZEB1 and NOTCH2 were included, supporting our hypothesis. Following these considerations and reanalysis, we changed our manuscript as follows;

Line144-154

The violin plot also showed that the number of binding sites of ZCCHC24 correlated with expression changes upon ZCCHC24 knockdown (**Fig. 3E, Dataset EV4**), and the motif sites within 3'UTR of mRNAs were enriched in downregulated DEGs in transcriptome analysis (**Appendix Fig. S6B, Dataset EV5**), supporting the validity of the analysis. Importantly, peak binding sites were observed for ZCCHC24 on the motif sequences on 3'UTRs of genes critical for the characterization of the CSC population (**Fig. 3F**). We identified 31 pure target genes (bound in PAR-CLIP, stabilized or destabilized in BRIC-Seq, and upregulated or downregulated in RNA-Seq), including ZEB1 and NOTCH2, that were positively regulated by ZCCHC24 expression, and 16 genes that were negatively regulated by ZCCHC24 expression (**Dataset EV6**).

Line 585-591

Counting the number of motif sites in 3'UTR of genes

Motif sequences “UGUWHWWA” within 3'UTR of mRNAs on all genes were searched on Human Refseq release 215 (Nov, 2022) with GGGenome software (https://gggenome.dbcls.jp/hnm_refseq215/+/UGUWHWWA). The results were acquired as bed files, and the number of motif sites was calculated for each transcript. Representative transcripts for each gene were selected using MANE software (<https://www.ncbi.nlm.nih.gov/refseq/MANE/>).

3) In line 128 the authors state: "Based on the RNA-Seq results, 124 genes with destabilized transcripts were 129 identified as differently expressed genes with destabilized transcripts (Dataset EV3)."

Where does this number come from? What genes are these? The authors point the reader to Data Set EV3 which contains 1985 genes...

We appreciate the reviewer's comments. Following the first comment, we have re-identified differentially expressed genes with destabilized transcripts. We identified 93 downregulated genes with destabilized transcripts and 126 upregulated genes with stabilized transcripts and listed the gene names in **Dataset EV3**. Following these changes, we performed GO analysis for the 93 downregulated genes with destabilized transcripts, and the results are shown in (**Fig. EV1D**).

Fortunately, we can reach the same conclusion as the one we reached in the former threshold.

Term	P-value
negative regulation of apoptotic process	7.1.E-05
positive regulation of cardiac epithelial to mesenchymal transition	1.3.E-04
angiogenesis	1.3.E-04
maintenance of blood-brain barrier	4.3.E-04
positive regulation of peptidyl-tyrosine phosphorylation	5.0.E-04
negative regulation of extrinsic apoptotic signaling pathway	6.5.E-04
positive regulation of transcription by RNA polymerase II	1.0.E-03
cell-matrix adhesion	1.4.E-03
cell migration	1.5.E-03

Following these changes, we also changed the manuscript as follows;

Line 126-131

Based on the RNA-Seq results, 93 genes with destabilized transcripts were identified as differentially expressed genes (DEGs) with destabilized transcripts (**Dataset EV3**). Gene Ontology (GO) analysis of these 93 genes with DAVID showed that genes function with “negative regulation of the apoptotic process,” “angiogenesis,” “cell-matrix adhesion,” and “cell migration,” which are important for cancer stemness, are regulated by ZCCHC24 (**Fig. EV1D**).

Prof. Hiroshi Asahara
Tokyo Medical and Dental University
Department of Systems BioMedicine
1-5-45 Yushima, Bunkyo-ku
Tokyo 113-8510
Japan

Dear Prof. Asahara,

Thanks for the submission of your further revised manuscript. As you can see below, referee #1 is now satisfied and supports the publication of your study. I am thus very pleased to accept your manuscript for publication in the next available issue of EMBO reports. Thank you for your contribution to our journal.

Yours sincerely,

Referee #1:

The authors answered all my remaining questions and provided the requested data. I have no further comments.
